# Stochastic Runge-Kutta Accelerates
# Langevin Monte Carlo and Beyond

**Xuechen Li[1,2], Denny Wu[1,2], Lester Mackey[3], Murat A. Erdogdu[1,2]**
University of Toronto[1], Vector Institute[2], Microsoft Research[3]
{lxuechen, dennywu, erdogdu}@cs.toronto.edu, lmackey@microsoft.com

## Abstract

Sampling with Markov chain Monte Carlo methods often amounts to discretizing some continuous-time dynamics with numerical integration. In this paper, we establish the convergence rate of sampling algorithms obtained by discretizing smooth Itô diffusions exhibiting fast Wasserstein-2 contraction, based on local deviation properties of the integration scheme. In particular, we study a sampling algorithm constructed by discretizing the overdamped Langevin diffusion with the method of stochastic Runge-Kutta. For strongly convex potentials that are smooth up to a certain order, its iterates converge to the target distribution in 2-Wasserstein distance in $\tilde{\mathcal{O}}(d\epsilon^{-2/3})$ iterations. This improves upon the best-known rate for strongly log-concave sampling based on the overdamped Langevin equation using only the gradient oracle without adjustment. In addition, we extend our analysis of stochastic Runge-Kutta methods to uniformly dissipative diffusions with possibly non-convex potentials and show they achieve better rates compared to the Euler-Maruyama scheme in terms of the dependence on tolerance $\epsilon$. Numerical studies show that these algorithms lead to better stability and lower asymptotic errors.

## 1 Introduction

Sampling from a probability distribution is a fundamental problem that arises in machine learning, statistics, and optimization. In many situations, the goal is to obtain samples from a target distribution given only the unnormalized density [2, 27, 40]. A prominent approach to this problem is the method of Markov chain Monte Carlo (MCMC), where an ergodic Markov chain is simulated so that iterates converge exactly or approximately to the distribution of interest [43, 2].

MCMC samplers based on numerically integrating continuous-time dynamics have proven very useful due to their ability to accommodate a stochastic gradient oracle [65]. Moreover, when used as optimizations algorithms, these methods can deliver strong theoretical guarantees in non-convex settings [50]. A popular example in this regime is the unadjusted Langevin Monte Carlo (LMC) algorithm [51]. Fast mixing of LMC is inherited from exponential Wasserstein decay of the Langevin diffusion, and numerical integration using the Euler-Maruyama scheme with a sufficiently small step size ensures the Markov chain tracks the diffusion. Asymptotic guarantees of this algorithm are well-studied [51, 26, 42], and non-asymptotic analyses specifying explicit constants in convergence bounds were recently conducted [14, 11, 18, 7, 20, 9].

To the best of our knowledge, the best known rate of LMC in 2-Wasserstein distance is due to Durmus and Moulines [18] – $\tilde{\mathcal{O}}(d\epsilon^{-1})$ iterations are required to reach $\epsilon$ accuracy to $d$-dimensional target distributions with strongly convex potentials under the additional Lipschitz Hessian assumption, where $\tilde{\mathcal{O}}$ hides insubstantial poly-logarithmic factors. Due to its simplicity and well-understood theoretical properties, LMC and its derivatives have found numerous applications in statistics and machine learning [65, 15]. However, from the numerical integration point of view, the Euler-Maruyama

scheme is usually less preferred for many problems due to its inferior stability compared to implicit schemes [1] and large integration error compared to high-order schemes [46].

In this paper, we study the convergence rate of MCMC samplers devised from discretizing Itô diffusions with exponential Wasserstein-2 contraction. Our result provides a general framework for establishing convergence rates of existing numerical schemes in the SDE literature when used as sampling algorithms. In particular, we establish *non-asymptotic* convergence bounds for sampling with *stochastic Runge-Kutta* (SRK) methods. For strongly convex potentials, iterates of a variant of SRK applied to the overdamped Langevin diffusion has a convergence rate of $\tilde{\mathcal{O}}(d\epsilon^{-2/3})$. Similar to LMC, the algorithm only queries the gradient oracle of the potential during each update and improves upon the best known rate of $\tilde{\mathcal{O}}(d\epsilon^{-1})$ for strongly log-concave sampling based on the overdamped Langevin diffusion without Metropolis adjustment, under the mild extra assumption that the potential is smooth up to the third order. In addition, we extend our analysis to *uniformly dissipative* diffusions, which enables sampling from non-convex potentials by choosing a non-constant diffusion coefficient. We study a different variant of SRK and obtain the convergence rate of $\tilde{\mathcal{O}}(d^{3/4}m^2\epsilon^{-1})$ for general Itô diffusions, where $m$ is the dimensionality of the Brownian motion. This improves upon the convergence rate of $\tilde{\mathcal{O}}(d\epsilon^{-2})$ for the Euler-Maruyama scheme in terms of the tolerance $\epsilon$, while potentially trading off dimension dependence.

Our contributions can be summarized as follows:
- We provide a broadly applicable theorem for establishing convergence rates of sampling algorithms based on discretizing Itô diffusions exhibiting exponential Wasserstein-2 contraction to the target invariant measure. The convergence rate is explicitly expressed in terms of the contraction rate of the diffusion and local properties of the numerical scheme, both of which can be easily derived.
- We show for strongly convex potentials, a variant of SRK applied to the overdamped Langevin diffusion achieves the improved convergence rate of $\tilde{\mathcal{O}}(d\epsilon^{-2/3})$ by accessing only the gradient oracle, under mild additional smoothness conditions on the potential.
- We establish the convergence rate of a different variant of SRK applied to uniformly dissipative diffusions. By choosing an appropriate diffusion coefficient, we show the corresponding algorithm can sample from certain non-convex potentials and achieves the rate of $\tilde{\mathcal{O}}(d^{3/4}m^2\epsilon^{-1})$.
- We provide examples and numerical studies of sampling from both convex and non-convex potentials with SRK methods and show they lead to better stability and lower asymptotic errors.

## 1.1 Additional Related Work

**High-Order Schemes.** Numerically solving SDEs has been a research area for decades [46, 32]. We refer the reader to [3] for a review and to [32] for technical foundations. Chen et al. [5] studied the convergence of smooth functions evaluated at iterates of sampling algorithms obtained by discretizing the Langevin diffusion with high-order numerical schemes. Their focus was on convergence rates of function evaluations under a stochastic gradient oracle using asymptotic arguments. This convergence assessment pertains to analyzing numerical schemes in the weak sense. By contrast, we establish non-asymptotic convergence bounds in the 2-Wasserstein metric, which covers a broader class of functions by the Kantorovich duality [28, 62], and our techniques are based on the mean-square convergence analysis of numerical schemes. Notably, a key ingredient in the proofs by Chen et al. [5], i.e. moment bounds in the guise of a Lyapunov function argument, is assumed without justification, whereas we derive this formally and obtain convergence bounds with explicit dimension dependent constants. Durmus et al. [19] considered convergence of function evaluations of schemes obtained using Richardson-Romberg extrapolation. Sabanis and Zhang [53] introduced a numerical scheme that queries the gradient of the Laplacian based on an integrator that accommodates superlinear drifts [54]. In particular, for potentials with a Lipschitz gradient, they obtained the convergence rate of $\tilde{\mathcal{O}}(d^{4/3}\epsilon^{-2/3})$. In optimization, high-order ordinary differential equation (ODE) integration schemes were introduced to discretize a second-order ODE and achieved acceleration [68].

**Non-Convex Learning.** The convergence analyses of sampling using the overdamped and underdamped Langevin diffusion were extended to the non-convex setting [9, 39]. For the Langevin diffusion, the most common assumption on the potential is strong convexity outside a ball of finite radius, in addition to Lipschitz smoothness and twice differentiability [9, 38, 39]. More generally, Vempala and Wibisono [61] showed that convergence in the KL divergence of LMC can be derived assuming a log-Sobolev inequality of the target measure with a positive log-Sobolev constant holds. For general Itô diffusions, the notion of *distant dissipativity* [30, 22, 23] is used to study convergence

Table 1: Convergence rates in $W_2$ for algorithms sampling from strongly convex potentials by discretizing the overdamped Langevin diffusion. "Oracle" refers to highest derivative used in the update. "Smoothness" refers to Lipschitz conditions. Note that faster algorithms exist by discretizing high-order Langevin equations [13, 8, 9, 47, 56] or applying Metropolis adjustment [21, 6].

| Method | Convergence Rate | Oracle | Smoothness |
|---|---|---|---|
| Euler-Maruyama [18] | $\tilde{\mathcal{O}}(d\epsilon^{-2})$ | 1st order | gradient |
| Euler-Maruyama [18] | $\tilde{\mathcal{O}}(d\epsilon^{-1})$ | 1st order | gradient & Hessian |
| Ozaki's [11] [1] | $\tilde{\mathcal{O}}(d\epsilon^{-1})$ | 2nd order | gradient & Hessian |
| Tamed Order 1.5 [53] [2] | $\tilde{\mathcal{O}}(d^{4/3}\epsilon^{-2/3})$ | 3rd order | 1st to 3rd derivatives |
| **Stochastic Runge-Kutta (this work)** | $\tilde{\mathcal{O}}(d\epsilon^{-2/3})$ | 1st order | 1st to 3rd derivatives |

to target measures with non-convex potentials in the 1-Wasserstein distance. Different from these works, our non-convex convergence analysis, due to conducted in $W_2$, requires the slightly stronger uniform dissipativity condition [30]. In optimization, non-asymptotic results for stochastic gradient Langevin dynamics and its variants have been established for non-convex objectives [50, 67, 24, 69].

**Notation.** We denote the $p$-norm of a real vector $x \in \mathbb{R}^d$ by $\|x\|_p$. For a function $f : \mathbb{R}^d \to \mathbb{R}$, we denote its $i$th derivative by $\nabla^i f(x)$ and its Laplacian by $\Delta f = \sum_{i=1}^d \partial^2 f_i(x)/\partial x_i^2$. For a vector-valued function $g : \mathbb{R}^d \to \mathbb{R}^m$, we denote its vector Laplacian by $\vec{\Delta}(g)$, i.e. $\vec{\Delta}(g)_i = \Delta(g_i)$. For a tensor $T \in \mathbb{R}^{d_1 \times d_2 \times \cdots \times d_m}$, we define its operator norm recursively as $\|T\|_{\mathrm{op}} = \sup_{\|u\|_2 \le 1} \|T[u]\|_{\mathrm{op}}$, where $T[u]$ denotes the tensor-vector product. For $f$ sufficiently differentiable, we denote the Lipschitz and polynomial coefficients of its $i$th order derivative as

$$\mu_0(f) = \sup_{x \in \mathbb{R}^d} \|f(x)\|_{\mathrm{op}}, \quad \mu_i(f) = \sup_{x,y \in \mathbb{R}^d, x \ne y} \frac{\|\nabla^{i-1} f(x) - \nabla^{i-1} f(y)\|_{\mathrm{op}}}{\|x-y\|_2}, \text{ and } \pi_{i,n}(f) = \sup_{x \in \mathbb{R}^d} \frac{\|\nabla^{i-1} f(x)\|_{\mathrm{op}}^n}{1+\|x\|_2^n},$$

with the exception in Theorem 3, where $\pi_{1,n}(\sigma)$ is used for a sublinear growth condition. We denote Lipschitz and growth coefficients under the Frobenius norm $\|\cdot\|_{\mathrm{F}}$ as $\mu_1^{\mathrm{F}}(\cdot)$ and $\pi_{1,n}^{\mathrm{F}}(\cdot)$, respectively.

**Coupling and Wasserstein Distance.** We denote by $\mathcal{B}(\mathbb{R}^d)$ the Borel $\sigma$-field of $\mathbb{R}^d$. Given probability measures $\nu$ and $\nu'$ on $(\mathbb{R}^d, \mathcal{B}(\mathbb{R}^d))$, we define a coupling (or transference plan) $\zeta$ between $\nu$ and $\nu'$ as a probability measure on $(\mathbb{R}^d \times \mathbb{R}^d, \mathcal{B}(\mathbb{R}^d \times \mathbb{R}^d))$ such that $\zeta(A \times \mathbb{R}^d) = \nu(A)$ and $\zeta(\mathbb{R}^d \times A) = \nu'(A)$ for all $A \in \mathcal{B}(\mathbb{R}^d)$. Let couplings$(\nu, \nu')$ denote the set of all such couplings. We define the 2-Wasserstein distance between a pair of probability measures $\nu$ and $\nu'$ as

$$W_2(\nu, \nu') = \inf_{\zeta \in \mathrm{couplings}(\nu, \nu')} \left( \int \|x-y\|_2^2 \, \mathrm{d}\zeta(\nu, \nu') \right)^{1/2}.$$

## 2 Sampling with Discretized Diffusions

We study the problem of sampling from a target distribution $p(x)$ with the help of a candidate Itô diffusion [37, 44] given as the solution to the following stochastic differential equation (SDE):

$$\mathrm{d}X_t = b(X_t) \, \mathrm{d}t + \sigma(X_t) \, \mathrm{d}B_t, \quad \text{with} \quad X_0 = x_0, \tag{1}$$

where $b : \mathbb{R}^d \to \mathbb{R}^d$ and $\sigma : \mathbb{R}^d \to \mathbb{R}^{d \times m}$ are termed as the drift and diffusion coefficients, respectively. Here, $\{B_t\}_{t \ge 0}$ is an $m$-dimensional Brownian motion adapted to the filtration $\{\mathcal{F}_t\}_{t \ge 0}$, whose $i$th dimension we denote by $\{B_t^{(i)}\}_{t \ge 0}$. A candidate diffusion should be chosen so that (i) its invariant measure is the target distribution $p(x)$ and (ii) it exhibits fast mixing properties. Under mild conditions, one can design a diffusion with the target invariant measure by choosing the drift coefficient as (see e.g. [37, Thm. 2])

$$b(x) = \frac{1}{2p(x)} \langle \nabla, p(x)w(x) \rangle, \quad \text{where} \quad w(x) = \sigma(x)\sigma(x)^\top + c(x), \tag{2}$$

$c(x) \in \mathbb{R}^{d \times d}$ is any skew-symmetric matrix and $\langle \nabla, \cdot \rangle$ is the divergence operator for a matrix-valued function, i.e. $\langle \nabla, w(x) \rangle_i = \sum_{j=1}^d \partial w_{i,j}(x) / \partial x_j$ for $w : \mathbb{R}^d \to \mathbb{R}^{d \times d}$. To guarantee that this diffusion has fast convergence properties, we will require certain dissipativity conditions to be introduced later. For example, if the target is the Gibbs measure of a strongly convex potential $f : \mathbb{R}^d \to \mathbb{R}$, i.e., $p(x) \propto \exp(-f(x))$, a popular candidate diffusion is the (overdamped) Langevin diffusion which is the solution to the following SDE:

$$\mathrm{d}X_t = -\nabla f(X_t) \, \mathrm{d}t + \sqrt{2} \, \mathrm{d}B_t, \quad \text{with} \quad X_0 = x_0. \tag{3}$$

It is straightforward to verify (2) for the above diffusion which implies that the target $p(x)$ is its invariant measure. Moreover, strong convexity of $f$ implies uniform dissipativity and ensures that the diffusion achieves fast convergence.

## 2.1 Numerical Schemes and the Itô-Taylor Expansion

In practice, the Itô diffusion (1) (similarly (3)) cannot be simulated in continuous time and is instead approximated by a discrete-time numerical integration scheme. Owing to its simplicity, a common choice is the Euler-Maruyama (EM) scheme [32], which relies on the following update rule,

$$\tilde{X}_{k+1} = \tilde{X}_k + h \, b(\tilde{X}_k) + \sqrt{h} \, \sigma(\tilde{X}_k) \xi_{k+1}, \quad k = 0, 1, \ldots, \tag{4}$$

where $h$ is the step size and $\xi_{k+1} \overset{\text{i.i.d.}}{\sim} \mathcal{N}(0, I_d)$ is independent of $\tilde{X}_k$ for all $k \in \mathbb{N}$. The above iteration defines a Markov chain and due to discretization error, its invariant measure $\tilde{p}(x)$ is different from the target distribution $p(x)$; yet, for a sufficiently small step size, the difference between $\tilde{p}(x)$ and $p(x)$ can be characterized (see e.g. [42, Thm. 7.3]).

Analogous to ODE solvers, numerical schemes such as the EM scheme and SRK schemes are derived based on approximating the continuous-time dynamics locally. Similar to the standard Taylor expansion, Itô's lemma induces a stochastic version of the Taylor expansion of a smooth function evaluated at a stochastic process at time $t$. This is known as the Itô-Taylor (or Wagner-Platen) expansion [46], and one can also interpret the expansion as recursively applying Itô's lemma to terms in the integral form of an SDE. Specifically, for $g : \mathbb{R}^d \to \mathbb{R}^d$, we define the operators:

$$L(g)(x) = \nabla g(x) \cdot b(x) + \tfrac{1}{2} \sum_{i=1}^m \nabla^2 g(x)[\sigma_i(x), \sigma_i(x)], \quad \Lambda_j(g)(x) = \nabla g(x) \cdot \sigma_j(x), \tag{5}$$

where $\sigma_i(x)$ denotes the $i$th column of $\sigma(x)$. Then, applying Itô's lemma to the integral form of the SDE (1) with the starting point $X_0$ yields the following expansion around $X_0$ [32, 46]:

$$X_t = \overbrace{\underbrace{X_0 + t \, b(X_0) + \sigma(X_0) B_t}_{\text{Euler-Maruyama update}} + \sum_{i,j=1}^m \int_0^t \int_0^s \Lambda_j(\sigma_i)(X_u) \, \mathrm{d}B_u^{(j)} \, \mathrm{d}B_u^{(i)}}^{\text{mean-square order 1.0 stochastic Runge-Kutta update}} + \int_0^t \int_0^s L(b)(X_u) \, \mathrm{d}u \, \mathrm{d}s$$

$$+ \sum_{i=1}^m \int_0^t \int_0^s L(\sigma_i)(X_u) \, \mathrm{d}u \, \mathrm{d}B_s^{(i)} + \sum_{i=1}^m \int_0^t \int_0^s \Lambda_i(b)(X_u) \, \mathrm{d}B_u^{(i)} \, \mathrm{d}s. \tag{6}$$

The expansion justifies the update rule of the EM scheme, since the discretization is nothing more than taking the first three terms on the right hand side of (6). Similarly, a mean-square order 1.0 SRK scheme for general Itô diffusions – introduced in Section 4.2 – approximates the first four terms. In principle, one may recursively apply Itô's lemma to terms in the expansion to obtain a more fine-grained approximation. However, the appearance of non-Gaussian terms in the guise of iterated Brownian integrals presents a challenge for simulation. Nevertheless, it is clear that the above SRK scheme will be a more accurate local approximation than the EM scheme, due to accounting more terms in the expansion. As a result, the local deviation between the continuous-time process and Markov chain will be smaller. We characterize this property of a numerical scheme as follows.

**Definition 2.1** (Uniform Local Deviation Orders). *Let $\{\tilde{X}_k\}_{k \in \mathbb{N}}$ denote the discretization of an Itô diffusion $\{X_t\}_{t \geq 0}$ based on a numerical integration scheme with constant step size $h$, and its governing Brownian motion $\{B_t\}_{t \geq 0}$ be adapted to the filtration $\{\mathcal{F}_t\}_{t \geq 0}$. Suppose $\{X_s^{(k)}\}_{s \geq 0}$ is another instance of the same diffusion starting from $\tilde{X}_{k-1}$ at $s = 0$ and governed by the Brownian motion $\{B_{s+h(k-1)}\}_{s \geq 0}$. Then, the numerical integration scheme has local deviation $D_h^{(k)} = \tilde{X}_k - X_h^{(k)}$ with uniform orders $(p_1, p_2)$ if*

$$\mathcal{E}_k^{(1)} = \mathbb{E}\left[\mathbb{E}[\|D_h^{(k)}\|_2^2 | \mathcal{F}_{t_{k-1}}]\right] \leq \lambda_1 h^{2p_1}, \quad \mathcal{E}_k^{(2)} = \mathbb{E}\left[\left\|\mathbb{E}[D_h^{(k)} | \mathcal{F}_{t_{k-1}}]\right\|_2^2\right] \leq \lambda_2 h^{2p_2}, \tag{7}$$

*for all $k \in \mathbb{N}_+$ and $0 \leq h < C_h$, where constants $0 < \lambda_1, \lambda_2, C_h < \infty$. We say that $\mathcal{E}_k^{(1)}$ and $\mathcal{E}_k^{(2)}$ are the local mean-square deviation and the local mean deviation at iteration $k$, respectively.*

In the SDE literature, local deviation orders are defined to derive the mean-square order (or strong order) of numerical schemes [46], where the mean-square order is defined as the maximum half-integer $p$ such that $\mathbb{E}[\|X_{t_k} - \tilde{X}_k\|_2^2] \leq C h^{2p}$ for a constant $C$ independent of step size $h$ and all $k \in \mathbb{N}$ where $t_k < T$. Here, $\{X_t\}_{t \geq 0}$ is the continuous-time process, $\tilde{X}_k (k = 0, 1, \dots)$ is the Markov chain with the same Brownian motion as the continuous-time process, and $T < \infty$ is the terminal time. The key difference between our definition of *uniform* local deviation orders and local deviation orders in the SDE literature is we require the extra step of ensuring the expectations of $\varepsilon_k^{(1)}$ and $\varepsilon_k^{(2)}$ are bounded across all iterations, instead of merely requiring the two deviation variables to be bounded by a function of the previous iterate.

## 3 Convergence Rates of Numerical Schemes for Sampling

We present a user-friendly and broadly applicable theorem that establishes the convergence rate of a diffusion-based sampling algorithm. We develop our explicit bounds in the 2-Wasserstein distance based on two crucial steps. We first verify that the candidate diffusion exhibits exponential Wasserstein-2 contraction and thereafter compute the uniform local deviation orders of the scheme.

**Definition 3.1** (Wasserstein-2 rate). *A diffusion $X_t$ has Wasserstein-2 ($W_2$) rate $r : \mathbb{R}_{\geq 0} \to \mathbb{R}$ if for two instances of the diffusion $X_t$ initiated respectively from $x$ and $y$, we have*

$$W_2(\delta_x P_t, \delta_y P_t) \leq r(t) \|x - y\|_2, \quad \text{for all } x, y \in \mathbb{R}^d, t \geq 0,$$

*where $\delta_x P_t$ denotes the distribution of the diffusion $X_t$ starting from $x$. Moreover, if $r(t) = e^{-\alpha t}$ for some $\alpha > 0$, then we say the diffusion has exponential $W_2$-contraction.*

The above condition guarantees fast mixing of the sampling algorithm. For Itô diffusions, uniform dissipativity suffices to ensure exponential $W_2$-contraction $r(t) = e^{-\alpha t}$ [24, Prop. 3.3].

**Definition 3.2** (Uniform Dissipativity). *A diffusion defined by (1) is $\alpha$-uniformly dissipative if*

$$\langle b(x) - b(y), x - y \rangle + \tfrac{1}{2} \|\sigma(x) - \sigma(y)\|_{\mathrm{F}}^2 \leq -\alpha \|x - y\|_2^2, \quad \text{for all } x, y \in \mathbb{R}^d.$$

For Itô diffusions with a constant diffusion coefficient, uniform dissipativity is equivalent to one-sided Lipschitz continuity of the drift with coefficient $-2\alpha$. In particular, for the overdamped Langevin diffusion (3), this reduces to strong convexity of the potential. Moreover, for this special case, exponential $W_2$-contraction of the diffusion and strong convexity of the potential are equivalent [4]. We will ultimately verify uniform dissipativity for the candidate diffusions, but we first use $W_2$-contraction to derive the convergence rate of a diffusion-based sampling algorithm.

**Theorem 1** ($W_2$-rate of a numerical scheme). *For a diffusion with invariant measure $\nu^*$, exponentially contracting $W_2$-rate $r(t) = e^{-\alpha t}$, and Lipschitz drift and diffusion coefficients, suppose its discretization based on a numerical integration scheme has uniform local deviation orders $(p_1, p_2)$ where $p_1 \geq 1/2$ and $p_2 \geq p_1 + 1/2$. Let $\nu_k$ be the measure associated with the Markov chain obtained from the discretization after $k$ steps starting from the dirac measure $\nu_0 = \delta_{x_0}$. Then, for constant step size $h$ satisfying*

$$h < 1 \wedge C_h \wedge \frac{1}{2\alpha} \wedge \frac{1}{8\mu_1(b)^2 + 8\mu_1^{\mathrm{F}}(\sigma)^2},$$

*where $C_h$ is the step size constraint for obtaining the uniform local deviation orders, we have*

$$W_2(\nu_k, \nu^*) \leq \left(1 - \frac{\alpha h}{2}\right)^k W_2(\nu_0, \nu^*) + \left(\frac{8\,(16\mu_1(b)\lambda_1 + \lambda_2)}{\alpha^2} + \frac{2\lambda_1}{\alpha}\right)^{1/2} h^{p_1 - 1/2}. \tag{8}$$

*Moreover, if $p_1 > 1/2$ and the step size additionally satisfies*

$$h < \left(\frac{2}{\epsilon}\sqrt{\frac{64(16\lambda_1\mu_1(b) + \lambda_2)}{\alpha^2} + \frac{2\lambda_1}{\alpha}}\right)^{-1/(p_1 - 1/2)},$$

*then $W_2(\nu_k, \nu^*)$ converges in $\tilde{\mathcal{O}}(\epsilon^{-1/(p_1 - 1/2)})$ iterations within a sufficiently small positive error $\epsilon$.*

Theorem 1 directly translates mean-square order results in the SDE literature to convergence rates of sampling algorithms in $W_2$. The proof deferred to Appendix A follows from an inductive argument

over the local deviation at each step (see e.g. [46]), and the convergence is provided by the exponential $W_2$-contraction of the diffusion. To invoke the theorem and obtain convergence rates of a sampling algorithm, it suffices to (i) show that the candidate diffusion is uniformly dissipative and (ii) derive the local deviation orders for the underlying discretization. Below, we demonstrate this on both the overdamped Langevin and general Itô diffusions when the EM scheme is used for discretization, as well as the underdamped Langevin diffusion when a linearization is used for discretization [8]. For these schemes, local deviation orders are either well-known or straightforward to derive. Thus, convergence rates for corresponding sampling algorithms can be easily obtained using Theorem 1.

**Example 1.** Consider sampling from a target distribution whose potential is strongly convex using the overdamped Langevin diffusion (3) discretized by the EM scheme. The scheme has local deviation of orders $(1.5, 2.0)$ for Itô diffusions with constant diffusion coefficients and drift coefficients that are sufficiently smooth [3] (see e.g. [46, Sec. 1.5.4]). Since the potential is strongly convex, the Langevin diffusion is uniformly dissipative and achieves exponential $W_2$-contraction [18, Prop. 1]. Elementary algebra shows that Markov chain moments are bounded [24, Lem. A.2]. Therefore, Theorem 1 implies that the rate of the sampling is $\tilde{\mathcal{O}}(d\epsilon^{-1})$, where the dimension dependence can be extracted from the explicit bound. This recovers the result by Durmus and Moulines [18, Thm. 8].

**Example 2.** If a general Itô diffusion (1) with Lipschitz smooth drift and diffusion coefficients is used for the sampling task, local deviation orders of the EM scheme reduce to $(1.0, 1.5)$ due to the approximation of the diffusion term [46] – this term is exact for Langevin diffusion. If we further have uniform dissipativity, it can be shown that Markov chain moments are bounded [24, Lem. A.2]. Hence, Theorem 1 concludes that the convergence rate is $\tilde{\mathcal{O}}(d\epsilon^{-2})$. We note that for the diffusion coefficient, we use the Frobenius norm for the Lipschitz and growth constants which potentially hides dimension dependence factors. The dimension dependence worsens if one were to convert all bounds to be based on the operator norm using the pessimistic inequality $\|\sigma(x)\|_{\mathrm{F}} \leq (d^{1/2} + m^{1/2}) \|\sigma(x)\|_{\mathrm{op}}$. Appendix D provides a convergence bound with explicit constants.

**Example 3.** Consider sampling from a target distribution whose potential is strongly convex using the underdamped Langevin diffusion:

$$\mathrm{d}X_t = V_t \, \mathrm{d}t, \quad \mathrm{d}V_t = -\gamma V_t \, \mathrm{d}t - u\nabla f(X_t) \, \mathrm{d}t + \sqrt{2\gamma u} \, \mathrm{d}B_t.$$

Cheng et al. [8] show that the continuous-time process $\{(X_t, X_t + V_t)\}_{t \geq 0}$ exhibits exponential $W_2$-contraction when the coefficients $\gamma$ and $u$ are appropriately chosen [8, Thm. 5]. Moreover, the scheme devised by linearizing the degenerate SDE has uniform local deviation orders $(1.5, 2.0)$ [4] [8, Thm. 9]. Theorem 1 implies that the convergence rate is $\mathcal{O}(d^{1/2}\epsilon^{-1})$, where the dimension dependence is extracted from explicit bounds. This recovers the result by Cheng et al. [8, Thm. 1].

While computing the local deviation orders of a numerical scheme for a single step is often straightforward, it is not immediately clear how one might verify them uniformly for each iteration. This requires a uniform bound on moments of the Markov chain defined by the numerical scheme. As our second principal contribution, we explicitly bound the Markov chain moments of SRK schemes which, combined with Theorem 1, leads to improved rates by only accessing the first-order oracle.

## 4 Sampling with Stochastic Runge-Kutta and Improved Rates

We show that convergence rates of sampling can be significantly improved if an Itô diffusion with exponential $W_2$-contraction is discretized using SRK methods. Compared to the EM scheme, SRK schemes we consider query the same order oracle and improve on the deviation orders.

Theorem 1 hints that one may expect the convergence rate of sampling to improve as more terms of the Itô-Taylor expansion are incorporated in the numerical integration scheme. However, in practice, a challenge for simulation is the appearance of non-Gaussian terms in the form of iterated Itô integrals. Fortunately, since the overdamped Langevin diffusion has a constant diffusion coefficient, efficient SRK methods can still be applied to accelerate convergence.

## 4.1 Sampling from Strongly Convex Potentials with the Langevin Diffusion

We provide a non-asymptotic analysis for integrating the overdamped Langevin diffusion based on a mean-square order 1.5 SRK scheme for SDEs with constant diffusion coefficients [46]. We refer to the sampling algorithm as SRK-LD. Specifically, given a sample from the previous iteration $\tilde{X}_k$,

$$\tilde{H}_1 = \tilde{X}_k + \sqrt{2h}\left[\left(\frac{1}{2} + \frac{1}{\sqrt{6}}\right)\xi_{k+1} + \frac{1}{\sqrt{12}}\eta_{k+1}\right],$$

$$\tilde{H}_2 = \tilde{X}_k - h\nabla f(\tilde{X}_k) + \sqrt{2h}\left[\left(\frac{1}{2} - \frac{1}{\sqrt{6}}\right)\xi_{k+1} + \frac{1}{\sqrt{12}}\eta_{k+1}\right],$$

$$\tilde{X}_{k+1} = \tilde{X}_k - \frac{h}{2}\left(\nabla f(\tilde{H}_1) + \nabla f(\tilde{H}_2)\right) + \sqrt{2h}\xi_{k+1}, \tag{9}$$

where $h$ is the step size and $\xi_{k+1}, \eta_{k+1} \overset{\text{i.i.d.}}{\sim} \mathcal{N}(0, I_d)$ are independent of $\tilde{X}_k$ for all $k \in \mathbb{N}$. We refer the reader to [46, Sec. 1.5] for a detailed derivation of the scheme and other background information.

**Theorem 2** (SRK-LD). *Let $\nu^*$ be the target distribution with a strongly convex potential that is four-times differentiable with Lipschitz continuous first three derivatives. Let $\nu_k$ be the distribution of the kth Markov chain iterate defined by (9) starting from the dirac measure $\nu_0 = \delta_{x_0}$. Then, for a sufficiently small step size, 1.5 SRK scheme has uniform local deviation orders $(2.0, 2.5)$, and $W_2(\nu_k, \nu^*)$ converges within $\epsilon$ error in $\tilde{\mathcal{O}}(d\epsilon^{-2/3})$ iterations.*

The proof of this theorem is given in Appendix B where we provide explicit constants. The basic idea of the proof is to match up the terms in the Itô-Taylor expansion to terms in the Taylor expansion of the discretization scheme. However, extreme care is needed to ensure a tight dimension dependence.

***Remark.*** For large-scale Bayesian inference, computing the full gradient of the potential can be costly. Fortunately, SRK-LD can be easily adapted to use an unbiased stochastic oracle, provided queries of the latter have a variance not overly large. We provide an informal discussion in Appendix E.

We emphasize that the 1.5 SRK scheme (9) only queries the gradient of the potential and improves the best available $W_2$-rate of LMC in the same setting from $\tilde{\mathcal{O}}(d\epsilon^{-1})$ to $\tilde{\mathcal{O}}(d\epsilon^{-2/3})$, with merely two extra gradient evaluations per iteration. Remarkably, the dimension dependence stays the same.

## 4.2 Sampling from Non-Convex Potentials with Itô Diffusions

For the Langevin diffusion, the conclusions of Theorem 1 only apply to distributions with strongly convex potentials, as exponential $W_2$-contraction of the Langevin diffusion is equivalent to strong convexity of the potential. This shortcoming can be addressed using a non-constant diffusion coefficient which allows us to sample from non-convex potentials using uniformly dissipative candidate diffusions. Below, we use a mean-square order 1.0 SRK scheme for general diffusions [52] and achieve an improved convergence rate compared to sampling with the EM scheme.

We refer to the sampling algorithm as SRK-ID, which has the following update rule:

$$\tilde{H}_1^{(i)} = \tilde{X}_k + \sum_{j=1}^m \sigma_l(\tilde{X}_k)\frac{I_{(j,i)}}{\sqrt{h}}, \qquad \tilde{H}_2^{(i)} = \tilde{X}_k - \sum_{j=1}^m \sigma_l(\tilde{X}_k)\frac{I_{(j,i)}}{\sqrt{h}},$$

$$\tilde{X}_{k+1} = \tilde{X}_k + hb(\tilde{X}_k) + \sum_{i=1}^m \sigma_i(\tilde{X}_k)I_{(i)} + \frac{\sqrt{h}}{2}\sum_{i=1}^m\left(\sigma_i(\tilde{H}_1^{(i)}) - \sigma_i(\tilde{H}_2^{(i)})\right), \tag{10}$$

where $I_{(i)} = \int_{t_k}^{t_{k+1}} dB_s^{(i)}$, $I_{(j,i)} = \int_{t_k}^{t_{k+1}}\int_{t_k}^s dB_u^{(j)} dB_s^{(i)}$. We note that schemes of higher order exist for general diffusions, but they typically require advanced approximations of iterated Itô integrals of the form $\int_0^{t_0}\cdots\int_0^{t_{n-1}} dB_{t_n}^{(k_n)}\cdots dB_{t_1}^{(k_1)}$.

**Theorem 3** (SRK-ID). *For a uniformly dissipative diffusion with invariant measure $\nu_*$, Lipschitz drift and diffusion coefficients that have Lipschitz gradients, assume that the diffusion coefficient further satisfies the sublinear growth condition $\|\sigma(x)\|_{\text{op}} \leq \pi_{1,1}(\sigma)\left(1 + \|x\|_2^{1/2}\right)$ for all $x \in \mathbb{R}^d$. Let $\nu_k$ be the distribution of the kth Markov chain iterate defined by (10) starting from the dirac measure $\nu_0 = \delta_{x_0}$. Then for a sufficiently small step size, iterates of the 1.0 SRK scheme have uniform local deviation orders $(1.5, 2.0)$, and $W_2(\nu_k, \nu^*)$ converges within $\epsilon$ error in $\tilde{\mathcal{O}}(d^{3/4}m^2\epsilon^{-1})$ iterations.*

The proof is given in Appendix C where we present explicit constants. We note that the dimension dependence in this case is only better than that of EM due to the extra growth condition on the diffusion. The extra $m$-dependence comes from the $2m$ evaluations of the diffusion coefficient at $\tilde{H}_1^{(i)}$ and $\tilde{H}_2^{(i)}$ $(i = 1, \ldots, m)$. In the above theorem, we use the Frobenius norm for the Lipschitz and growth constants for the diffusion coefficient which potentially hides dimension dependence. One may convert all bounds to be based on the operator norm with our constants given in the Appendix.

In practice, accurately simulating both the iterated Itô integrals $I_{(j,i)}$ and the Brownian motion increments $I_{(i)}$ simultaneously is difficult. We comment on two possible approximations based on truncating an infinite series in Appendix H.2.

# 5 Examples and Numerical Studies

We provide examples of our theory and numerical studies showing SRK methods achieve lower asymptotic errors, are stable under large step sizes, and hence converge faster to a prescribed tolerance. We sample from strongly convex potentials with SRK-LD and non-convex potentials with SRK-ID. Since our theory is in $W_2$, we compare with EM on $W_2$ and mean squared error (MSE) between iterates of the Markov chain and the target. We do not compare to schemes that require computing derivatives of the drift and diffusion coefficients. Since directly computing $W_2$ is infeasible, we estimate it using samples instead. However, sample-based estimators have a bias of order $\Omega(n^{-1/d})$ [64], so we perform a heuristic correction whose description is in Appendix G.

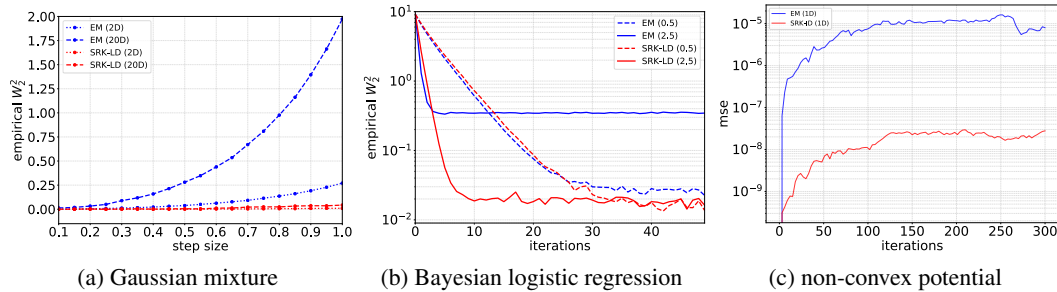

(a) Gaussian mixture      (b) Bayesian logistic regression      (c) non-convex potential

Figure 1: (a) Estimated asymptotic error against step size. (b) Estimated error against number of iterations. (c) MSE against number of iterations. Legends of (a) and (c) denote "scheme (dimensionality)". Legend of (b) denotes "scheme (step size)".

## 5.1 Strongly Convex Potentials

**Gaussian Mixture.** We consider sampling from a multivariate Gaussian mixture with density

$$\pi(\theta) \propto \exp\left(-\tfrac{1}{2}\|\theta - a\|_2^2\right) + \exp\left(-\tfrac{1}{2}\|\theta + a\|_2^2\right), \quad \theta \in \mathbb{R}^d,$$

where $a \in \mathbb{R}^d$ is a parameter that measures the separation of two modes. The potential is strongly convex when $\|a\|_2 < 1$ and has Lipschitz gradient and Hessian [11]. Moreover, one can verify that its third derivative is also Lipschitz.

**Bayesian Logistic Regression.** We consider Bayesian logistic regression (BLR) [11]. Given data samples $X = \{x_i\}_{i=1}^n \in \mathbb{R}^{n \times d}$, $Y = \{y_i\}_{i=1}^n \in \mathbb{R}^n$, and parameter $\theta \in \mathbb{R}^d$, logistic regression models the Bernoulli conditional distribution with probability $\Pr(y_i = 1 | x_i) = 1/(1 + \exp(-\theta^\top x_i))$. We place a Gaussian prior on $\theta$ with mean zero and covariance proportional to $\Sigma_X^{-1}$, where $\Sigma_X = X^\top X/n$ is the sample covariance matrix. We sample from the posterior density

$$\pi(\theta) \propto \exp(-f(\theta)) = \exp\left(Y^\top X\theta - \sum_{i=1}^n \log(1 + \exp(-\theta^\top x_i)) - \tfrac{\alpha}{2}\|\Sigma_X^{1/2}\theta\|_2^2\right).$$

The potential is strongly convex and has Lipschitz gradient and Hessian [11]. One can also verify that it has a Lipschitz third derivative.

To obtain the potential, we generate data from the model with the parameter $\theta_* = \mathbf{1}_d$ following [11, 21]. To obtain each $x_i$, we sample a vector whose components are independently drawn from the

Rademacher distribution and normalize it by the Frobenius norm of the sample matrix $X$ times $d^{-1/2}$. Note that our normalization scheme is different from that adopted in [11, 21], where each $x_i$ is normalized by its Euclidean norm. We sample the corresponding $y_i$ from the model and fix the regularizer $\alpha = 0.3d/\pi^2$.

To characterize the true posterior, we sample 50k particles driven by EM with a step size of $0.001$ until convergence. We subsample from these particles 5k examples to represent the true posterior each time we intend to estimate squared $W_2$. We monitor the kernel Stein discrepancy [5] (KSD) [29, 10, 36] using the inverse multiquadratic kernel [29] with hyperparameters $\beta = -1/2$ and $c = 1$ to measure the distance between the 100k particles and the true posterior. We confirm that these particles faithfully approximate the true posterior with the squared KSD being less than $0.002$ in all settings.

When sampling from a Gaussian mixture and the posterior of BLR, we observe that SRK-LD leads to a consistent improvement in the asymptotic error compared to the EM scheme when the same step size is used. In particular, Figure 1 (a) plots the estimated asymptotic error in squared $W_2$ of different step sizes for 2D and 20D Gaussian mixture problems and shows that SRK-LD is surprisingly stable for exceptionally large step sizes. Figure 1 (b) plots the estimated error in squared $W_2$ as the number of iterations increases for 2D BLR. We include additional results on problems in 2D and 20D with error estimates in squared $W_2$ and the energy distance [58] along with a wall time analysis in Appendix H.

## 5.2 Non-Convex Potentials

We consider sampling from the non-convex potential
$$f(x) = \left(\beta + \|x\|_2^2\right)^{1/2} + \gamma \log\left(\beta + \|x\|_2^2\right), \quad x \in \mathbb{R}^d,$$
where $\beta, \gamma > 0$ are scalar parameters of the distribution. The corresponding density is a simplified abstraction for the posterior distribution of Student's t regression with a pseudo-Huber prior [30]. One can verify that when $\beta + \|x\|_2^2 < 1$ and $(4\gamma + 1)\|x\|_2^2 < (2\gamma + 1)\sqrt{\beta + \|x\|_2^2}$, the Hessian has a negative eigenvalue. The candidate diffusion, where the drift coefficient is given by (2) and diffusion coefficient $\sigma(x) = g(x)^{1/2} I_d$ with $g(x) = \left(\beta + \|x\|_2^2\right)^{1/2}$, is uniformly dissipative if $\frac{1}{2} - |\gamma - \frac{1}{2}|\frac{2}{\beta^{1/2}} - \frac{d}{8\beta^{1/2}} > 0$. Indeed, one can verify that $\mu_1(g) \le 1$, $\mu_2(g) \le \frac{2}{\beta^{1/2}}$, and $\mu_1(\sigma) \le \frac{1}{2\beta^{1/4}}$. Therefore,

$$\langle b(x) - b(y), x - y \rangle + \tfrac{1}{2}\|\sigma(x) - \sigma(y)\|_F^2 \le -\left(\tfrac{1}{2} - |\gamma - \tfrac{1}{2}|\mu_2(g) - \tfrac{d}{2}\mu_1(\sigma)^2\right)\|x - y\|_2^2,$$
$$\le -\left(\tfrac{1}{2} - |\gamma - \tfrac{1}{2}|\tfrac{2}{\beta^{1/2}} - \tfrac{d}{8\beta^{1/2}}\right)\|x - y\|_2^2.$$

Moreover, $b$ and $\sigma$ have Lipschitz first two derivatives, and the latter satisfies the sublinear growth condition in Theorem 3.

To study the behavior of SRK-ID, we simulate using both SRK-ID and EM. For both schemes, we simulate with a step size of $10^{-3}$ initiated from the same 50k particles approximating the stationary distribution obtained by simulating EM with a step size of $10^{-6}$ until convergence. We compute the MSE between the continuous-time process and the Markov chain with the same Brownian motion for 300 iterations when we observe the MSE curve plateaus. We approximate the continuous-time process by simulating using the EM scheme with a step size of $10^{-6}$ similar to the setting in [52]. To obtain final results, we average across ten independent runs. We note that the MSE upper bounds $W_2$ due to the latter being an infimum over all couplings. Hence, the MSE value serves as an indication of the convergence performance in $W_2$.

Figure 1 (c) shows that for $\beta = 0.33$, $\gamma = 0.5$ and $d = 1$, when simulating from a good approximation to the target distribution with the same step size, the MSE of SRK-ID remains small, whereas the MSE of EM converges to a larger value. However, this improvement diminishes as the dimensionality of the sampling problem increases. We report additional results with other parameter settings in Appendix H.2.2. Notably, we did not observe significant differences in the estimated squared $W_2$ values. We suspect this is due to the discrepancy being dominated by the bias of our estimator.

## Acknowledgments

MAE is partially funded by NSERC [2019-06167] and CIFAR AI Chairs program at the Vector Institute.

## Footnotes

[1] We obtain a rate in $W_2$ from the discretization analysis in KL [11] via standard techniques [50, 61].

[2] Sabanis and Zhang [53] use the Frobenius norm for matrices and the Euclidean norm of Frobenius norms for 3-tensors. For a fair comparison, we convert their Lipschitz constants to be based on the operator norm.

[3]In fact, it suffices to ensure the drift is three-times differentiable with Lipschitz gradient and Hessian.

[4]Cheng et al. [8] derive the uniform local mean-square deviation order. Jensen's inequality implies that the local mean deviation is of the same uniform order. This entails uniform local deviation orders are $(2.0, 2.0)$ and hence also $(1.5, 2.0)$ when step size constraint $C_h \leq 1$; note $p_2 \geq p_1 + 1/2$ is required to invoke Theorem 1.

[5]Unfortunately, there appear to be two definitions for KSD and the energy distance in the literature, differing in whether a square root is taken or not. We adopt the version with the square root taken.

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
