[Supplementary Material]

# A   Proof of Theorem 1

*Proof.* Let $\{X_t\}_{t\geq 0}$ denote the continuous-time process defined by the SDE (1) initiated from the target stationary distribution, driven by the Brownian motion $\{B_t\}_{t\geq 0}$. Since the continuous-time transition kernel preserves the stationary distribution, the marginal distribution of $\{X_t\}_{t\geq 0}$ remains to be the stationary distribution for all $t \geq 0$.

We denote by $t_k$ $(k = 0, 1, \dots)$ the timestamps of the Markov chain obtained by discretizing the continuous-time process with a numerical integration scheme and assume the Markov chain has a constant step size $h$ that satisfies the conditions in the theorem statement. We denote by $\tilde{X}_k$ the $k$th iterate of the Markov chain. In the following, we derive a recursion for the quantity

$$A_k = \mathbb{E}\left[\left\|X_{t_k} - \tilde{X}_k\right\|_2^2\right]^{1/2}.$$

Fix $k \in \mathbb{N}$. We define the process $\{\bar{X}_t\}_{t\geq 0}$ such that it is the Markov chain until $t_k$, starting from which it follows the continuous-time process defined by the SDE (1). We let $\{\bar{X}_t\}_{t\geq 0}$ and the Markov chain $\tilde{X}_k$ $(k = 0, 1, \dots)$ share the same Brownian motion $\{\bar{B}_t\}_{t\geq 0}$. Suppose $\{\mathcal{F}_t\}_{t\geq 0}$ is a filtration to which both $\{B_t\}_{t\geq 0}$ and $\{\bar{B}_t\}_{t\geq 0}$ are adapted. Conditional on $\mathcal{F}_{t_k}$, let $X_{t_{k+1}}$ and $\bar{X}_{t_{k+1}}$ be coupled such that

$$\mathbb{E}\left[\left\|X_{t_{k+1}} - \bar{X}_{t_{k+1}}\right\|_2^2 | \mathcal{F}_{t_k}\right] \leq e^{-2\alpha h}\left\|X_{t_k} - \bar{X}_{t_k}\right\|_2^2. \tag{11}$$

This we can achieve due to exponential $W_2$-contraction. We define the process $\{Z_s\}_{s\geq t_k}$ as follows

$$Z_s = \left(X_s - \bar{X}_s\right) - \left(X_{t_k} - \bar{X}_{t_k}\right).$$

Note $\int_{t_k}^{t_k+t} \sigma(X_s)\,\mathrm{d}B_s - \int_{t_k}^{t_k+t} \sigma(\bar{X}_s)\,\mathrm{d}\bar{B}_s$ is a Martingale w.r.t. $\{\mathcal{F}_{t_k+t}\}_{t\geq 0}$, since it is adapted and the two component Itô integrals are Martingales w.r.t. the considered filtration. By Fubini's theorem, we switch the order of integrals and obtain

$$\mathbb{E}\left[Z_{t_{k+1}}|\mathcal{F}_{t_k}\right] = \int_{t_k}^{t_{k+1}} \mathbb{E}\left[b(X_s) - b(\bar{X}_s)|\mathcal{F}_{t_k}\right]\,\mathrm{d}s.$$

By Jensen's inequality,

$$\left\|\mathbb{E}\left[Z_{t_{k+1}}|\mathcal{F}_{t_k}\right]\right\|_2^2 \leq h \int_{t_k}^{t_{k+1}} \mathbb{E}\left[\left\|b(X_s) - b(\bar{X}_s)\right\|_2^2 |\mathcal{F}_{t_k}\right]\,\mathrm{d}s$$

$$\leq \mu_1(b)^2 h \int_{t_k}^{t_{k+1}} \mathbb{E}\left[\left\|X_s - \bar{X}_s\right\|_2^2 |\mathcal{F}_{t_k}\right]\,\mathrm{d}s. \tag{12}$$

For $s \in [t_k, t_k + h]$, by Young's inequality, Jensen's inequality, and Itô isometry,

$$\mathbb{E}\left[\left\|X_s - \bar{X}_s\right\|_2^2 |\mathcal{F}_{t_k}\right]$$

$$= \mathbb{E}\left[\left\|X_{t_k} - \bar{X}_{t_k} + \int_{t_k}^s \left(b(X_u) - b(\bar{X}_u)\right)\,\mathrm{d}u + \int_{t_k}^s \left(\sigma(X_u) - \sigma(\bar{X}_u)\right)\,\mathrm{d}B_u\right\|_2^2 |\mathcal{F}_{t_k}\right]$$

$$\leq 4\left\|X_{t_k} - \bar{X}_{t_k}\right\|_2^2 + 4(s - t_k)\int_{t_k}^s \mathbb{E}\left[\left\|b(X_u) - b(\bar{X}_u)\right\|_2^2 |\mathcal{F}_{t_k}\right]\,\mathrm{d}u$$

$$+ 4\int_{t_k}^s \mathbb{E}\left[\left\|\sigma(X_u) - \sigma(\bar{X}_u)\right\|_F^2 |\mathcal{F}_{t_k}\right]\,\mathrm{d}u$$

$$\leq 4\left\|X_{t_k} - \bar{X}_{t_k}\right\|_2^2 + 4(s - t_k)\mu_1(b)^2 \int_{t_k}^s \mathbb{E}\left[\left\|X_u - \bar{X}_u\right\|_2^2 |\mathcal{F}_{t_k}\right]\,\mathrm{d}u$$

$$+ 4\mu_1^F(\sigma)^2 \int_{t_k}^s \mathbb{E}\left[\left\|X_u - \bar{X}_u\right\|_2^2 |\mathcal{F}_{t_k}\right]u$$

$$\leq 4\left\|X_{t_k} - \bar{X}_{t_k}\right\|_2^2 + 4\left(\mu_1(b)^2 + \mu_1^F(\sigma)^2\right)\int_{t_k}^s \mathbb{E}\left[\left\|X_u - \bar{X}_u\right\|_2^2 |\mathcal{F}_{t_k}\right]\,\mathrm{d}u.$$

By the integral form of Grönwall's inequality for continuous functions,

$$\mathbb{E}\left[\left\|X_s - \bar{X}_s\right\|_2^2 \big| \mathcal{F}_{t_k}\right] \leq 4\exp\left(4\left(\mu_1(b)^2 + \mu_1^{\mathrm{F}}(\sigma)^2\right)(s - t_k)\right)\left\|X_{t_k} - \bar{X}_{t_k}\right\|_2^2.$$

Plugging this result into (12), by $h < 1/\left(8\mu_1(b)^2 + 8\mu_1^{\mathrm{F}}(\sigma)^2\right)$,

$$\begin{aligned}
\left\|\mathbb{E}\left[Z_{t_{k+1}} | \mathcal{F}_{t_k}\right]\right\|_2^2 &\leq \frac{\mu_1(b)^2 h}{\mu_1(b)^2 + \mu_1^{\mathrm{F}}(\sigma)^2}\left[\exp\left(4\left(\mu_1(b)^2 + \mu_1^{\mathrm{F}}(\sigma)^2\right)h\right) - 1\right]\left\|X_{t_k} - \bar{X}_{t_k}\right\|_2^2 \\
&\leq \frac{8\mu_1(b)^2 h^2}{\mu_1(b)^2 + \mu_1^{\mathrm{F}}(\sigma)^2}\left(\mu_1(b)^2 + \mu_1^{\mathrm{F}}(\sigma)^2\right)\left\|X_{t_k} - \bar{X}_{t_k}\right\|_2^2 \\
&\leq 8\mu_1(b)^2 h^2 \left\|X_{t_k} - \bar{X}_{t_k}\right\|_2^2.
\end{aligned} \tag{13}$$

By direct expansion,

$$\mathbb{E}\left[\left\|X_{t_{k+1}} - \bar{X}_{t_{k+1}}\right\|_2^2 \big| \mathcal{F}_{t_k}\right] = \left\|X_{t_k} - \bar{X}_{t_k}\right\|_2^2 + \mathbb{E}\left[\left\|Z_{t_{k+1}}\right\|_2^2 \big| \mathcal{F}_{t_k}\right] + 2\left\langle X_{t_k} - \bar{X}_{t_k}, \mathbb{E}\left[Z_{t_{k+1}} | \mathcal{F}_{t_k}\right]\right\rangle. \tag{14}$$

Combining (11) (13) and (14), by the Cauchy-Schwarz inequality,

$$\begin{aligned}
\mathbb{E}\left[\left\|Z_{t_{k+1}}\right\|_2^2 \big| \mathcal{F}_{t_k}\right] &\leq \left(e^{-2\alpha h} - 1\right)\left\|X_{t_k} - \bar{X}_{t_k}\right\|_2^2 - 2\left\langle X_{t_k} - \bar{X}_{t_k}, \mathbb{E}\left[Z_{t_{k+1}} | \mathcal{F}_{t_k}\right]\right\rangle \\
&\leq 2\left\|X_{t_k} - \bar{X}_{t_k}\right\|_2 \left\|\mathbb{E}\left[Z_{t_{k+1}} | \mathcal{F}_{t_k}\right]\right\|_2 \\
&\leq 8\mu_1(b)h\left\|X_{t_k} - \bar{X}_{t_k}\right\|_2^2 \\
&= 8\mu_1(b)h\left\|X_{t_k} - \tilde{X}_k\right\|_2^2.
\end{aligned}$$

Hence,

$$\mathbb{E}\left[\left\|Z_{t_{k+1}}\right\|_2^2\right] = \mathbb{E}\left[\mathbb{E}\left[\left\|Z_{t_{k+1}}\right\|_2^2 \big| \mathcal{F}_{t_k}\right]\right] \leq 8\mu_1(b)h\mathbb{E}\left[\left\|X_{t_k} - \tilde{X}_k\right\|_2^2\right] = 8\mu_1(b)hA_k^2.$$

Let $\lambda_3 = 8\lambda_1^{1/2}\mu_1(b)^{1/2} + 2\lambda_2^{1/2}$. Then, by the Cauchy–Schwarz inequality, we obtain a recursion

$$\begin{aligned}
A_{k+1}^2 =& \mathbb{E}\left[\left\|X_{t_{k+1}} - \tilde{X}_{k+1}\right\|_2^2\right] \\
=& \mathbb{E}\left[\left\|X_{t_{k+1}} - \bar{X}_{t_{k+1}} + \bar{X}_{t_{k+1}} - \tilde{X}_{k+1}\right\|_2^2\right] \\
=& \mathbb{E}\left[\left\|X_{t_{k+1}} - \bar{X}_{t_{k+1}}\right\|_2^2 + \left\|\bar{X}_{t_{k+1}} - \tilde{X}_{k+1}\right\|_2^2 + 2\left\langle X_{t_{k+1}} - \bar{X}_{t_{k+1}}, \bar{X}_{t_{k+1}} - \tilde{X}_{k+1}\right\rangle\right] \\
=& \mathbb{E}\left[\mathbb{E}\left[\left\|X_{t_{k+1}} - \bar{X}_{t_{k+1}}\right\|_2^2 \big| \mathcal{F}_{t_k}\right]\right] + \mathbb{E}\left[\mathbb{E}\left[\left\|\bar{X}_{t_{k+1}} - \tilde{X}_{k+1}\right\|_2^2 \big| \mathcal{F}_{t_k}\right]\right] \\
& + 2\mathbb{E}\left[\mathbb{E}\left[\left\langle X_{t_{k+1}} - \bar{X}_{t_{k+1}}, \bar{X}_{t_{k+1}} - \tilde{X}_{k+1}\right\rangle \big| \mathcal{F}_{t_k}\right]\right] \\
=& \mathbb{E}\left[\mathbb{E}\left[\left\|X_{t_{k+1}} - \bar{X}_{t_{k+1}}\right\|_2^2 \big| \mathcal{F}_{t_k}\right]\right] + \mathbb{E}\left[\mathbb{E}\left[\left\|\bar{X}_{t_{k+1}} - \tilde{X}_{k+1}\right\|_2^2 \big| \mathcal{F}_{t_k}\right]\right] \\
& + 2\mathbb{E}\left[\left\langle X_{t_k} - \bar{X}_{t_k}, \mathbb{E}\left[\bar{X}_{t_{k+1}} - \tilde{X}_{k+1} | \mathcal{F}_{t_k}\right]\right\rangle\right] \\
& + 2\mathbb{E}\left[\left\langle Z_{t_{k+1}}, \bar{X}_{t_{k+1}} - \tilde{X}_{k+1}\right\rangle\right] \\
\leq& \mathbb{E}\left[\mathbb{E}\left[\left\|X_{t_{k+1}} - \bar{X}_{t_{k+1}}\right\|_2^2 \big| \mathcal{F}_{t_k}\right]\right] + \mathbb{E}\left[\mathbb{E}\left[\left\|\bar{X}_{t_{k+1}} - \tilde{X}_{k+1}\right\|_2^2 \big| \mathcal{F}_{t_k}\right]\right] \\
& + 2\mathbb{E}\left[\left\|X_{t_k} - \bar{X}_{t_k}\right\|_2^2\right]^{1/2}\mathbb{E}\left[\left\|\mathbb{E}\left[\bar{X}_{t_{k+1}} - \tilde{X}_{k+1} | \mathcal{F}_{t_k}\right]\right\|_2^2\right]^{1/2} \\
& + 2\mathbb{E}\left[\left\|Z_{t_{k+1}}\right\|_2^2\right]^{1/2}\mathbb{E}\left[\left\|\bar{X}_{t_{k+1}} - \tilde{X}_{k+1}\right\|_2^2\right]^{1/2}
\end{aligned}$$

$$\leq e^{-2\alpha h}A_k^2 + \lambda_1 h^{2p_1} + 2\lambda_2^{1/2}h^{p_2}A_k + 8\lambda_1^{1/2}\mu_1(b)^{1/2}h^{p_1+1/2}A_k$$
$$\leq (1-\alpha h)A_k^2 + \lambda_3 h^{p_1+1/2}A_k + \lambda_1 h^{2p_1}$$
$$\leq (1-\alpha h)A_k^2 + \frac{\alpha h}{2}A_k^2 + \frac{8}{\alpha}\lambda_3^2 h^{2p_1} + \lambda_1 h^{2p_1}$$
$$\leq (1-\alpha h/2)A_k^2 + \left(8\lambda_3^2/\alpha + \lambda_1\right)h^{2p_1}, \tag{15}$$

where the third to last inequality follows from $e^{-2\alpha h} < 1 - \alpha h$ when $\alpha h < 1/2$, and the second to last inequality follows from the elementary relation below with the choice of $\kappa = \alpha/2$

$$A_k h^{1/2} \cdot \lambda_3 h^{p_1} \leq \kappa A_k^2 h + \frac{4}{\kappa}\lambda_3^2 h^{2p_1}.$$

Let $\eta = 1 - \alpha h/2 \leq e^{-\alpha h/2} \leq 1$. By unrolling the recursion,

$$A_k^2 \leq (1-\alpha h/2)A_{k-1}^2 + \left(8\lambda_3^2/\alpha + \lambda_1\right)h^{2p_1}$$
$$\leq \eta^k A_0^2 + \left(1 + \eta + \cdots + \eta^{k-1}\right)\left(8\lambda_3^2/\alpha + \lambda_1\right)h^{2p_1}$$
$$\leq \eta^k A_0^2 + \left(8\lambda_3^2/\alpha + \lambda_1\right)h^{2p_1}/(1-\eta)$$
$$= \eta^k A_0^2 + (16\lambda_3^2/\alpha^2 + 2\lambda_1/\alpha)h^{2p_1-1}.$$

Let $\nu_k$ and $\nu^*$ be the measures associated with the $k$th iterate of the Markov chain and the target distribution, respectively. Since $W_2$ is defined as an infimum over all couplings,

$$W_2(\nu_k, \nu^*) \leq A_k \leq e^{-\alpha hk/4}A_0 + (16\lambda_3^2/\alpha^2 + 2\lambda_1/\alpha)^{1/2}h^{p_1-1/2}.$$

To ensure $W_2$ is less than some small positive tolerance $\epsilon$, we need only ensure the two terms in the above inequality are each less than $\epsilon/2$. Some simple calculations show that it suffices that

$$h < \left(\frac{2}{\epsilon}\sqrt{\frac{64(16\lambda_1\mu_1(b) + \lambda_2)}{\alpha^2} + \frac{2\lambda_1}{\alpha}}\right)^{-1/(p_1-1/2)} \wedge \frac{1}{2\alpha} \wedge \frac{1}{8\mu_1(b)^2 + 8\mu_1^F(\sigma)^2}, \tag{16}$$

$$k > \left[\left(\frac{2}{\epsilon}\sqrt{\frac{64(16\lambda_1\mu_1(b) + \lambda_2)}{\alpha^2} + \frac{2\lambda_1}{\alpha}}\right)^{1/(p_1-1/2)} \vee 2\alpha \vee \left(8\mu_1(b)^2 + 8\mu_1^F(\sigma)^2\right)\right]\frac{4}{\alpha}\log\left(\frac{2A_0}{\epsilon}\right).$$

Note that for small enough positive tolerance $\epsilon$, when the step size satisfies (16), it suffices that

$$k = \left\lceil\left(\frac{2}{\epsilon}\sqrt{\frac{64(16\lambda_1\mu_1(b) + \lambda_2)}{\alpha^2} + \frac{2\lambda_1}{\alpha}}\right)^{1/(p_1-1/2)}\frac{4}{\alpha}\log\left(\frac{2A_0}{\epsilon}\right)\right\rceil = \tilde{\mathcal{O}}(\epsilon^{-1/(p_1-1/2)}).$$

$\square$

# B  Proof of Theorem 2

## B.1  Moment Bounds

Verifying the order conditions in Theorem 1 for SRK-LD requires bounding the second, fourth, and sixth moments of the Markov chain. In principle, one may employ an exponential moment bound argument using a Lyapunov function. However, in this case, the tightness of the final convergence bound may depend on the selection of the Lyapunov function, and reasoning about the dimension dependence can become less obvious. Here, we directly bound all the even moments by expanding the expression. Intuitively, one expects the $2n$th moments of the Markov chain iterates to be $\mathcal{O}(d^n)$. The following proofs assume Lipschitz smoothness of the potential to a certain order and dissipativity.

**Definition B.1** (Dissipativity). *For constants $\alpha, \beta > 0$, the diffusion satisfies the following*

$$\langle\nabla f(x), x\rangle \geq \frac{\alpha}{2}\|x\|_2^2 - \beta, \quad \forall x \in \mathbb{R}^d.$$

For the Langevin diffusion, dissipativity directly follows from strong convexity of the potential [24]. Here, $\alpha$ can be chosen as the strong convexity parameter, provided $\beta$ is an appropriate constant of order $\mathcal{O}(d)$.

Additionally, we assume the discretization has a constant step size $h$ and the timestamp of the $k$th iterate is $t_k$ as per the proof of Theorem 1. To simplify notation, we define the following

$$\tilde{\nabla} f = \frac{1}{2} \left( \nabla f(\tilde{H}_1) + \nabla f(\tilde{H}_2) \right),$$

$$v_1 = \sqrt{2} \left( \frac{1}{2} + \frac{1}{\sqrt{6}} \right) \xi_{k+1} \sqrt{h},$$

$$v_1' = \sqrt{2} \left( \frac{1}{2} - \frac{1}{\sqrt{6}} \right) \xi_{k+1} \sqrt{h},$$

$$v_2 = \frac{1}{\sqrt{6}} \eta_{k+1} \sqrt{h},$$

where $\xi_{k+1}, \eta_{k+1} \overset{\text{i.i.d.}}{\sim} \mathcal{N}(0, I_d)$ independent of $\tilde{X}_k$ for all $k \in \mathbb{N}$. We rewrite $\tilde{H}_1$ and $\tilde{H}_2$ as

$$\tilde{H}_1 = \tilde{X}_k + \Delta \tilde{H}_1 = \tilde{X}_k + v_1 + v_2,$$
$$\tilde{H}_2 = \tilde{X}_k + \Delta \tilde{H}_2 = \tilde{X}_k + v_1' + v_2 - \nabla f(\tilde{X}_k)h.$$

### B.1.1 Second Moment Bound

**Lemma 4.** *If the second moment of the initial iterate is finite, then the second moments of Markov chain iterates defined in* (9) *are uniformly bounded by a constant of order $\mathcal{O}(d)$, i.e.*

$$\mathbb{E}\left[ \left\| \tilde{X}_k \right\|_2^2 \right] \leq \mathcal{U}_2, \quad \text{for all } k \in \mathbb{N},$$

*where $\mathcal{U}_2 = \mathbb{E}\left[ \left\| \tilde{X}_0 \right\|_2^2 \right] + N_6$, and constants $N_1$ to $N_6$ are given in the proof, if the step size*

$$h < 1 \wedge \frac{2d}{\pi_{2,2}(f)} \wedge \frac{2\pi_{2,1}(f)}{\pi_{2,2}(f)} \wedge \frac{\alpha}{4\mu_2(f)\pi_{2,2}(f)} \wedge \frac{3\alpha}{2N_1 + 4}.$$

*Proof.* By direct computation,

$$\left\| \tilde{X}_{k+1} \right\|_2^2 = \left\| \tilde{X}_k - \left( \nabla f(\tilde{H}_1) + \nabla f(\tilde{H}_2) \right) \frac{h}{2} + 2^{1/2} \xi_{k+1} h^{1/2} \right\|_2^2$$

$$= \left\| \tilde{X}_k \right\|_2^2 + \left\| \nabla f(\tilde{H}_1) + \nabla f(\tilde{H}_2) \right\|_2^2 \frac{h^2}{4} + 2 \left\| \xi_{k+1} \right\|_2^2 h$$

$$- \left\langle \tilde{X}_k, \nabla f(\tilde{H}_1) + \nabla f(\tilde{H}_2) \right\rangle h$$

$$+ 2^{3/2} \left\langle \tilde{X}_k, \xi_{k+1} \right\rangle h^{1/2}$$

$$- 2^{1/2} \left\langle \nabla f(\tilde{H}_1) + \nabla f(\tilde{H}_2), \xi_{k+1} \right\rangle h^{3/2}.$$

In the following, we bound each term in the expansion separately and obtain a recursion. To achieve this, we first upper bound the second moments of $\tilde{H}_1$ and $\tilde{H}_2$ for $h < 2d \wedge 2\pi_{2,1}(f)/\pi_{2,2}(f)$,

$$\mathbb{E}\left[ \left\| \tilde{H}_1 \right\|_2^2 \bigg| \mathcal{F}_{t_k} \right] = \left\| \tilde{X}_k \right\|_2^2 + \mathbb{E}\left[ \left\| v_1 \right\|_2^2 \big| \mathcal{F}_{t_k} \right] + \mathbb{E}\left[ \left\| v_2 \right\|_2^2 \big| \mathcal{F}_{t_k} \right] \leq \left\| \tilde{X}_k \right\|_2^2 + 3dh,$$

$$\mathbb{E}\left[ \left\| \tilde{H}_2 \right\|_2^2 \bigg| \mathcal{F}_{t_k} \right] = \left\| \tilde{X}_k \right\|_2^2 + \left\| \nabla f(\tilde{X}_k) \right\|_2^2 h^2 + \mathbb{E}\left[ \left\| v_1' \right\|_2^2 \big| \mathcal{F}_{t_k} \right] + \mathbb{E}\left[ \left\| v_2 \right\|_2^2 \big| \mathcal{F}_{t_k} \right]$$

$$+ 2 \left\langle \tilde{X}_k, \nabla f(\tilde{X}_k) \right\rangle h$$

$$\leq \left\| \tilde{X}_k \right\|_2^2 + \pi_{2,2}(f) \left( 1 + \left\| \tilde{X}_k \right\|_2^2 \right) h^2 + dh + 2\pi_{2,1}(f) \left\| \tilde{X}_k \right\|_2^2 h$$

$$\leq \left\| \tilde{X}_k \right\|_2^2 + 4\pi_{2,1}(f)h \left\| \tilde{X}_k \right\|_2^2 + 3dh.$$

Thus,

$$\mathbb{E}\left[\left\|\nabla f(\tilde{H}_1) + \nabla f(\tilde{H}_2)\right\|_2^2 |\mathcal{F}_{t_k}\right] \leq 2\mathbb{E}\left[\left\|\nabla f(\tilde{H}_1)\right\|_2^2 + \left\|\nabla f(\tilde{H}_2)\right\|_2^2 |\mathcal{F}_{t_k}\right]$$

$$\leq 2\pi_{2,2}(f)\mathbb{E}\left[2 + \left\|\tilde{H}_1\right\|_2^2 + \left\|\tilde{H}_2\right\|_2^2 |\mathcal{F}_{t_k}\right]$$

$$= N_1 \left\|\tilde{X}_k\right\|_2^2 + N_2,$$

where $N_1 = 2\pi_{2,2}(f)\left(2 + 4\pi_{2,1}(f)\right)$ and $N_2 = 2\pi_{2,2}(f)\left(6d+2\right)$.

Additionally, by the Cauchy-Schwarz inequality,

$$-\mathbb{E}\left[\left\langle \nabla f(\tilde{H}_1), \xi_{k+1}\right\rangle |\mathcal{F}_{t_k}\right] \leq \mathbb{E}\left[\left\|\nabla f(\tilde{H}_1)\right\|_2 \|\xi_{k+1}\|_2 |\mathcal{F}_{t_k}\right]$$

$$\leq \mathbb{E}\left[\left\|\nabla f(\tilde{H}_1)\right\|_2^2 |\mathcal{F}_{t_k}\right]^{1/2} \mathbb{E}\left[\|\xi_{k+1}\|_2^2\right]^{1/2}$$

$$\leq \sqrt{d\pi_{2,2}(f)}\left(1 + \mathbb{E}\left[\left\|\tilde{H}_1\right\|_2^2 |\mathcal{F}_{t_k}\right]^{1/2}\right)$$

$$\leq \sqrt{d\pi_{2,2}(f)}\left(1 + \left\|\tilde{X}_k\right\|_2 + \sqrt{3dh}\right). \tag{17}$$

Similarly,

$$-\mathbb{E}\left[\left\langle \nabla f(\tilde{H}_2), \xi_{k+1}\right\rangle |\mathcal{F}_{t_k}\right] \leq \mathbb{E}\left[\left\|\nabla f(\tilde{H}_2)\right\|_2 \|\xi_{k+1}\|_2 |\mathcal{F}_{t_k}\right]$$

$$\leq \mathbb{E}\left[\left\|\nabla f(\tilde{H}_2)\right\|_2^2 |\mathcal{F}_{t_k}\right]^{1/2} \mathbb{E}\left[\|\xi_{k+1}\|_2^2 |\mathcal{F}_{t_k}\right]^{1/2}$$

$$\leq \sqrt{d\pi_{2,2}(f)}\left(1 + \mathbb{E}\left[\left\|\tilde{H}_2\right\|_2^2 |\mathcal{F}_{t_k}\right]^{1/2}\right)$$

$$\leq \sqrt{d\pi_{2,2}(f)}\left(1 + \left\|\tilde{X}_k\right\|_2 + 2\sqrt{\pi_{2,1}(f)h}\left\|\tilde{X}_k\right\|_2 + \sqrt{3dh}\right). \tag{18}$$

Combining (17) and (18), we obtain the following using AM–GM,

$$-2^{1/2}\mathbb{E}\left[\left\langle \nabla f(\tilde{H}_1) + \nabla f(\tilde{H}_2), \xi_{k+1}\right\rangle |\mathcal{F}_{t_k}\right] h^{3/2} \leq N_3 \left\|\tilde{X}_k\right\|_2 h^{3/2} + N_4$$

$$\leq \frac{1}{2}\left\|\tilde{X}_k\right\|_2^2 h^2 + \frac{N_3^2}{2}h + N_4 h^{3/2}.$$

where $N_3 = 2\sqrt{2d\pi_{2,2}(f)}\left(1 + \sqrt{\pi_{2,1}(f)}\right)$ and $N_4 = 2\sqrt{2d\pi_{2,2}(f)}\left(1 + \sqrt{3d}\right)$.

Now, we lower bound the second moments of $\tilde{H}_1$ and $\tilde{H}_2$ by dissipativity,

$$\mathbb{E}\left[\left\|\tilde{H}_1\right\|_2^2 |\mathcal{F}_{t_k}\right] = \mathbb{E}\left[\left\|\tilde{X}_k + v_1 + v_2\right\|_2^2 |\mathcal{F}_{t_k}\right]$$

$$= \left\|\tilde{X}_k\right\|_2^2 + \mathbb{E}\left[\|v_1\|_2^2 |\mathcal{F}_{t_k}\right] + \mathbb{E}\left[\|v_2\|_2^2 |\mathcal{F}_{t_k}\right] \geq \left\|\tilde{X}_k\right\|_2^2, \tag{19}$$

$$\mathbb{E}\left[\left\|\tilde{H}_2\right\|_2^2 |\mathcal{F}_{t_k}\right] = \mathbb{E}\left[\left\|\tilde{X}_k - \nabla f(\tilde{X}_k)h + v_1' + v_2\right\|_2^2 |\mathcal{F}_{t_k}\right]$$

$$= \left\|\tilde{X}_k\right\|_2^2 + \left\|\nabla f(\tilde{X}_k)\right\|_2^2 h^2 + \mathbb{E}\left[\|v_1'\|_2^2 |\mathcal{F}_{t_k}\right] + \mathbb{E}\left[\|v_2\|_2^2 |\mathcal{F}_{t_k}\right]$$

$$+ 2\left\langle \tilde{X}_k, \nabla f(\tilde{X}_k)\right\rangle h$$

$$\geq \left\|\tilde{X}_k\right\|_2^2 + 2\left(\frac{\alpha}{2}\left\|\tilde{X}_k\right\|_2^2 - \beta\right)h$$

$$\geq \left\| \tilde{X}_k \right\|_2^2 - 2\beta h.$$

Additionally, by Stein's lemma for multivariate Gaussians,

$$\mathbb{E}\left[\left\langle \nabla f(\tilde{H}_1), v_1 \right\rangle |\mathcal{F}_{t_k}\right] = 2h\left(\frac{1}{2} + \frac{1}{\sqrt{6}}\right)^2 \mathbb{E}\left[\Delta(f)(\tilde{H}_1)|\mathcal{F}_{t_k}\right] \leq 2d\mu_3(f)h,$$

$$\mathbb{E}\left[\left\langle \nabla f(\tilde{H}_1), v_2 \right\rangle |\mathcal{F}_{t_k}\right] = \frac{1}{6}h\mathbb{E}\left[\Delta(f)(\tilde{H}_1)|\mathcal{F}_{t_k}\right] \leq \frac{1}{6}d\mu_3(f)h,$$

$$\mathbb{E}\left[\left\langle \nabla f(\tilde{H}_2), v_1' \right\rangle |\mathcal{F}_{t_k}\right] = 2h\left(\frac{1}{2} - \frac{1}{\sqrt{6}}\right)^2 \mathbb{E}\left[\Delta(f)(\tilde{H}_2)|\mathcal{F}_{t_k}\right] \leq d\mu_3(f)h,$$

$$\mathbb{E}\left[\left\langle \nabla f(\tilde{H}_2), v_2 \right\rangle |\mathcal{F}_{t_k}\right] = \frac{1}{6}h\mathbb{E}\left[\Delta(f)(\tilde{H}_2)|\mathcal{F}_{t_k}\right] \leq d\mu_3(f)h.$$

Therefore, by dissipativity and the lower bound (19),

$$-\mathbb{E}\left[\left\langle \nabla f(\tilde{H}_1), \tilde{X}_k \right\rangle |\mathcal{F}_{t_k}\right] = -\mathbb{E}\left[\left\langle \nabla f(\tilde{H}_1), \tilde{H}_1 \right\rangle |\mathcal{F}_{t_k}\right] + \mathbb{E}\left[\left\langle \nabla f(\tilde{H}_1), v_1 + v_2 \right\rangle |\mathcal{F}_{t_k}\right]$$

$$\leq -\frac{\alpha}{2}\mathbb{E}\left[\left\| \tilde{H}_1 \right\|_2^2 |\mathcal{F}_{t_k}\right] + \beta + \mathbb{E}\left[\left\langle \nabla f(\tilde{H}_1), v_1 + v_2 \right\rangle |\mathcal{F}_{t_k}\right]$$

$$\leq -\frac{\alpha}{2}\left\| \tilde{X}_k \right\|_2^2 + \beta + 3d\mu_3(f)h. \tag{20}$$

To bound the expectation of $-\left\langle \nabla f(\tilde{H}_2), \tilde{X}_k \right\rangle$, we first bound the second moment of $\Delta \tilde{H}_2$,

$$\mathbb{E}\left[\left\| \Delta \tilde{H}_2 \right\|_2^2 |\mathcal{F}_{t_k}\right] = \mathbb{E}\left[\left\| -\nabla f(\tilde{X}_k)h + v_1' + v_2 \right\|_2^2 |\mathcal{F}_{t_k}\right]$$

$$= \left\| \nabla f(\tilde{X}_k) \right\|_2^2 h^2 + \mathbb{E}\left[\|v_1'\|_2^2 |\mathcal{F}_{t_k}\right] + \mathbb{E}\left[\|v_2\|_2^2 |\mathcal{F}_{t_k}\right]$$

$$\leq \pi_{2,2}(f)\left(1 + \left\| \tilde{X}_k \right\|_2^2\right)h^2 + dh. \tag{21}$$

Notice the second equality above also implies

$$\left\| \nabla f(\tilde{X}_k) \right\|_2 h \leq \mathbb{E}\left[\left\| \Delta \tilde{H}_2 \right\|_2^2 |\mathcal{F}_{t_k}\right]^{1/2}. \tag{22}$$

By Taylor's Theorem with the remainder in integral form,

$$\nabla f(\tilde{H}_2) = \nabla f(\tilde{X}_k) + R(t_{k+1}) = \nabla f(\tilde{X}_k) + \int_0^1 \nabla^2 f\left(\tilde{X}_k + \tau \Delta \tilde{H}_2\right)\Delta \tilde{H}_2 \,\mathrm{d}\tau.$$

Since $\nabla f$ is Lipschitz, $\nabla^2 f$ is bounded, and

$$\|R(t_{k+1})\|_2 \leq \int_0^1 \left\| \nabla^2 f\left(\tilde{X}_k + \tau \Delta \tilde{H}_2\right) \right\|_{\mathrm{op}} \left\| \Delta \tilde{H}_2 \right\|_2 \,\mathrm{d}\tau \leq \mu_2(f)\left\| \Delta \tilde{H}_2 \right\|_2.$$

By (21) and (22),

$$-\mathbb{E}\left[\left\langle \nabla f(\tilde{H}_2), \nabla f(\tilde{X}_k) \right\rangle |\mathcal{F}_{t_k}\right] = -\left\| \nabla f(\tilde{X}_k) \right\|_2^2 - \left\langle \mathbb{E}\left[R(t_{k+1})|\mathcal{F}_{t_k}\right], \nabla f(\tilde{X}_k) \right\rangle$$

$$\leq \|\mathbb{E}\left[R(t_{k+1})|\mathcal{F}_{t_k}\right]\|_2 \left\| \nabla f(\tilde{X}_k) \right\|_2$$

$$\leq \mathbb{E}\left[\|R(t_{k+1})\|_2 |\mathcal{F}_{t_k}\right]\left\| \nabla f(\tilde{X}_k) \right\|_2$$

$$\leq \mu_2(f)\mathbb{E}\left[\left\| \Delta \tilde{H}_2 \right\|_2 |\mathcal{F}_{t_k}\right]\left\| \nabla f(\tilde{X}_k) \right\|_2$$

$$\leq \mu_2(f)\mathbb{E}\left[\left\| \Delta \tilde{H}_2 \right\|_2^2 |\mathcal{F}_{t_k}\right]^{1/2}\left\| \nabla f(\tilde{X}_k) \right\|_2$$

$$\leq \mu_2(f)\mathbb{E}\left[\left\| \Delta \tilde{H}_2 \right\|_2^2 |\mathcal{F}_{t_k}\right]h^{-1}$$

$$\leq \mu_2(f)\pi_{2,2}(f)\left(1+\left\|\tilde{X}_k\right\|_2^2\right)h+d.$$

Therefore, for $h < 1 \wedge \alpha/(4\mu_2(f)\pi_{2,2}(f))$,

$$-\mathbb{E}\left[\left\langle \nabla f(\tilde{H}_2), \tilde{X}_k\right\rangle|\mathcal{F}_{t_k}\right]$$
$$=-\mathbb{E}\left[\left\langle \nabla f(\tilde{H}_2), \tilde{H}_2\right\rangle + \left\langle \nabla f(\tilde{H}_2), \nabla f(\tilde{X}_k)\right\rangle h - \left\langle \nabla f(\tilde{H}_2), v_1'+v_2\right\rangle|\mathcal{F}_{t_k}\right]$$
$$\leq -\frac{\alpha}{2}\mathbb{E}\left[\left\|\tilde{H}_2\right\|_2^2|\mathcal{F}_{t_k}\right] + \beta - \mathbb{E}\left[\left\langle \nabla f(\tilde{H}_2), \nabla f(\tilde{X}_k)\right\rangle|\mathcal{F}_{t_k}\right]h + \mathbb{E}\left[\left\langle \nabla f(\tilde{H}_2), v_1'+v_2\right\rangle|\mathcal{F}_{t_k}\right]$$
$$\leq -\frac{\alpha}{2}\left\|\tilde{X}_k\right\|_2^2 + \alpha\beta h + \beta + \mu_2(f)\pi_{2,2}(f)\left(1+\left\|\tilde{X}_k\right\|_2^2\right)h^2 + dh + 2d\mu_3(f)h$$
$$\leq -\frac{\alpha}{4}\left\|\tilde{X}_k\right\|_2^2 + (\alpha\beta + \mu_2(f)\pi_{2,2}(f) + d + 2d\mu_3(f))h + \beta. \tag{23}$$

Combining (20) and (23), we have

$$-\mathbb{E}\left[\left\langle \nabla f(\tilde{H}_1)+\nabla f(\tilde{H}_2), \tilde{X}_k\right\rangle|\mathcal{F}_{t_k}\right] \leq -\frac{3}{4}\alpha\left\|\tilde{X}_k\right\|_2^2 + N_5, \tag{24}$$

where $N_5 = (\alpha\beta + \mu_2(f)\pi_{2,2}(f) + d + 5d\mu_3(f)) + 2\beta$.

Putting things together, for $h < 3\alpha/(2N_1+4)$, we obtain

$$\mathbb{E}\left[\left\|\tilde{X}_{k+1}\right\|_2^2|\mathcal{F}_{t_k}\right] = \left\|\tilde{X}_k\right\|_2^2 + \mathbb{E}\left[\left\|\nabla f(\tilde{H}_1)+\nabla f(\tilde{H}_2)\right\|_2^2|\mathcal{F}_{t_k}\right]\frac{h^2}{4} + 2dh$$
$$- \mathbb{E}\left[\left\langle \tilde{X}_k, \nabla f(\tilde{H}_1)+\nabla f(\tilde{H}_2)\right\rangle|\mathcal{F}_{t_k}\right]h$$
$$- 2^{1/2}\mathbb{E}\left[\left\langle \nabla f(\tilde{H}_1)+\nabla f(\tilde{H}_2), \xi_{k+1}\right\rangle|\mathcal{F}_{t_k}\right]h^{3/2}$$
$$\leq \left\|\tilde{X}_k\right\|_2^2 + \frac{N_1}{4}\left\|\tilde{X}_k\right\|_2^2 h^2 + \frac{N_2}{4}h^2 + 2dh$$
$$- \frac{3}{4}\alpha h\left\|\tilde{X}_k\right\|_2^2 + N_5 h$$
$$+ \frac{1}{2}\left\|\tilde{X}_k\right\|_2^2 h^2 + \frac{N_3^2}{2}h + N_4 h^{3/2}$$
$$\leq \left(1 - \frac{3}{4}\alpha h + \frac{N_1+2}{4}h^2\right)\left\|\tilde{X}_k\right\|_2^2$$
$$+ N_2 h^2/4 + 2dh + N_5 h + N_3^2 h/2 + N_4 h^{3/2}$$
$$\leq \left(1 - \frac{3}{8}\alpha h\right)\left\|\tilde{X}_k\right\|_2^2 + N_2 h^2/4 + 2dh + N_5 h + N_3^2 h/2 + N_4 h^{3/2},$$

For $h < 1$, by unrolling the recursion, we obtain the following

$$\mathbb{E}\left[\left\|\tilde{X}_k\right\|_2^2\right] \leq \mathbb{E}\left[\left\|\tilde{X}_0\right\|_2^2\right] + N_6, \quad \text{for all } k \in \mathbb{N},$$

where

$$N_6 = \frac{1}{3\alpha}\left(2N_2 + 16d + 8N_5 + 4N_3^2 + 8N_4\right) = \mathcal{O}(d).$$

$\square$

### B.1.2 $2n$th Moment Bound

**Lemma 5.** *For $n \in \mathbb{N}_+$, if the $2n$th moment of the initial iterate is finite, then the $2n$th moments of Markov chain iterates defined in (9) are uniformly bounded by a constant of order $\mathcal{O}(d^n)$, i.e.*

$$\mathbb{E}\left[\left\|\tilde{X}_k\right\|_2^{2n}\right] \leq \mathcal{U}_{2n}, \quad \text{for all } k \in \mathbb{N},$$

*where*

$$\mathcal{U}_{2n} = \mathbb{E}\left[\left\|\tilde{X}_0\right\|_2^{2n}\right] + \frac{8}{3\alpha n}\left(N_{7,n} + N_{12,n}\right),$$

*and constants $N_{7,n}$ to $N_{12,n}$ are given in the proof, if the step size*

$$h < 1 \wedge \frac{2d}{\pi_{2,2}(f)} \wedge \frac{2\pi_{2,1}(f)}{\pi_{2,2}(f)} \wedge \frac{\alpha}{4\mu_2(f)\pi_{2,2}(f)} \wedge \frac{3\alpha}{2N_1 + 4} \wedge \min\left\{\left(\frac{3\alpha l}{8N_{11,l}}\right)^2 : l = 2,\ldots,n\right\}.$$

*Proof.* Our proof is by induction. The base case is given in Lemma 4. For the inductive case, we prove that the $2n$th moment is uniformly bounded by a constant of order $\mathcal{O}(d^n)$, assuming the $2(n\text{-}1)$th moment is uniformly bounded by a constant of order $\mathcal{O}(d^{n-1})$.

By the multinomial theorem,

$$
\begin{aligned}
\mathbb{E}\left[\left\|\tilde{X}_{k+1}\right\|_2^{2n}\right] =& \mathbb{E}\left[\left\|X_k - \tilde{\nabla}f h + 2^{1/2}\xi_{k+1}h^{1/2}\right\|_2^{2n}\right] \\
=& \mathbb{E}\Big[\Big(\|X_k\|_2^2 + \left\|\tilde{\nabla}f\right\|_2^2 h^2 + 2\|\xi_{k+1}\|_2^2 h \\
& - 2\left\langle\tilde{X}_k, \tilde{\nabla}f\right\rangle h + 2^{3/2}\left\langle\tilde{X}_k, \xi_{k+1}\right\rangle h^{1/2} - 2^{3/2}\left\langle\tilde{\nabla}f, \xi_{k+1}\right\rangle h^{3/2}\Big)^n\Big] \\
=& \mathbb{E}\Bigg[\sum_{k_1+\cdots+k_6=n}(-1)^{k_4+k_6}\binom{n}{k_1 \ \ldots \ k_6}2^{k_3+k_4+\frac{3k_5}{2}+\frac{3k_6}{2}}h^{2k_2+k_3+k_4+\frac{k_5}{2}+\frac{3k_6}{2}} \\
& \left\|\tilde{X}_k\right\|_2^{2k_1}\left\|\tilde{\nabla}f\right\|_2^{2k_2}\|\xi_{k+1}\|_2^{2k_3}\left\langle\tilde{X}_k, \tilde{\nabla}f\right\rangle^{k_4}\left\langle\tilde{X}_k, \xi_{k+1}\right\rangle^{k_5}\left\langle\tilde{\nabla}f, \xi_{k+1}\right\rangle^{k_6}\Bigg] \\
=& \mathbb{E}\left[\left\|\tilde{X}_k\right\|_2^{2n} + Ah + Bh^{3/2}\right],
\end{aligned}
$$

where

$$A = 2n\left\|\tilde{X}_k\right\|_2^{2(n-1)}\|\xi_{k+1}\|_2^2 - 2n\left\|\tilde{X}_k\right\|_2^{2(n-1)}\left\langle\tilde{X}_k, \tilde{\nabla}f\right\rangle + 4n(n-1)\left\|\tilde{X}_k\right\|_2^{2(n-2)}\left\langle\tilde{X}_k, \xi_{k+1}\right\rangle^2,$$

$$B \leq \sum_{\substack{k_1+\cdots+k_6=n \\ 2k_2+k_3+k_4+\frac{k_5}{2}+\frac{3k_6}{2}>1}} 2^{\frac{3n}{2}}\binom{n}{k_1 \ \ldots \ k_6}\left\|\tilde{X}_k\right\|_2^{2k_1+k_4+k_5}\left\|\tilde{\nabla}f\right\|_2^{2k_2+k_4+k_6}\|\xi_{k+1}\|_2^{2k_3+k_5+k_6}.$$

Now, we bound the expectation of $A$ using (24),

$$
\begin{aligned}
\mathbb{E}\left[A|\mathcal{F}_{t_k}\right] \leq& 2dn\left\|\tilde{X}_k\right\|_2^{2(n-1)} + 2n\left\|\tilde{X}_k\right\|_2^{2(n-1)}\left(-\frac{3}{8}\alpha\left\|\tilde{X}_k\right\|_2^2 + \frac{N_5}{2}\right) + 4dn(n-1)\left\|\tilde{X}_k\right\|_2^{2(n-1)} \\
\leq& -\frac{3}{4}\alpha n\left\|\tilde{X}_k\right\|_2^{2n} + (2dn + nN_5 + 4dn(n-1))\left\|\tilde{X}_k\right\|_2^{2(n-1)}.
\end{aligned}
$$

Moreover, by the inductive hypothesis,

$$\mathbb{E}[A] = \mathbb{E}\left[\mathbb{E}[A|\mathcal{F}_{t_k}]\right] \leq -\frac{3}{4}\alpha n\mathbb{E}\left[\left\|\tilde{X}_k\right\|_2^{2n}\right] + N_{7,n}, \tag{25}$$

where $N_{7,n} = (2dn + nN_5 + 4dn(n-1))\mathcal{U}_{2(n-1)} = \mathcal{O}(d^n)$.

Next, we bound the expectation of $B$. By the Cauchy–Schwarz inequality,

$$
\begin{aligned}
\mathbb{E}[B|\mathcal{F}_{t_k}] =& \sum_{\substack{k_1+\cdots+k_6=n \\ 2k_2+k_3+k_4+\frac{k_5}{2}+\frac{3k_6}{2}>1}} 2^{\frac{3n}{2}}\binom{n}{k_1 \ \ldots \ k_6}\left\|\tilde{X}_k\right\|_2^{2k_1+k_4+k_5}\mathbb{E}\left[\left\|\tilde{\nabla}f\right\|_2^{2k_2+k_4+k_6}\|\xi_{k+1}\|_2^{2k_3+k_5+k_6}\bigg|\mathcal{F}_{t_k}\right] \\
\leq& \sum_{\substack{k_1+\cdots+k_6=n \\ 2k_2+k_3+k_4+\frac{k_5}{2}+\frac{3k_6}{2}>1}} 2^{\frac{3n}{2}}\binom{n}{k_1 \ \ldots \ k_6}\left\|\tilde{X}_k\right\|_2^{2k_1+k_4+k_5}
\end{aligned}
$$

$$\times \mathbb{E}\left[\left\|\tilde{\nabla}f\right\|_2^{4k_2+2k_4+2k_6}|\mathcal{F}_{t_k}\right]^{1/2}\mathbb{E}\left[\|\xi_{k+1}\|_2^{4k_3+2k_5+2k_6}|\mathcal{F}_{t_k}\right]^{1/2}.$$

Let $\chi(d)^2$ be a chi-squared random variable with $d$ degrees of freedom. Recall its $n$th moment has a closed form solution and is of order $\mathcal{O}(d^n)$ [57]. Now, we bound the $2p$th moments of $\tilde{H}_1$ and $\tilde{H}_2$ for positive integer $p$. To achieve this, we first expand the expressions,

$$
\begin{aligned}
\left\|\tilde{H}_1\right\|_2^{2p} &= \left\|\tilde{X}_k + v_1 + v_2\right\|_2^{2p}\\
&= \left(\left\|\tilde{X}_k\right\|_2^2 + \|v_1\|_2^2 + \|v_2\|_2^2 + 2\left\langle\tilde{X}_k, v_1\right\rangle + 2\left\langle\tilde{X}_k, v_2\right\rangle + 2\left\langle v_1, v_2\right\rangle\right)^p\\
&\leq \sum_{j_1+\cdots+j_6=p} 2^{j_4+j_5+j_6}\binom{p}{j_1\ \ldots\ j_6}\left\|\tilde{X}_k\right\|_2^{2j_1+j_4+j_5}\|v_1\|_2^{2j_2+j_4+j_6}\|v_2\|_2^{2j_3+j_5+j_6}\\
&\leq \sum_{j_1+\cdots+j_6=p} 2^{j_2+\frac{3}{2}j_4+j_5+\frac{3}{2}j_6}h^{j_2+j_3+\frac{j_4}{2}+\frac{j_5}{2}+j_6}\binom{p}{j_1\ \ldots\ j_6}\left\|\tilde{X}_k\right\|_2^{2j_1+j_4+j_5}\\
&\qquad \times \|\xi_{k+1}\|_2^{2j_2+j_4+j_6}\|\eta_{k+1}\|_2^{2j_3+j_5+j_6}\\
&\leq \sum_{j_1+\cdots+j_6=p} 2^{3p}\binom{p}{j_1\ \ldots\ j_6}\left\|\tilde{X}_k\right\|_2^{2j_1+j_4+j_5}\|\xi_{k+1}\|_2^{2j_2+j_4+j_6}\|\eta_{k+1}\|_2^{2j_3+j_5+j_6}\\
&\leq \sum_{j_1+\cdots+j_6=p} 2^{3p}\binom{p}{j_1\ \ldots\ j_6}\left(\tfrac{2j_1+j_4+j_5}{2p}\left\|\tilde{X}_k\right\|_2^{2p}+\tfrac{2j_2+j_4+j_6}{2p}\|\xi_{k+1}\|_2^{2p}+\tfrac{2j_3+j_5+j_6}{2p}\|\eta_{k+1}\|_2^{2p}\right)\\
&\leq 2^{4p}3^p\left(\left\|\tilde{X}_k\right\|_2^{2p}+\|\xi_{k+1}\|_2^{2p}+\|\eta_{k+1}\|_2^{2p}\right),
\end{aligned}
$$

where the second to last inequality follows from Young's inequality for products with three variables. Therefore,

$$\mathbb{E}\left[\left\|\tilde{H}_1\right\|_2^{2p}|\mathcal{F}_{t_k}\right] \leq 2^{4p}3^p\left\|\tilde{X}_k\right\|_2^{2p}+2^{4p+1}3^p\mathbb{E}\left[\chi(d)^{2p}\right]. \tag{26}$$

Similarly,

$$
\begin{aligned}
\left\|\tilde{H}_2\right\|_2^{2p} &= \left\|\tilde{X}_k - \nabla f(\tilde{X}_k)h + v_1' + v_2\right\|_2^{2p}\\
&\leq \Big(\left\|\tilde{X}_k\right\|_2^2 + \left\|\nabla f(\tilde{X}_k)\right\|_2^2 h^2 + \|v_1'+v_2\|_2^2\\
&\qquad - 2\left\langle\tilde{X}_k, \nabla f(\tilde{X}_k)\right\rangle h + 2\left\langle\tilde{X}_k, v_1'+v_2\right\rangle - 2\left\langle\nabla f(\tilde{X}_k), v_1'+v_2\right\rangle\Big)^p\\
&\leq \sum_{j_1+\cdots+j_6=p} 2^{j_4+j_5+j_6}\binom{p}{j_1\ \ldots\ j_6}\left\|\tilde{X}_k\right\|_2^{2j_1+j_4+j_5}\left\|\nabla f(\tilde{X}_k)\right\|_2^{2j_2+j_4+j_6}\|v_1'+v_2\|_2^{2j_3+j_5+j_6}\\
&\leq 2^{4p}3^p\left(\left\|\tilde{X}_k\right\|_2^{2p}+\left\|\nabla f(\tilde{X}_k)\right\|_2^{2p}+\|\xi_{k+1}\|_2^{2p}+\|\eta_{k+1}\|_2^{2p}\right).
\end{aligned}
$$

Therefore,

$$\mathbb{E}\left[\left\|\tilde{H}_2\right\|_2^{2p}|\mathcal{F}_{t_k}\right] \leq 2^{4p}3^p\left(1+\pi_{2,2p}(f)\right)\left\|\tilde{X}_k\right\|_2^{2p}+2^{4p+1}3^p\left(\pi_{2,2p}(f)+\mathbb{E}\left[\chi(d)^{2p}\right]\right). \tag{27}$$

Thus, combining (26) and (27),

$$
\begin{aligned}
\mathbb{E}\left[\left\|\tilde{\nabla}f\right\|_2^{2p}|\mathcal{F}_{t_k}\right] &\leq \frac{1}{2}\mathbb{E}\left[\left\|\nabla f(\tilde{H}_1)\right\|_2^{2p}+\left\|\nabla f(\tilde{H}_2)\right\|_2^{2p}|\mathcal{F}_{t_k}\right]\\
&\leq \frac{1}{2}\pi_{2,2p}(f)\mathbb{E}\left[2+\left\|\tilde{H}_1\right\|_2^{2p}+\left\|\tilde{H}_2\right\|_2^{2p}|\mathcal{F}_{t_k}\right]\\
&\leq N_{8,n}(p)^2\left\|\tilde{X}_k\right\|_2^{2p}+N_{9,n}(p)^2,
\end{aligned}
$$

where the $p$-dependent constants are

$$N_{8,n}(p) = 2^{2p}3^{\frac{p}{2}}\left(\pi_{2,2p}(f)\left(1+\frac{1}{2}\pi_{2,2p}(f)\right)\right)^{\frac{1}{2}},$$

$$N_{9,n}(p) = \left(\pi_{2,2p}(f)\left(2^{4p+1}3^p\mathbb{E}\left[\chi(d)^{2p}\right]+2^{4p}3^p\pi_{2,2p}(f)+1\right)\right)^{\frac{1}{2}} = \mathcal{O}(d^{\frac{p}{2}}).$$

Since $N_{8,n}(p)$ does not depend on the dimension, let

$$N_{8,n} = \max\left\{N_{8,n}\left(2k_2+k_4+k_6\right) : k_1,\ldots,k_6\in\mathbb{N},\ k_1+\cdots+k_6=n,\ 2k_2+k_3+k_4+\tfrac{k_5}{2}+\tfrac{3k_6}{2}>1\right\}.$$

The bound on $B$ reduces to

$$\mathbb{E}\left[B|\mathcal{F}_{t_k}\right] \le \sum_{\substack{k_1+\cdots+k_6=n \\ 2k_2+k_3+k_4+\frac{k_5}{2}+\frac{3k_6}{2}>1}} 2^{\frac{3n}{2}}\binom{n}{k_1\ \ldots\ k_6}\left\|\tilde{X}_k\right\|_2^{2k_1+k_4+k_5}\mathbb{E}\left[\chi(d)^{4k_3+2k_5+2k_6}\right]^{1/2}$$

$$\times\left(N_{8,n}\left\|\tilde{X}_k\right\|_2^{2k_2+k_4+k_6}+N_{9,n}\left(2k_2+k_4+k_6\right)\right)$$

$$\le B_1+B_2,$$

where

$$B_1 = \sum_{\substack{k_1+\cdots+k_6=n \\ 2k_2+k_3+k_4+\frac{k_5}{2}+\frac{3k_6}{2}>1}} 2^{\frac{3n}{2}}\binom{n}{k_1\ \ldots\ k_6}\mathbb{E}\left[\chi(d)^{4k_3+2k_5+2k_6}\right]^{1/2}N_{8,n}\left\|\tilde{X}_k\right\|_2^{2k_1+2k_2+2k_4+k_5+k_6},$$

$$B_2 = \sum_{\substack{k_1+\cdots+k_6=n \\ 2k_2+k_3+k_4+\frac{k_5}{2}+\frac{3k_6}{2}>1}} 2^{\frac{3n}{2}}\binom{n}{k_1\ \ldots\ k_6}\mathbb{E}\left[\chi(d)^{4k_3+2k_5+2k_6}\right]^{1/2}N_{9,n}\left(2k_2+k_4+k_6\right)\left\|\tilde{X}_k\right\|_2^{2k_1+k_4+k_5}.$$

In the following, we bound the expectations of $B_1$ and $B_2$ separately. By Young's inequality for products and the function $x\mapsto x^{1/(2k_3+k_5+k_6)}$ being concave on the positive domain,

$$\mathbb{E}\left[\chi(d)^{4k_3+2k_5+2k_6}\right]^{1/2}N_{8,n}\left\|\tilde{X}_k\right\|_2^{2k_1+2k_2+2k_4+k_5+k_6}$$

$$\le N_{8,n}\left(\frac{2k_3+k_5+k_6}{2n}\mathbb{E}\left[\chi(d)^{4k_3+2k_5+2k_6}\right]^{\frac{2n}{4k_3+2k_5+2k_6}}+\frac{2k_1+2k_2+2k_4+k_5+k_6}{2n}\left\|\tilde{X}_k\right\|_2^{2n}\right)$$

$$\le N_{8,n}\left(\mathbb{E}\left[\chi(d)^2\right]^n+\left\|\tilde{X}_k\right\|_2^{2n}\right).$$

Hence,

$$\mathbb{E}\left[B_1|\mathcal{F}_{t_k}\right] \le \sum_{k_1+\cdots+k_6=n} 2^{\frac{3n}{2}}\binom{n}{k_1\ \ldots\ k_6}N_{8,n}\left(\mathbb{E}\left[\chi(d)^2\right]^n+\left\|\tilde{X}_k\right\|_2^{2n}\right)$$

$$= 2^{\frac{3n}{2}}6^n N_{8,n}\left(d^n+\left\|\tilde{X}_k\right\|_2^{2n}\right). \tag{28}$$

Similarly,

$$\mathbb{E}\left[\chi(d)^{4k_3+2k_5+2k_6}\right]^{\frac{1}{2}}N_{9,n}\left(2k_2+k_4+k_6\right)\left\|\tilde{X}_k\right\|_2^{2k_1+k_4+k_5}$$

$$\le\left(\mathbb{E}\left[\chi(d)^{4k_3+2k_5+2k_6}\right]^{\frac{1}{2}}N_{9,n}\left(2k_2+k_4+k_6\right)\right)^{\frac{2n}{2k_2+2k_3+k_4+k_5+2k_6}}+\left\|\tilde{X}_k\right\|_2^{2n}$$

$$\le N_{10,n}+\left\|\tilde{X}_k\right\|_2^{2n},$$

where

$$N_{10,n} = \max\left\{\left(\mathbb{E}\left[\chi(d)^{4k_3+2k_5+2k_6}\right]^{\frac{1}{2}}N_{9,n}\left(2k_2+k_4+k_6\right)\right)^{\frac{2n}{2k_2+2k_3+k_4+k_5+2k_6}}:\right.$$

$$\left. k_1,\ldots,k_6\in\mathbb{N},\ k_1+\cdots+k_6=n,\ 2k_2+k_3+k_4+\tfrac{k_5}{2}+\tfrac{3k_6}{2}>1\right\} = \mathcal{O}(d^n).$$

Hence,

$$\mathbb{E}\left[B_2|\mathcal{F}_{t_k}\right] \leq \sum_{k_1+\cdots+k_6=n} 2^{\frac{3n}{2}} \binom{n}{k_1 \ \ldots \ k_6} \left(N_{10,n} + \left\|\tilde{X}_k\right\|_2^{2n}\right)$$

$$\leq 2^{\frac{3n}{2}} 6^n \left(N_{10,n} + \left\|\tilde{X}_k\right\|_2^{2n}\right). \tag{29}$$

Therefore, combining (28) and (29),

$$\mathbb{E}\left[B\right] = \mathbb{E}\left[\mathbb{E}\left[B_1 + B_2|\mathcal{F}_{t_k}\right]\right] \leq N_{11,n}\mathbb{E}\left[\left\|\tilde{X}_k\right\|_2^{2n}\right] + N_{12,n}, \tag{30}$$

where

$$N_{11,n} = 2^{\frac{3n}{2}} 6^n \left(1 + N_{8,n}\right),$$
$$N_{12,n} = 2^{\frac{3n}{2}} 6^n \left(N_{8,n}d^n + N_{10,n}\right) = \mathcal{O}(d^n).$$

Thus, when $h < (3n\alpha/8N_{11,n})^2$, by (25) and (30),

$$\mathbb{E}\left[\left\|\tilde{X}_{k+1}\right\|^{2n}\right] \leq \left(1 - \frac{3}{4}\alpha n h + N_{11,n}h^{3/2}\right)\mathbb{E}\left[\left\|\tilde{X}_k\right\|_2^{2n}\right] + N_{7,n}h + N_{12,n}h^{3/2}$$

$$\leq \left(1 - \frac{3}{8}\alpha n h\right)\mathbb{E}\left[\left\|\tilde{X}_k\right\|_2^{2n}\right] + N_{7,n}h + N_{12,n}h^{3/2},$$

Hence,

$$\mathbb{E}\left[\left\|\tilde{X}_k\right\|_2^{2n}\right] \leq \mathbb{E}\left[\left\|\tilde{X}_0\right\|_2^{2n}\right] + \frac{8}{3\alpha n}\left(N_{7,n} + N_{12,n}\right).$$

$\square$

## B.2 Local Deviation Orders

We first provide two lemmas on bounding the second and fourth moments of the change in the continuous-time process. These will be used later when we verify the order conditions.

**Lemma 6.** *Suppose $X_t$ is the continuous-time process defined by (3) initiated from some iterate of the Markov chain $X_0$ defined by (9), then the second moment of $X_t$ is uniformly bounded by a constant of order $\mathcal{O}(d)$, i.e.*

$$\mathbb{E}\left[\|X_t\|_2^2\right] \leq \mathcal{U}_2', \quad \text{for all } t \geq 0,$$

*where $\mathcal{U}_2' = \mathcal{U}_2 + 2(\beta + d)/\alpha$.*

*Proof.* By Itô's lemma and dissipativity,

$$\frac{\mathrm{d}}{\mathrm{d}t}\mathbb{E}\left[\|X_t\|_2^2\right] = -2\mathbb{E}\left[\langle\nabla f(X_t), X_t\rangle\right] + 2d \leq -\alpha\mathbb{E}\left[\|X_t\|_2^2\right] + 2(\beta + d).$$

Moreover, by Grönwall's inequality,

$$\mathbb{E}\left[\|X_t\|_2^2\right] \leq e^{-\alpha t}\mathbb{E}\left[\|X_0\|_2^2\right] + 2(\beta + d)/\alpha \leq \mathcal{U}_2 + 2(\beta + d)/\alpha = \mathcal{U}_2'.$$

$\square$

**Lemma 7** (Second Moment of Change). *Suppose $X_t$ is the continuous-time process defined by (3) initiated from some iterate of the Markov chain $X_0$ defined by (9), then*

$$\mathbb{E}\left[\|X_t - X_0\|_2^2\right] \leq C_0 t = \mathcal{O}(dt), \quad \text{for all } 0 \leq t \leq 1,$$

*where $C_0 = 2\pi_{2,2}(f)\left(1 + \mathcal{U}_2'\right) + 4d$.*

*Proof.* By Young's inequality,

$$
\begin{aligned}
\mathbb{E}\left[\|X_t - X_0\|_2^2\right] = \mathbb{E}\left[\left\|-\int_0^t \nabla f(X_s)\,\mathrm{d}s + \sqrt{2}B_t\right\|_2^2\right] \\
\leq 2\mathbb{E}\left[\left\|\int_0^t \nabla f(X_s)\,\mathrm{d}s\right\|_2^2 + 2\|B_t\|_2^2\right] \\
\leq 2t\int_0^t \mathbb{E}\left[\|\nabla f(X_s)\|_2^2\right]\,\mathrm{d}s + 4\mathbb{E}\left[\|B_t\|_2^2\right] \\
\leq 2\pi_{2,2}(f)t\int_0^t \mathbb{E}\left[1 + \|X_s\|_2^2\right]\,\mathrm{d}s + 4dt \\
\leq 2\pi_{2,2}(f)\left(1 + \mathcal{U}_2'\right)t + 4dt.
\end{aligned}
$$

$\square$

**Lemma 8.** *Suppose $X_t$ is the continuous-time process defined by* (3) *initiated from some iterate of the Markov chain $X_0$ defined by* (9)*, then the fourth moment of $X_t$ is uniformly bounded by a constant of order $\mathcal{O}(d^2)$, i.e.*

$$
\mathbb{E}\left[\|X_t\|_2^4\right] \leq \mathcal{U}_4', \quad \text{for all } t \geq 0,
$$

*where $\mathcal{U}_4' = \mathcal{U}_4 + (2\beta + 6)\mathcal{U}_2'/\alpha$.*

*Proof.* By Itô's lemma, dissipativity, and Lemma 6,

$$
\begin{aligned}
\frac{\mathrm{d}}{\mathrm{d}t}\mathbb{E}\left[\|X_t\|_2^4\right] = &-4\mathbb{E}\left[\|X_t\|_2^2\langle \nabla f(X_t), X_t\rangle\right] + 12\mathbb{E}\left[\|X_t\|_2^2\right] \\
\leq &-2\alpha\mathbb{E}\left[\|X_t\|_2^4\right] + (4\beta + 12)\mathbb{E}\left[\|X_t\|_2^2\right] \\
\leq &-2\alpha\mathbb{E}\left[\|X_t\|_2^4\right] + (4\beta + 12)\mathcal{U}_2'.
\end{aligned}
$$

Moreover, by Grönwall's inequality,

$$
\begin{aligned}
\mathbb{E}\left[\|X_t\|_2^4\right] \leq &e^{-2\alpha t}\mathbb{E}\left[\|X_0\|_2^4\right] + (2\beta + 6)\mathcal{U}_2'/\alpha \\
\leq &\mathcal{U}_4 + (2\beta + 6)\mathcal{U}_2'/\alpha = \mathcal{U}_4'.
\end{aligned}
$$

$\square$

**Lemma 9** (Fourth Moment of Change). *Suppose $X_t$ is the continuous-time process defined by* (3) *initiated from some iterate of the Markov chain $X_0$ defined by* (9)*, then*

$$
\mathbb{E}\left[\|X_t - X_0\|_2^4\right] \leq C_1 t^2 = \mathcal{O}(d^2 t^2), \quad \text{for all } 0 \leq t \leq 1,
$$

*where $C_1 = 8\pi_{2,4}(f)\left(1 + \mathcal{U}_4'\right) + 32d(d + 2)$.*

*Proof.* By Young's inequality,

$$
\begin{aligned}
\mathbb{E}\left[\|X_t - X_0\|_2^4\right] =&\mathbb{E}\left[\left\|-\int_0^t \nabla f(X_s)\,\mathrm{d}s + \sqrt{2}B_t\right\|_2^4\right] \\
=&\mathbb{E}\left[\left(\left\|-\int_0^t \nabla f(X_s)\,\mathrm{d}s + \sqrt{2}B_t\right\|_2^2\right)^2\right] \\
\leq&\mathbb{E}\left[\left(2\left\|\int_0^t \nabla f(X_s)\,\mathrm{d}s\right\|_2^2 + 4\|B_t\|_2^2\right)^2\right]
\end{aligned}
$$

$$\leq \mathbb{E}\left[\left(2t\int_0^t \|\nabla f(X_s)\|_2^2 \, \mathrm{d}s + 4\|B_t\|_2^2\right)^2\right]$$

$$\leq \mathbb{E}\left[8t^2\left(\int_0^t \|\nabla f(X_s)\|_2^2 \, \mathrm{d}s\right)^2 + 32\|B_t\|_2^4\right]$$

$$\leq 8t^3\int_0^t \mathbb{E}\left[\|\nabla f(X_s)\|_2^4\right] \, \mathrm{d}s + 32\mathbb{E}\left[\|B_t\|_2^4\right]$$

$$\leq 8\pi_{2,4}(f)t^3\int_0^t \mathbb{E}\left[1 + \|X_s\|_2^4\right] \, \mathrm{d}s + 32d(d+2)t^2$$

$$\leq 8\pi_{2,4}(f)\left(1 + \mathcal{U}_4'\right)t^2 + 32d(d+2)t^2.$$

$\square$

### B.2.1 Local Mean-Square Deviation

**Lemma 10.** *Suppose $X_t$ and $\tilde{X}_t$ are the continuous-time process defined by* (3) *and Markov chain defined by* (9) *for time $t \geq 0$, respectively. If $X_t$ and $\tilde{X}_t$ are initiated from the same iterate of the Markov chain $X_0$ and share the same Brownian motion, then*

$$\mathbb{E}\left[\left\|X_t - \tilde{X}_t\right\|_2^2\right] \leq C_2 t^4 = \mathcal{O}(d^2 t^4), \quad \text{for all } 0 \leq t \leq 1,$$

*where*

$$C_2 = 8C_1^{1/2}(1 + \mathcal{U}_4')^{1/2}\left(\mu_2(f)^2\pi_{3,4}(f)^{1/2} + \mu_3(f)^2\pi_{2,4}(f)^{1/2}\right)$$
$$+ \left(8\pi_{2,4}(f)\left(1 + \mathcal{U}_4\right) + 116d^2 + 90d + 8C_0\right)\mu_3(f)^2.$$

*Proof.* Since the two processes share the same Brownian motion,

$$X_t - \tilde{X}_t = -\int_0^t \nabla f(X_s) \, \mathrm{d}s + \frac{t}{2}\left(\nabla f(\tilde{H}_1) + \nabla f(\tilde{H}_2)\right). \tag{31}$$

By Itô's lemma,

$$\nabla f(X_s) = \nabla f(X_0) - \int_0^s \left(\nabla^2 f(X_u)\nabla f(X_u) - \vec{\Delta}\left(\nabla f\right)(X_u)\right) \, \mathrm{d}u + \sqrt{2}\int_0^s \nabla^2 f(X_u) \, \mathrm{d}B_u$$

$$= \nabla f(X_0) - \nabla^2 f(X_0)\nabla f(X_0)s + \sqrt{2}\nabla^2 f(X_0)B_s + R(s),$$

where the remainder is

$$R(s) = \underbrace{\int_0^s \left(-\nabla^2 f(X_u)\nabla f(X_u) + \nabla^2 f(X_0)\nabla f(X_0)\right) \, \mathrm{d}u}_{R_1(s)} + \underbrace{\int_0^s \vec{\Delta}\left(\nabla f\right)(X_u) \, \mathrm{d}u}_{R_2(s)}$$

$$+ \underbrace{\sqrt{2}\int_0^s \left(\nabla^2 f(X_u) - \nabla^2 f(X_0)\right) \, \mathrm{d}B_u}_{R_3(s)}.$$

We bound the second moment of $R(s)$ by bounding those of $R_1(s)$, $R_2(s)$, and $R_3(s)$ separately. For $R_1(s)$, by the Cauchy–Schwarz inequality,

$$\mathbb{E}\left[\|R_1(s)\|_2^2\right] = \mathbb{E}\left[\left\|\int_0^s \left(\nabla^2 f(X_u)\nabla f(X_u) - \nabla^2 f(X_0)\nabla f(X_0)\right) \, \mathrm{d}u\right\|_2^2\right]$$

$$= 2\mathbb{E}\left[\left\|\int_0^s \left(\nabla^2 f(X_u)\nabla f(X_u) - \nabla^2 f(X_0)\nabla f(X_u)\right) \, \mathrm{d}u\right\|_2^2\right]$$

$$+ 2\mathbb{E}\left[\left\|\int_0^s \left(\nabla^2 f(X_0)\nabla f(X_u) - \nabla^2 f(X_0)\nabla f(X_0)\right) \, \mathrm{d}u\right\|_2^2\right]$$

$$\leq 2s \int_0^s \mathbb{E}\left[\left\|\nabla^2 f(X_u)\nabla f(X_u) - \nabla^2 f(X_0)\nabla f(X_u)\right\|_2^2\right] \mathrm{d}u$$

$$+ 2s \int_0^s \mathbb{E}\left[\left\|\nabla^2 f(X_0)\nabla f(X_u) - \nabla^2 f(X_0)\nabla f(X_0)\right\|_2^2\right] \mathrm{d}u$$

$$\leq 2s \int_0^s \mathbb{E}\left[\left\|\nabla^2 f(X_u) - \nabla^2 f(X_0)\right\|_{\mathrm{op}}^2 \left\|\nabla f(X_u)\right\|_2^2\right] \mathrm{d}u$$

$$+ 2s \int_0^s \mathbb{E}\left[\left\|\nabla^2 f(X_0)\right\|_{\mathrm{op}}^2 \left\|\nabla f(X_u) - \nabla f(X_0)\right\|_2^2\right] \mathrm{d}u$$

$$\leq 2\mu_3(f)^2 s \int_0^s \mathbb{E}\left[\left\|X_u - X_0\right\|_2^2 \left\|\nabla f(X_u)\right\|_2^2\right] \mathrm{d}u$$

$$+ 2\mu_2(f)^2 s \int_0^s \mathbb{E}\left[\left\|\nabla^2 f(X_0)\right\|_{\mathrm{op}}^2 \left\|X_u - X_0\right\|_2^2\right] \mathrm{d}u$$

$$\leq 2\mu_3(f)^2 s \int_0^s \mathbb{E}\left[\left\|X_u - X_0\right\|_2^4\right]^{1/2} \mathbb{E}\left[\left\|\nabla f(X_u)\right\|_2^4\right]^{1/2} \mathrm{d}u$$

$$+ 2\mu_2(f)^2 s \int_0^s \mathbb{E}\left[\left\|\nabla^2 f(X_0)\right\|_{\mathrm{op}}^4\right]^{1/2} \mathbb{E}\left[\left\|X_u - X_0\right\|_2^4\right]^{1/2} \mathrm{d}u$$

$$\leq 2\mu_3(f)^2 \pi_{2,4}(f)^{1/2} C_1^{1/2} \left(1 + \mathcal{U}_4'\right)^{1/2} \int_0^s u \, \mathrm{d}u$$

$$+ 2\mu_2(f)^2 \pi_{3,4}(f)^{1/2} C_1^{1/2} \left(1 + \mathcal{U}_4'\right)^{1/2} s \int_0^s u \, \mathrm{d}u$$

$$\leq C_1^{1/2} \left(1 + \mathcal{U}_4'\right)^{1/2} \left(\mu_2(f)^2 \pi_{3,4}(f)^{1/2} + \mu_3(f)^2 \pi_{2,4}(f)^{1/2}\right) s^3. \qquad (32)$$

For $R_2(s)$, by Lemma 34,

$$\mathbb{E}\left[\left\|R_2(s)\right\|_2^2\right] = \mathbb{E}\left[\left\|\int_0^s \vec{\Delta}\left(\nabla f\right)(X_u) \, \mathrm{d}u\right\|_2^2\right]$$

$$\leq s \int_0^s \mathbb{E}\left[\left\|\vec{\Delta}\left(\nabla f\right)(X_u)\right\|_2^2\right] \mathrm{d}u$$

$$\leq \mu_3(f)^2 d^2 s^2. \qquad (33)$$

For $R_3(s)$, by Itô isometry,

$$\mathbb{E}\left[\left\|R_3(s)\right\|_2^2\right] = 2\mathbb{E}\left[\left\|\int_0^s \left(\nabla^2 f(X_u) - \nabla^2 f(X_0)\right) \mathrm{d}B_u\right\|_2^2\right]$$

$$= 2\mathbb{E}\left[\int_0^s \left\|\nabla^2 f(X_u) - \nabla^2 f(X_0)\right\|_2^2 \mathrm{d}u\right]$$

$$\leq 2\mu_3(f)^2 \int_0^s \mathbb{E}\left[\left\|X_u - X_0\right\|_2^2\right] \mathrm{d}u$$

$$\leq 2\mu_3(f)^2 C_0 \int_0^s u \, \mathrm{d}u$$

$$\leq \mu_3(f)^2 C_0 s^2. \qquad (34)$$

Thus, combining (32), (33), and (34),

$$\mathbb{E}\left[\left\|R(s)\right\|_2^2\right] \leq 4\mathbb{E}\left[\left\|R_1(s)\right\|_2^2\right] + 4\mathbb{E}\left[\left\|R_2(s)\right\|_2^2\right] + 4\mathbb{E}\left[\left\|R_3(s)\right\|_2^2\right]$$

$$\leq 4C_1^{1/2}(1 + \mathcal{U}_4')^{1/2} \left(\mu_2(f)^2 \pi_{3,4}(f)^{1/2} + \mu_3(f)^2 \pi_{2,4}(f)^{1/2}\right) s^2$$

$$+ 4\mu_3(f)^2 \left(d^2 + C_0\right) s^2.$$

Next, we characterize the terms in the Markov chain update. By Taylor's theorem,

$$\nabla f(\tilde{H}_1) = \nabla f(X_0) + \nabla^2 f(X_0)\Delta\tilde{H}_1 + \rho_1(t),$$

$$\nabla f(\tilde{H}_2) = \nabla f(X_0) + \nabla^2 f(X_0)\Delta\tilde{H}_2 + \rho_2(t),$$

where

$$\rho_1(t) = \int_0^1 (1-\tau)\nabla^3 f(X_0 + \tau\Delta\tilde{H}_1)[\Delta\tilde{H}_1, \Delta\tilde{H}_1]\, \mathrm{d}\tau,$$

$$\rho_2(t) = \int_0^1 (1-\tau)\nabla^3 f(X_0 + \tau\Delta\tilde{H}_2)[\Delta\tilde{H}_2, \Delta\tilde{H}_2]\, \mathrm{d}\tau,$$

$$\Delta\tilde{H}_1 = \sqrt{2}\left(\frac{1}{t}\Psi(t) + \frac{1}{\sqrt{6}}B_t\right),$$

$$\Delta\tilde{H}_2 = -\nabla f(X_0)t + \sqrt{2}\left(\frac{1}{t}\Psi(t) - \frac{1}{\sqrt{6}}B_t\right),$$

$$\Psi(t) = \int_0^t B_s\, \mathrm{d}s.$$

We bound the fourth moments of $\Delta\tilde{H}_1$ and $\Delta\tilde{H}_2$,

$$
\begin{aligned}
\mathbb{E}\left[\left\|\Delta\tilde{H}_1\right\|_2^4\right] =&\mathbb{E}\left[\left\|\sqrt{2}\left(\frac{1}{t}\Psi(t) + \frac{1}{\sqrt{6}}B_t\right)\right\|_2^4\right]\\
\leq&\frac{32}{t^4}\mathbb{E}\left[\|\Psi(t)\|_2^4\right] + \frac{8}{9}\mathbb{E}\left[\|B_t\|_2^4\right]\\
=&\frac{32}{t^4}\sum_{i=1}^d \mathbb{E}\left[\Psi_i(t)^4\right] + \frac{32}{t^4}\sum_{i,j=1, i\neq j}^d \mathbb{E}\left[\Psi_i(t)^2\right]\mathbb{E}\left[\Psi_j(t)^2\right] + \frac{8}{9}d(d+2)t^2\\
\leq&\frac{32}{t^4}\frac{dt^6}{3} + \frac{32}{t^4}\frac{d(d-1)t^6}{9} + \frac{8d(d+2)t^2}{9}\\
=&\left(\frac{32d}{3} + \frac{32d(d-1)}{9} + \frac{8d(d+2)}{9}\right)t^2\\
\leq&2d(6d+5)t^2.
\end{aligned}
$$

Similarly,

$$
\begin{aligned}
\mathbb{E}\left[\left\|\Delta\tilde{H}_2\right\|_2^4\right] =&\mathbb{E}\left[\left\|-\nabla f(X_0)t + \sqrt{2}\left(\frac{1}{t}\Psi(t) - \frac{1}{\sqrt{6}}B_t\right)\right\|_2^4\right]\\
\leq&8\mathbb{E}\left[\|\nabla f(X_0)\|_2^4\right]t^4 + 8\mathbb{E}\left[\left\|\sqrt{2}\left(\frac{1}{t}\Psi(t) - \frac{1}{\sqrt{6}}B_t\right)\right\|_2^4\right]\\
\leq&8\pi_{2,4}(f)\mathbb{E}\left[1 + \|X_0\|_2^4\right]t^4 + 16d(6d+5)t^2\\
\leq&8\pi_{2,4}(f)\left(1 + \mathcal{U}_4\right)t^4 + 16d(6d+5)t^2\\
\leq&8\left(\pi_{2,4}(f)\left(1 + \mathcal{U}_4\right) + 2d(6d+5)\right)t^2.
\end{aligned}
$$

Using the above information, we bound the second moments of $\rho_1(t)$ and $\rho_2(t)$,

$$
\begin{aligned}
\mathbb{E}\left[\|\rho_1(t)\|_2^2\right] =&\mathbb{E}\left[\left\|\int_0^1 (1-\tau)\nabla^3 f(X_0 + \tau\Delta\tilde{H}_1)[\Delta\tilde{H}_1, \Delta\tilde{H}_1]\, \mathrm{d}\tau\right\|_2^2\right]\\
\leq&\int_0^1 \mathbb{E}\left[\left\|\nabla^3 f(X_0 + \tau\Delta\tilde{H}_1)[\Delta\tilde{H}_1, \Delta\tilde{H}_1]\right\|_2^2\right]\mathrm{d}\tau\\
\leq&\int_0^1 \mathbb{E}\left[\left\|\nabla^3 f(X_0 + \tau\Delta\tilde{H}_1)\right\|_{\mathrm{op}}^2\left\|\Delta\tilde{H}_1\right\|_2^4\right]\mathrm{d}\tau\\
\leq&\mu_3(f)^2\int_0^1 \mathbb{E}\left[\left\|\Delta\tilde{H}_1\right\|_2^4\right]\mathrm{d}\tau\\
\leq&2d(6d+5)\mu_3(f)^2t^2.
\end{aligned}
$$

Similarly,

$$
\begin{aligned}
\mathbb{E}\left[\|\rho_2(t)\|_2^2\right] \leq & \mu_3(f)^2 \int_0^1 \mathbb{E}\left[\left\|\Delta\tilde{H}_2\right\|_2^4\right] \mathrm{d}\tau \\
& \leq 8\left(\pi_{2,4}(f)\left(1+\mathcal{U}_4\right)+2d(6d+5)\right)\mu_3(f)^2 t^2.
\end{aligned}
$$

Plugging these results into (31),

$$
X_t - \tilde{X}_t = -\int_0^t R(s)\ \mathrm{d}s - \frac{t}{2}\left(\rho_1(t)+\rho_2(t)\right).
$$

Thus,

$$
\begin{aligned}
\mathbb{E}\left[\left\|X_t-\tilde{X}_t\right\|_2^2\right] = & \mathbb{E}\left[\left\|-\int_0^t R(s)\ \mathrm{d}s - \frac{t}{2}\left(\rho_1(t)+\rho_2(t)\right)\right\|_2^2\right] \\
\leq & 4t\int_0^t \mathbb{E}\left[\|R(s)\|_2^2\right]\ \mathrm{d}s + t^2\mathbb{E}\left[\|\rho_1(t)\|_2^2\right]+t^2\mathbb{E}\left[\|\rho_2(t)\|_2^2\right] \\
\leq & 8C_1^{1/2}(1+\mathcal{U}_4')^{1/2}\left(\mu_2(f)^2\pi_{3,4}(f)^{1/2}+\mu_3(f)^2\pi_{2,4}(f)^{1/2}\right)t^4 \\
& + \left(8\pi_{2,4}(f)\left(1+\mathcal{U}_4\right)+116d^2+90d+8C_0\right)\mu_3(f)^2 t^4 \\
\leq & C_2 t^4.
\end{aligned}
$$

$\square$

### B.2.2 Local Mean Deviation

**Lemma 11.** *Suppose $X_t$ and $\tilde{X}_t$ are the continuous-time process defined by* (3) *and Markov chain defined by* (9) *for time $t \geq 0$, respectively. If $X_t$ and $\tilde{X}_t$ are initiated from the same iterate of the Markov chain $X_0$ and share the same Brownian motion, then*

$$
\mathbb{E}\left[\left\|\mathbb{E}\left[X_t-\tilde{X}_t|\mathcal{F}_0\right]\right\|_2^2\right] \leq C_3 t^5 = \mathcal{O}(d^3 t^5), \quad \text{for all } 0 \leq t \leq 1,
$$

*where*

$$
\begin{aligned}
C_3 = & 4\left(C_1^{1/2}\left(1+\mathcal{U}_4'\right)^{1/2}\left(\mu_2(f)^2\pi_{3,4}(f)^{1/2}+\mu_3(f)^2\pi_{2,4}(f)^{1/2}\right)+C_0 d\mu_4(f)^2\right) \\
& + \frac{1}{4}\mu_3(f)^2\pi_{2,4}(f)\left(1+\mathcal{U}_4\right)+8\mu_4(f)^2\left(\pi_{2,6}(f)\left(1+\mathcal{U}_6\right)+73(d+4)^3\right).
\end{aligned}
$$

*Proof.* The proof is similiar to that of Lemma 10 with slight variations on truncating the expansions. Recall since the two processes share the same Brownian motion,

$$
X_t - \tilde{X}_t = -\int_0^t \nabla f(X_s)\ \mathrm{d}s + \frac{t}{2}\left(\nabla f(\tilde{H}_1)+\nabla f(\tilde{H}_2)\right).
$$

By Itô's lemma,

$$
\begin{aligned}
\nabla f(X_s) = & \nabla f(X_0)-\int_0^s\left(\nabla^2 f(X_u)\nabla f(X_u)-\vec{\Delta}\left(\nabla f\right)(X_u)\right)\ \mathrm{d}u + \sqrt{2}\int_0^s \nabla^2 f(X_u)\ \mathrm{d}B_u \\
= & \nabla f(X_0)-\nabla^2 f(X_0)\nabla f(X_0)s+\sqrt{2}\nabla^2 f(X_0)B_s+\vec{\Delta}(\nabla f)(X_0)s+\bar{R}(s),
\end{aligned}
$$

where the remainder is

$$
\begin{aligned}
\bar{R}(s) = & \underbrace{\int_0^s\left(-\nabla^2 f(X_u)\nabla f(X_u)+\nabla^2 f(X_0)\nabla f(X_0)\right)\ \mathrm{d}u}_{\bar{R}_1(s)} \\
& + \underbrace{\int_0^s\left(\vec{\Delta}\left(\nabla f\right)(X_u)-\vec{\Delta}\left(\nabla f\right)(X_0)\right)\ \mathrm{d}u}_{\bar{R}_2(s)}
\end{aligned}
$$

$$+ \underbrace{\sqrt{2}\int_0^s \left(\nabla^2 f(X_u) - \nabla^2 f(X_0)\right)\,\mathrm{d}B_u}_{\bar{R}_3(s)}.$$

By Taylor's theorem with the remainder in integral form,

$$\nabla f(\tilde{H}_1) = \nabla f(X_0) + \nabla^2 f(X_0)\Delta\tilde{H}_1 + \frac{1}{2}\nabla^3 f(X_0)[\Delta\tilde{H}_1,\,\Delta\tilde{H}_1] + \bar{\rho}_1(t),$$

$$\nabla f(\tilde{H}_2) = \nabla f(X_0) + \nabla^2 f(X_0)\Delta\tilde{H}_2 + \frac{1}{2}\nabla^3 f(X_0)[\Delta\tilde{H}_2,\,\Delta\tilde{H}_2] + \bar{\rho}_2(t),$$

where

$$\bar{\rho}_1(t) = \frac{1}{2}\int_0^1 (1-\tau)^2 \nabla^4 f(X_0 + \tau\Delta\tilde{H}_1)[\Delta\tilde{H}_1,\,\Delta\tilde{H}_1,\,\Delta\tilde{H}_1]\,\mathrm{d}\tau,$$

$$\bar{\rho}_2(t) = \frac{1}{2}\int_0^1 (1-\tau)^2 \nabla^4 f(X_0 + \tau\Delta\tilde{H}_2)[\Delta\tilde{H}_2,\,\Delta\tilde{H}_2,\,\Delta\tilde{H}_2]\,\mathrm{d}\tau.$$

Now, we show the following equality in a component-wise manner,

$$\frac{t^2}{2}\mathbb{E}\left[\vec{\Delta}\left(\nabla f\right)(X_0)\right] + \frac{t^3}{4}\mathbb{E}\left[\nabla^3 f(X_0)[\nabla f(X_0),\,\nabla f(X_0)]\right] = $$
$$\frac{t}{4}\mathbb{E}\left[\nabla^3 f(X_0)[\Delta\tilde{H}_1,\,\Delta\tilde{H}_1]\right] + \frac{t}{4}\mathbb{E}\left[\nabla^3 f(X_0)[\Delta\tilde{H}_2,\,\Delta\tilde{H}_2]\right]. \quad (35)$$

To see this, recall that odd moments of the Brownian motion is zero. So, for each $\partial_i f$,

$$\mathbb{E}\left[\left\langle\Delta\tilde{H}_1, \nabla^2(\partial_i f)(X_0)\Delta\tilde{H}_1\right\rangle\right] = \mathbb{E}\left[\mathbb{E}\left[\mathrm{Tr}\left((\Delta\tilde{H}_1)^\top\Delta\tilde{H}_1\nabla^2(\partial_i f)(X_0)\right)\Big|\mathcal{F}_0\right]\right]$$
$$= \mathbb{E}\left[\mathrm{Tr}\left(\mathbb{E}\left[(\Delta\tilde{H}_1)^\top\Delta\tilde{H}_1|\mathcal{F}_0\right]\nabla^2(\partial_i f)(X_0)\right)\right]$$
$$= 2t\left(\frac{1}{2} + \frac{1}{\sqrt{6}}\right)\mathbb{E}\left[\Delta(\partial_i f)(X_0)\right].$$

Similarly,

$$\mathbb{E}\left[\left\langle\Delta\tilde{H}_2, \nabla^2(\partial_i f)(X_0)\Delta\tilde{H}_2\right\rangle\right] = \mathbb{E}\left[\mathbb{E}\left[\mathrm{Tr}\left((\Delta\tilde{H}_2)^\top\Delta\tilde{H}_2\nabla^2(\partial_i f)(X_0)\right)\Big|\mathcal{F}_0\right]\right]$$
$$= \mathbb{E}\left[\mathrm{Tr}\left(\mathbb{E}\left[(\Delta\tilde{H}_2)^\top\Delta\tilde{H}_2|\mathcal{F}_0\right]\nabla^2\partial_i f(X_0)\right)\right]$$
$$= 2t\left(\frac{1}{2} - \frac{1}{\sqrt{6}}\right)\mathbb{E}\left[\Delta(\partial_i f)(X_0)\right]$$
$$+ t^2\mathbb{E}\left[\left\langle\nabla f(X_0), \nabla^2(\partial_i f)(X_0)\nabla f(X_0)\right\rangle\right].$$

Adding the previous two equations together, we obtain the desired equality (35).

Next, we bound the second moments of $\bar{R}_1(s)$ and $\bar{R}_2(s)$. For $\bar{R}_1(s)$, recall from the proof of Lemma 10,

$$\mathbb{E}\left[\left\|\bar{R}_1(s)\right\|_2^2\right] = \mathbb{E}\left[\left\|R_1(s)\right\|_2^2\right] \leq C_1^{1/2}(1 + \mathcal{U}_4')^{1/2}\left(\mu_2(f)^2\pi_{3,4}(f)^{1/2} + \mu_3(f)^2\pi_{2,4}(f)^{1/2}\right)s^3.$$

Additionally for $\bar{R}_2(s)$,

$$\mathbb{E}\left[\left\|\bar{R}_2(s)\right\|_2^2\right] = \mathbb{E}\left[\left\|\int_0^s \left(\vec{\Delta}\left(\nabla f\right)(X_u) - \vec{\Delta}\left(\nabla f\right)(X_0)\right)\,\mathrm{d}u\right\|_2^2\right]$$
$$\leq s\int_0^s \mathbb{E}\left[\left\|\vec{\Delta}\left(\nabla f\right)(X_u) - \vec{\Delta}\left(\nabla f\right)(X_0)\right\|_2^2\right]\,\mathrm{d}u$$
$$\leq d^2\mu_4(f)^2 s\int_0^s \mathbb{E}\left[\left\|X_u - X_0\right\|_2^2\right]\,\mathrm{d}u$$
$$\leq C_0 d^2\mu_4(f)^2 s\int_0^s u\,\mathrm{d}u$$

$$\leq C_0 d^2 \mu_4(f)^2 \frac{s^3}{2}.$$

Since $\bar{R}_3(s)$ is a Martingale,

$$\left\| \mathbb{E}\left[ \int_0^t \bar{R}(s) \, ds | \mathcal{F}_0 \right] \right\|_2^2 = \left\| \mathbb{E}\left[ \int_0^t \bar{R}_1(s) \, ds | \mathcal{F}_0 \right] + \mathbb{E}\left[ \int_0^t \bar{R}_2(s) \, ds | \mathcal{F}_0 \right] \right\|_2^2$$

$$\leq 2 \left\| \mathbb{E}\left[ \int_0^t \bar{R}_1(s) \, ds | \mathcal{F}_0 \right] \right\|_2^2 + 2 \left\| \mathbb{E}\left[ \int_0^t \bar{R}_2(s) \, ds | \mathcal{F}_0 \right] \right\|_2^2$$

$$\leq 2t \int_0^t \mathbb{E}\left[ \left\| \bar{R}_1(s) \right\|_2^2 + \left\| \bar{R}_2(s) \right\|_2^2 | \mathcal{F}_0 \right] \, ds.$$

Therefore,

$$\mathbb{E}\left[ \left\| \mathbb{E}\left[ \int_0^t \bar{R}(s) \, ds | \mathcal{F}_0 \right] \right\|_2^2 \right] \leq 2t \int_0^t \mathbb{E}\left[ \left\| \bar{R}_1(s) \right\|_2^2 + \left\| \bar{R}_2(s) \right\|_2^2 \right] \, ds$$

$$\leq C_1^{1/2} \left( 1 + \mathcal{U}_4' \right)^{1/2} \left( \mu_2(f)^2 \pi_{3,4}(f)^{1/2} + \mu_3(f)^2 \pi_{2,4}(f)^{1/2} \right) t^5$$

$$+ C_0 d \mu_4(f)^2 t^5.$$

Next, we bound the sixth moments of $\Delta \tilde{H}_1$ and $\Delta \tilde{H}_2$. Note for two random vectors $a$ and $b$, by Young inequality and Lemma 31, we have

$$\mathbb{E}\left[ \| a + b \|_2^6 \right] \leq \mathbb{E}\left[ \left( 2\| a \|_2^2 + 2\| b \|_2^2 \right)^3 \right] \leq 32 \mathbb{E}\left[ \| a \|_2^6 + \| b \|_2^6 \right].$$

To simplify notation, we define

$$v_1 = \sqrt{2} \left( \frac{1}{2} + \frac{1}{\sqrt{6}} \right) \xi \sqrt{t}, \quad v_1' = \sqrt{2} \left( \frac{1}{2} - \frac{1}{\sqrt{6}} \right) \xi \sqrt{t},$$

$$v_2 = \frac{1}{\sqrt{6}} \eta \sqrt{t} \quad \text{where} \quad \xi, \eta \overset{\text{i.i.d.}}{\sim} \mathcal{N}(0, I_d),$$

We bound the sixth moments of $v_1$, $v_1'$ and $v_2$ using $1/2 + 1/\sqrt{6} < 1$, $1/2 - 1/\sqrt{6} < 1/2$ and the closed form moments of a chi-squared random variable with $d$ degrees of freedom $\chi(d)^2$ [57],

$$\mathbb{E}\left[ \| v_1 \|_2^6 \right] \leq 8 \mathbb{E}\left[ \| \xi \|_2^6 \right] t^3 = 8 \mathbb{E}\left[ \chi(d)^6 \right] t^3 = 8d(d+2)(d+4)t^3 < 8(d+4)^3 t^3,$$

$$\mathbb{E}\left[ \| v_1' \|_2^6 \right] \leq \mathbb{E}\left[ \| \xi \|_2^6 \right] t^3 = \mathbb{E}\left[ \chi(d)^6 \right] t^3 = d(d+2)(d+4)t^3 < (d+4)^3 t^3,$$

$$\mathbb{E}\left[ \| v_2 \|_2^6 \right] = \frac{1}{216} \mathbb{E}\left[ \| \eta \|_2^6 \right] t^3 = \frac{1}{216} \mathbb{E}\left[ \chi(d)^6 \right] t^3 = \frac{1}{216} d(d+2)(d+4)t^3 < \frac{1}{216}(d+4)^3 t^3.$$

Then,

$$\mathbb{E}\left[ \left\| \Delta \tilde{H}_1 \right\|_2^6 \right] = \mathbb{E}\left[ \| v_1 + v_2 \|_2^6 \right] \leq 32 \mathbb{E}\left[ \| v_1 \|_2^6 + \| v_2 \|_2^6 \right] \leq 288(d+4)^3 t^3,$$

$$\mathbb{E}\left[ \left\| \Delta \tilde{H}_2 \right\|_2^6 \right] = \mathbb{E}\left[ \| -\nabla f(X_0)t + v_1' + v_2 \|_2^6 \right]$$

$$\leq 32 \mathbb{E}\left[ \| \nabla f(X_0)t \|_2^6 \right] t^6 + 32 \mathbb{E}\left[ \| v_1' + v_2 \|_2^6 \right]$$

$$\leq 32 \pi_{2,6}(f) \left( 1 + \mathbb{E}\left[ \| X_0 \|_2^6 \right] \right) t^6 + 1024 \mathbb{E}\left[ \| v_1' \|_2^6 + \| v_2 \|_2^6 \right]$$

$$\leq 32 \pi_{2,6}(f) \left( 1 + \mathcal{U}_6 \right) t^3 + 2048(d+4)^3 t^3$$

$$\leq 32 \left( \pi_{2,6}(f) \left( 1 + \mathcal{U}_6 \right) + 64(d+4)^3 \right) t^3.$$

Now, we bound the second moments of $\bar{\rho}_1(t)$ and $\bar{\rho}_2(t)$ using the derived sixth-moment bounds,

$$\mathbb{E}\left[ \| \bar{\rho}_1(t) \|_2^2 \right] = \mathbb{E}\left[ \left\| \frac{1}{2} \int_0^1 (1-\tau)^2 \nabla^4 f(X_0 + \tau \Delta \tilde{H}_1)[\Delta \tilde{H}_1, \, \Delta \tilde{H}_1, \, \Delta \tilde{H}_1] \right\|_2^2 \right]$$

$$\leq \frac{1}{4} \sup_{z \in \mathbb{R}^d} \left\| \nabla^4 f(z) \right\|_{\mathrm{op}}^2 \mathbb{E}\left[ \left\| \Delta \tilde{H}_1 \right\|_2^6 \right]$$

$$\leq 72 \mu_4(f)^2 (d+4)^3 t^3.$$

Similarly,

$$\mathbb{E}\left[ \left\| \bar{\rho}_2(t) \right\|_2^2 \right] = \mathbb{E}\left[ \left\| \frac{1}{2} \int_0^1 (1-\tau)^2 \nabla^4 f(X_0 + \tau \Delta \tilde{H}_2)[\Delta \tilde{H}_2, \, \Delta \tilde{H}_2, \, \Delta \tilde{H}_2] \right\|_2^2 \right]$$

$$\leq \frac{1}{4} \sup_{z \in \mathbb{R}^d} \left\| \nabla^4 f(z) \right\|_{\mathrm{op}}^2 \mathbb{E}\left[ \left\| \Delta \tilde{H}_2 \right\|_2^6 \right]$$

$$\leq 8 \mu_4(f)^2 \left( \pi_{2,6}(f) \left(1 + \mathcal{U}_6\right) + 64(d+4)^3 \right) t^3.$$

Thus,

$$\mathbb{E}\left[ \left\| \mathbb{E}\left[ X_t - \tilde{X}_t | \mathcal{F}_0 \right] \right\|_2^2 \right]$$

$$= \mathbb{E}\left[ \left\| \mathbb{E}\left[ -\int_0^t \bar{R}(s) \, \mathrm{d}s + \frac{t^3}{4} \nabla^3 f(X_0)[\nabla f(X_0), \, \nabla f(X_0)] + \frac{t}{2} \bar{\rho}_1(t) + \frac{t}{2} \bar{\rho}_2(t) | \mathcal{F}_0 \right] \right\|^2 \right]$$

$$\leq 4 \mathbb{E}\left[ \left\| \mathbb{E}\left[ \int_0^t \bar{R}(s) \, \mathrm{d}s | \mathcal{F}_0 \right] \right\|_2^2 \right] + \frac{t^6}{4} \mathbb{E}\left[ \left\| \nabla^3 f(X_0)[\nabla f(X_0), \, \nabla f(X_0)] \right\|_2^2 \right]$$

$$+ t^2 \mathbb{E}\left[ \left\| \bar{\rho}_1(t) \right\|_2^2 + \left\| \bar{\rho}_2(t) \right\|_2^2 \right]$$

$$\leq 4 \left( C_1^{1/2} \left(1 + \mathcal{U}_4'\right)^{1/2} \left( \mu_2(f)^2 \pi_{3,4}(f)^{1/2} + \mu_3(f)^2 \pi_{2,4}(f)^{1/2} \right) + C_0 d \mu_4(f)^2 \right) t^5$$

$$+ \frac{1}{4} \mu_3(f)^2 \mathbb{E}\left[ \left\| \nabla f(X_0) \right\|_2^4 \right] t^6$$

$$+ 72 \mu_4(f)^2 (d+4)^3 t^5 + 8 \mu_4(f)^2 \left( \pi_{2,6} \left(1 + \mathcal{U}_6\right) + 64(d+4)^3 \right) t^5$$

$$\leq 4 \left( C_1^{1/2} \left(1 + \mathcal{U}_4'\right)^{1/2} \left( \mu_2(f)^2 \pi_{3,4}(f)^{1/2} + \mu_3(f)^2 \pi_{2,4}(f)^{1/2} \right) + C_0 d \mu_4(f)^2 \right) t^5$$

$$+ \frac{1}{4} \mu_3(f)^2 \pi_{2,4}(f) \left(1 + \mathcal{U}_4\right) t^5$$

$$+ 8 \mu_4(f)^2 \left( \pi_{2,6}(f) \left(1 + \mathcal{U}_6\right) + 73(d+4)^3 \right) t^5$$

$$\leq C_3 t^5.$$

$\square$

### B.3 Invoking Theorem 1

Now, we invoke Theorem 1 with our derived constants. We obtain that if the constant step size

$$h < 1 \wedge C_h \wedge \frac{1}{2\alpha} \wedge \frac{1}{8\mu_1(b)^2 + 8\mu_1^{\mathrm{F}}(\sigma)^2},$$

where

$$C_h = \frac{2d}{\pi_{2,2}(f)} \wedge \frac{2\pi_{2,1}(f)}{\pi_{2,2}(f)} \wedge \frac{\alpha}{4\mu_2(f)\pi_{2,2}(f)} \wedge \frac{3\alpha}{2N_1 + 2N_2 + 4} \wedge \min\left\{ \left( \frac{3\alpha l}{8N_{11,l}} \right)^2 : l = 2, 3 \right\},$$

and the smoothness conditions on the strongly convex potential in Theorem 2 holds, then the uniform local deviation bounds (7) hold with $\lambda_1 = C_2$ and $\lambda_2 = C_3$, and consequently the bound (8) holds. This concludes that to converge to a sufficiently small positive tolerance $\epsilon$, $\tilde{\mathcal{O}}(d\epsilon^{-2/3})$ iterations are required, since $C_2$ is of order $\mathcal{O}(d^2)$, and $C_3$ is of order $\mathcal{O}(d^3)$.

## C  Proof of Theorem 3

### C.1  Moment Bounds

Verifying the order conditions in Theorem 1 for SRK-ID requires bounding the second and fourth moments of the Markov chain.

The following proofs only assume Lipschitz smoothness of the drift coefficient $b$ and diffusion coefficient $\sigma$ to a certain order and a generalized notion of dissipativity for Itô diffusions.

**Definition C.1** (Dissipativity). *For constants $\alpha, \beta > 0$, the diffusion satisfies the following*

$$-2 \langle b(x), x \rangle - \|\sigma(x)\|_{\mathrm{F}}^2 \geq \alpha \|x\|_2^2 - \beta, \quad \text{for all } x \in \mathbb{R}^d.$$

For general Itô diffusions, dissipativity directly follows from uniform dissipativity, where $\beta$ is an appropriate constant of order $\mathcal{O}(d)$. Additionally, we assume the discretization has a constant step size $h$ and the timestamp of the $k$th iterate is $t_k$ as per the proof of Theorem 1. To simplify notation, we rewrite the update as

$$\tilde{X}_{k+1} = \tilde{X}_k + b(\tilde{X}_k)h + \sigma(\tilde{X}_k)\xi_{k+1}h^{1/2} + \tilde{Y}_{k+1}, \quad \xi_{k+1} \sim \mathcal{N}(0, I_d),$$

where

$$\tilde{Y}_{k+1}^{(i)} = \left(\sigma_i(\tilde{H}_1^{(i)}) - \sigma_i(\tilde{H}_2^{(i)})\right) h^{1/2}, \quad \tilde{Y}_{k+1} = \frac{1}{2} \sum_{i=1}^{m} \tilde{Y}_{k+1}^{(i)}.$$

Note that $\xi_{k+1}$ and $\tilde{Y}_{k+1}$ are not independent, since we model $I_{(\cdot)} = (I_{(1)}, \ldots, I_{(m)})^\top$ as $\xi_{k+1}h^{1/2}$. Moreover, we define the following notation

$$I_{(\cdot, i)} = (I_{(1,i)}, \ldots, I_{(m,i)})^\top, \quad \Delta \tilde{H}^{(i)} = \sigma(\tilde{X}_k)I_{(\cdot, i)}h^{-1/2}, \quad i = 1, \ldots, m.$$

Hence, the variables $\tilde{H}_1^{(i)}$ and $\tilde{H}_2^{(i)}$ can be written as

$$\tilde{H}_1^{(i)} = \tilde{X}_k + \Delta \tilde{H}^{(i)}, \quad \tilde{H}_2^{(i)} = \tilde{X}_k - \Delta \tilde{H}^{(i)}.$$

We first bound the second moments of $\tilde{Y}_k$, using the following moment inequality.

**Theorem 12** ([41, Sec. 1.7, Thm. 7.1]). *Let $p \geq 2$. If $\{G_s\}_{s \geq 0}$ is a $d \times m$ matrix-valued process, and $\{B_t\}_{t \geq 0}$ is a $d$-dimensional Brownian motion, both of which are adapted to the filtration $\{\mathcal{F}_s\}_{s \geq 0}$ such that for some fixed $t > 0$, the following relation holds*

$$\mathbb{E}\left[\int_0^t \|G_s\|_{\mathrm{F}}^p \, \mathrm{d}s\right] < \infty.$$

*Then,*

$$\mathbb{E}\left[\left\|\int_0^t G_s \, \mathrm{d}B_s\right\|_2^p\right] \leq \left(\frac{p(p-1)}{2}\right)^{p/2} t^{(p-2)/2} \mathbb{E}\left[\int_0^t \|G_s\|_{\mathrm{F}}^p \, \mathrm{d}s\right].$$

*In particular, equality holds when $p = 2$.*

The above theorem can be proved directly using Itô's lemma and Itô isometry, with the help of Hölder's inequality. The theorem can also be seen as a natural consequence of the Burkholder-Davis-Gundy Inequality [41].

**Corollary 13.** *Let even integer $p \geq 2$. Then, the following relation holds*

$$\mathbb{E}\left[\left\|\Delta \tilde{H}^{(i)}\right\|_2^p \middle| \mathcal{F}_{t_k}\right] \leq \left(\frac{p(p-1)}{2}\right)^p \pi_{1,p}^{\mathrm{F}}(\sigma) \left(1 + \left\|\tilde{X}_k\right\|_2^{p/2}\right) h^{p/2}.$$

*Proof.* It is clear that the integrability condition in Theorem 12 holds for the inner and outer integrals of $\Delta \tilde{H}^{(i)}$. Hence, by repeatedly applying the theorem,

$$\begin{aligned}
\mathbb{E}\left[\left\|\Delta \tilde{H}^{(i)}\right\|_2^p \middle| \mathcal{F}_{t_k}\right] &= \mathbb{E}\left[\left\|\sigma(\tilde{X}_k)I_{(\cdot, i)}\right\|_2^p \middle| \mathcal{F}_{t_k}\right] h^{-p/2} \\
&= \mathbb{E}\left[\left\|\int_{t_k}^{t_{k+1}} \int_{t_k}^s \sigma(\tilde{X}_k) \, \mathrm{d}B_u \, \mathrm{d}B_s^{(i)}\right\|_2^p \middle| \mathcal{F}_{t_k}\right] h^{-p/2} \\
&\leq \left(\frac{p(p-1)}{2}\right)^{p/2} h^{-1} \int_{t_k}^{t_{k+1}} \mathbb{E}\left[\left\|\int_{t_k}^s \sigma(\tilde{X}_k) \, \mathrm{d}B_u\right\|_2^p \middle| \mathcal{F}_{t_k}\right] \mathrm{d}s
\end{aligned}$$

$$\leq \left( \frac{p(p-1)}{2} \right)^p h^{-1} \int_{t_k}^{t_{k+1}} s^{(p-2)/2} \int_{t_k}^{s} \mathbb{E}\left[ \left\| \sigma(\tilde{X}_k) \right\|_{\mathrm{F}}^p | \mathcal{F}_{t_k} \right] \mathrm{d}u \, \mathrm{d}s$$

$$\leq \left( \frac{p(p-1)}{2} \right)^p \pi_{1,p}^{\mathrm{F}}(\sigma) \left( 1 + \left\| \tilde{X}_k \right\|_2^{p/2} \right) h^{p/2}.$$

$\square$

**Lemma 14** (Second Moment Bounds for $\tilde{Y}_k$). *The following relation holds*

$$\mathbb{E}\left[ \left\| \tilde{Y}_{k+1} \right\|_2^2 | \mathcal{F}_{t_k} \right] \leq 2^2 3^4 m^2 \mu_2(\sigma)^2 \pi_{1,4}^{\mathrm{F}}(\sigma) \left( 1 + \left\| \tilde{X}_k \right\|_2^2 \right) h^3.$$

*Proof.* By Taylor's Theorem with the remainder in integral form,

$$
\begin{aligned}
\left\| \tilde{Y}_{k+1}^{(i)} \right\|_2 &= \left\| \sigma_i(\tilde{X}_k + \Delta\tilde{H}^{(i)}) - \sigma_i(\tilde{X}_k - \Delta\tilde{H}^{(i)}) \right\|_2 h^{1/2} \\
&= \left\| \int_0^1 \left( \nabla\sigma_i(\tilde{X}_k + \tau\Delta\tilde{H}^{(i)}) - \nabla\sigma_i(\tilde{X}_k - \tau\Delta\tilde{H}^{(i)}) \right) \Delta\tilde{H}^{(i)} \, \mathrm{d}\tau \right\|_2 h^{1/2} \\
&\leq h^{1/2} \int_0^1 \left\| \nabla\sigma_i(\tilde{X}_k + \tau\Delta\tilde{H}^{(i)}) - \nabla\sigma_i(\tilde{X}_k - \tau\Delta\tilde{H}^{(i)}) \right\|_{\mathrm{op}} \left\| \Delta\tilde{H}^{(i)} \right\|_2 \, \mathrm{d}\tau \\
&\leq \mu_2(\sigma) h^{1/2} \left\| \Delta\tilde{H}^{(i)} \right\|_2^2 \int_0^1 2\tau \, \mathrm{d}\tau \\
&\leq \mu_2(\sigma) h^{1/2} \left\| \Delta\tilde{H}^{(i)} \right\|_2^2.
\end{aligned}
\tag{36}
$$

By (36) and Corollary 13,

$$\mathbb{E}\left[ \left\| \tilde{Y}_{k+1}^{(i)} \right\|_2^2 | \mathcal{F}_{t_k} \right] \leq \mu_2(\sigma)^2 \mathbb{E}\left[ \left\| \Delta\tilde{H}^{(i)} \right\|_2^4 | \mathcal{F}_{t_k} \right] h \leq 6^4 \mu_2(\sigma)^2 \pi_{1,4}^{\mathrm{F}}(\sigma) \left( 1 + \left\| \tilde{X}_k \right\|_2^2 \right) h^3$$

Therefore,

$$\mathbb{E}\left[ \left\| \tilde{Y}_{k+1} \right\|_2^2 | \mathcal{F}_{t_k} \right] \leq \frac{m}{4} \sum_{i=1}^m \mathbb{E}\left[ \left\| \tilde{Y}_{k+1}^{(i)} \right\|_2^2 | \mathcal{F}_{t_k} \right] \leq 2^2 3^4 m^2 \mu_2(\sigma)^2 \pi_{1,4}^{\mathrm{F}}(\sigma) \left( 1 + \left\| \tilde{X}_k \right\|_2^2 \right) h^3.$$

$\square$

To prove the following moment bound lemmas for SRK-ID, we recall a standard quadratic moment bound result whose proof we omit and provide a reference of.

**Lemma 15** ([24, Lemma F.1]). *Let even integer $p \geq 2$ and $f : \mathbb{R}^d \to \mathbb{R}^{d \times m}$ be Lipschitz. For $\xi \sim \mathcal{N}(0, I_m)$ independent from the $d$-dimensional random vector $X$, the following relation holds*

$$\mathbb{E}\left[ \|f(X)\xi\|_2^p \right] \leq (p-1)!! \mathbb{E}\left[ \|f(X)\|_{\mathrm{F}}^p \right].$$

### C.1.1 Second Moment Bound

**Lemma 16.** *If the second moment of the initial iterate is finite, then the second moments of Markov chain iterates defined in* (10) *are uniformly bounded, i.e.*

$$\mathbb{E}\left[ \left\| \tilde{X}_k \right\|_2^2 \right] \leq \mathcal{V}_2, \quad \text{for all } k \in \mathbb{N}$$

*where*

$$\mathcal{V}_2 = \mathbb{E}\left[ \left\| \tilde{X}_0 \right\|_2^2 \right] + M_2,$$

*and constants $M_1$ and $M_2$ are given in the proof, if the constant step size*

$$h < 1 \wedge \frac{1}{m^2} \wedge \frac{\alpha^2}{4M_1^2}.$$

*Proof.* By direct computation,

$$\left\|\tilde{X}_{k+1}\right\|_2^2 = \left\|\tilde{X}_k\right\|_2^2 + \left\|b(\tilde{X}_k)\right\|_2^2 h^2 + \left\|\sigma(\tilde{X}_k)\xi_{k+1}\right\|_2^2 h + \left\|\tilde{Y}_{k+1}\right\|_2^2$$
$$+ 2\left\langle \tilde{X}_k, b(\tilde{X}_k)\right\rangle h + 2\left\langle \tilde{X}_k, \sigma(\tilde{X}_k)\xi_{k+1}\right\rangle h^{1/2} + 2\left\langle \tilde{X}_k, \tilde{Y}_{k+1}\right\rangle$$
$$+ 2\left\langle b(\tilde{X}_k), \sigma(\tilde{X}_k)\xi_{k+1}\right\rangle h^{3/2} + 2\left\langle b(\tilde{X}_k), \tilde{Y}_{k+1}\right\rangle h$$
$$+ 2\left\langle \sigma(\tilde{X}_k)\xi_{k+1}, \tilde{Y}_{k+1}\right\rangle h^{1/2}.$$

By Lemma 15 and dissipativity,

$$\mathbb{E}\left[2\left\langle \tilde{X}_k, b(\tilde{X}_k)\right\rangle h + \left\|\sigma(\tilde{X}_k)\xi_{k+1}\right\|_2^2 h \middle| \mathcal{F}_{t_k}\right] = 2\left\langle \tilde{X}_k, b(\tilde{X}_k)\right\rangle h + \left\|\sigma(\tilde{X}_k)\right\|_F^2 h$$
$$\leq -\alpha\left\|\tilde{X}_k\right\|_2^2 h + \beta h.$$

We bound the remaining terms by direct computation. By linear growth,

$$\left\|b(\tilde{X}_k)\right\|_2^2 h^2 \leq \pi_{1,2}(b)\left(1 + \left\|\tilde{X}_k\right\|_2^2\right)h^2.$$

By Lemma 14, for $h < 1 \wedge 1/m^2$,

$$\mathbb{E}\left[\left\|\tilde{Y}_{k+1}\right\|_2^2 \middle| \mathcal{F}_{t_k}\right] \leq 2^2 3^4 m^2 \mu_2(\sigma)^2 \pi_{1,4}^F(\sigma)\left(1 + \left\|\tilde{X}_k\right\|_2^2\right)h^3$$
$$\leq 2^2 3^4 m \mu_2(\sigma)^2 \pi_{1,4}^F(\sigma)\left(1 + \left\|\tilde{X}_k\right\|_2^2\right)h^{3/2}.$$

By Lemma 14,

$$\mathbb{E}\left[\left\langle \tilde{X}_k, \tilde{Y}_{k+1}\right\rangle \middle| \mathcal{F}_{t_k}\right] \leq \left\|\tilde{X}_k\right\|_2 \mathbb{E}\left[\left\|\tilde{Y}_{k+1}\right\|_2 \middle| \mathcal{F}_{t_k}\right]$$
$$\leq \left\|\tilde{X}_k\right\|_2 \mathbb{E}\left[\left\|\tilde{Y}_{k+1}\right\|_2^2 \middle| \mathcal{F}_{t_k}\right]^{1/2}$$
$$\leq 2^2 3^2 m \mu_2(\sigma)\pi_{1,4}^F(\sigma)^{1/2}\left(1 + \left\|\tilde{X}_k\right\|_2^2\right)h^{3/2}.$$

Similarly, by Lemma 14,

$$\mathbb{E}\left[\left\langle b(\tilde{X}_k), \tilde{Y}_{k+1}\right\rangle \middle| \mathcal{F}_{t_k}\right] \leq \left\|b(\tilde{X}_k)\right\|_2 \mathbb{E}\left[\left\|\tilde{Y}_{k+1}\right\|_2 \middle| \mathcal{F}_{t_k}\right]$$
$$\leq \left\|b(\tilde{X}_k)\right\|_2 \mathbb{E}\left[\left\|\tilde{Y}_{k+1}\right\|_2^2 \middle| \mathcal{F}_{t_k}\right]^{1/2}$$
$$\leq 2^2 3^2 m \mu_2(\sigma)\pi_{1,4}^F(\sigma)^{1/2}\pi_{1,1}(b)\left(1 + \left\|\tilde{X}_k\right\|_2^2\right)h^{3/2}.$$

By Lemma 14 and Lemma 15,

$$\mathbb{E}\left[\left\langle \sigma(\tilde{X}_k)\xi_{k+1}, \tilde{Y}_{k+1}\right\rangle \middle| \mathcal{F}_{t_k}\right] \leq \mathbb{E}\left[\left\|\sigma(\tilde{X}_k)\xi_{k+1}\right\|_2 \left\|\tilde{Y}_{k+1}\right\|_2 \middle| \mathcal{F}_{t_k}\right]$$
$$\leq \mathbb{E}\left[\left\|\sigma(\tilde{X}_k)\xi_{k+1}\right\|_2^2 \middle| \mathcal{F}_{t_k}\right]^{1/2} \mathbb{E}\left[\left\|\tilde{Y}_{k+1}\right\|_2^2 \middle| \mathcal{F}_{t_k}\right]^{1/2}$$
$$\leq \mathbb{E}\left[\left\|\sigma(\tilde{X}_k)\right\|_F^2 \middle| \mathcal{F}_{t_k}\right]^{1/2} \mathbb{E}\left[\left\|\tilde{Y}_{k+1}\right\|_2^2 \middle| \mathcal{F}_{t_k}\right]^{1/2}$$
$$\leq 2^2 3^2 m \mu_2(\sigma)\pi_{1,4}^F(\sigma)^{1/2}\pi_{1,2}^F(\sigma)^{1/2}\left(1 + \left\|\tilde{X}_k\right\|_2^2\right)h^{3/2}.$$

Putting things together, for $h < 1 \wedge \alpha^2/(4M_1^2)$,

$$\mathbb{E}\left[\left\|\tilde{X}_{k+1}\right\|_2^2 | \mathcal{F}_{t_k}\right] \leq \left(1 - \alpha h + M_1 h^{3/2}\right)\left\|\tilde{X}_k\right\|_2^2 + \beta h + M_1 h^{3/2}$$

$$\leq (1 - \alpha h/2)\left\|\tilde{X}_k\right\|_2^2 + \beta h + M_1 h^{3/2},$$

where

$$M_1 = \pi_{1,2}(b) + 2^3 3^2 m \mu_2(\sigma) \pi_{1,4}^{\mathrm{F}}(\sigma)^{1/2}\left(1 + \mu_2(\sigma)\pi_{1,4}^{\mathrm{F}}(\sigma)^{1/2} + \pi_{1,1}(b) + \pi_{1,2}^{\mathrm{F}}(\sigma)^{1/2}\right).$$

Unrolling the recursion gives the following for $h < 1 \wedge 1/m^2$

$$\mathbb{E}\left[\left\|\tilde{X}_k\right\|_2^2\right] \leq \mathbb{E}\left[\left\|\tilde{X}_0\right\|_2^2\right] + 2\left(\beta + M_1 h^{1/2}\right)/\alpha$$

$$\leq \mathbb{E}\left[\left\|\tilde{X}_0\right\|_2^2\right] + M_2, \quad \text{for all } k \in \mathbb{N},$$

where

$$M_2 = 2\left(\beta + \pi_{1,2}(b)\pi_{1,2}(b) + 2^3 3^2 \mu_2(\sigma)\pi_{1,4}^{\mathrm{F}}(\sigma)^{1/2}\left(1 + \mu_2(\sigma)\pi_{1,4}^{\mathrm{F}}(\sigma)^{1/2} + \pi_{1,1}(b) + \pi_{1,2}^{\mathrm{F}}(\sigma)^{1/2}\right)\right)/\alpha.$$

$\square$

### C.1.2 $2n$th Moment Bound

Before bounding the $2n$th moments, we first generalize Lemma 14 to arbitrary even moments.

**Lemma 17.** *Let even integer $p \geq 2$ and $\tilde{Z}_{k+1} = \tilde{Y}_{k+1} h^{-3/2}$. Then, the following relation holds*

$$\mathbb{E}\left[\left\|\tilde{Z}_{k+1}\right\|_2^p | \mathcal{F}_{t_k}\right] \leq m^p \mu_2(\sigma)^p \left(\frac{2p(2p-1)}{2}\right)^{2p} \pi_{1,2p}^{\mathrm{F}}(\sigma)\left(1 + \left\|\tilde{X}_k\right\|_2^p\right).$$

*Proof.* For $i \in \{1, 2, \ldots, m\}$, by (36),

$$\left\|\tilde{Z}_{k+1}^{(i)}\right\|_2 = \tilde{Y}_{k+1}^{(i)} h^{-3/2} \leq \mu_2(\sigma) h^{-1}\left\|\Delta \tilde{H}^{(i)}\right\|_2^2.$$

Hence, by Corollary 13,

$$\mathbb{E}\left[\left\|\tilde{Z}_{k+1}^{(i)}\right\|_2^p | \mathcal{F}_{t_k}\right] \leq \mu_2(\sigma)^p h^{-p}\mathbb{E}\left[\left\|\Delta \tilde{H}^{(i)}\right\|_2^{2p} | \mathcal{F}_{t_k}\right]$$

$$\leq \mu_2(\sigma)^p \left(\frac{2p(2p-1)}{2}\right)^{2p} \pi_{1,2p}^{\mathrm{F}}(\sigma)\left(1 + \left\|\tilde{X}_k\right\|_2^p\right).$$

The remaining follows easily from Lemma 31. $\square$

**Lemma 18.** *For $n \in \mathbb{N}_+$, if the $2n$th moment of the initial iterate is finite, then the $2n$th moments of Markov chain iterates defined in (10) are uniformly bounded, i.e.*

$$\mathbb{E}\left[\left\|\tilde{X}_k\right\|_2^{2n}\right] \leq \mathcal{V}_{2n}, \quad \text{for all } k \in \mathbb{N}$$

*where*

$$\mathcal{V}_{2n} = \mathbb{E}\left[\left\|\tilde{X}_0\right\|_2^{2n}\right] + \frac{2}{n\alpha}\left(\beta \mathcal{V}_{2(n-1)} + 2^{23n-1}10^n n^{8n}\pi_{1,2n}(b)\pi_{1,8n}^{\mathrm{F}}(\sigma)^{1/2}\mu_2(\sigma)^{2n}\right),$$

*if the step size*

$$h < 1 \wedge \frac{1}{m^2} \wedge \frac{\alpha^2}{4M_1^2} \wedge \min\left\{\left(\frac{\alpha l}{2M_{3,l}}\right)^2 : l = 2, \ldots, n\right\}.$$

*Proof.* Our proof is by induction. The base case is given in Lemma 16. For the inductive case, we prove that the $2n$th moment is uniformly bounded by a constant, assuming the $2(n\text{-}1)$th moment is uniformly bounded by a constant.

By the multinomial theorem,

$$
\begin{aligned}
\mathbb{E}\left[\left\|\tilde{X}_{k+1}\right\|_2^2 |\mathcal{F}_{t_k}\right] =&\mathbb{E}\Bigg[\Bigg(\left\|\tilde{X}_k\right\|_2^2 + \left\|b(\tilde{X}_k)\right\|_2^2 h^2 + \left\|\sigma(\tilde{X}_k)\xi_{k+1}\right\|_2^2 h + \left\|\tilde{Y}_{k+1}\right\|_2^2 \\
&+ 2\left\langle \tilde{X}_k, b(\tilde{X}_k)\right\rangle h + 2\left\langle \tilde{X}_k, \sigma(\tilde{X}_k)\xi_{k+1}\right\rangle h^{1/2} + 2\left\langle \tilde{X}_k, \tilde{Y}_{k+1}\right\rangle \\
&+ 2\left\langle b(\tilde{X}_k), \sigma(\tilde{X}_k)\xi_{k+1}\right\rangle h^{3/2} + 2\left\langle b(\tilde{X}_k), \tilde{Y}_{k+1}\right\rangle h \\
&+ 2\left\langle \sigma(\tilde{X}_k)\xi_{k+1}, \tilde{Y}_{k+1}\right\rangle h^{1/2}\Bigg)^n |\mathcal{F}_{t_k}\Bigg] \\
=&\mathbb{E}\Bigg[\Bigg(\left\|\tilde{X}_k\right\|_2^2 + \left\|b(\tilde{X}_k)\right\|_2^2 h^2 + \left\|\sigma(\tilde{X}_k)\xi_{k+1}\right\|_2^2 h + \left\|\tilde{Z}_{k+1}\right\|_2^2 h^3 \\
&+ 2\left\langle \tilde{X}_k, b(\tilde{X}_k)\right\rangle h + 2\left\langle \tilde{X}_k, \sigma(\tilde{X}_k)\xi_{k+1}\right\rangle h^{1/2} + 2\left\langle \tilde{X}_k, \tilde{Z}_{k+1}\right\rangle h^{3/2} \\
&+ 2\left\langle b(\tilde{X}_k), \sigma(\tilde{X}_k)\xi_{k+1}\right\rangle h^{3/2} + 2\left\langle b(\tilde{X}_k), \tilde{Z}_{k+1}\right\rangle h^{5/2} \\
&+ 2\left\langle \sigma(\tilde{X}_k)\xi_{k+1}, \tilde{Z}_{k+1}\right\rangle h^2\Bigg)^n |\mathcal{F}_{t_k}\Bigg] \\
=&\left\|\tilde{X}_k\right\|_2^{2n} + \mathbb{E}\left[A|\mathcal{F}_{t_k}\right] h + \mathbb{E}\left[B|\mathcal{F}_{t_k}\right] h^{3/2},
\end{aligned}
$$

where by the Cauchy–Schwarz inequality,

$$
A =n\|\tilde{X}_k\|_2^{2(n-1)}\left(2\langle \tilde{X}_k, b(\tilde{X}_k)\rangle + \|\sigma(\tilde{X}_k)\xi_{k+1}\|_2^2\right) + 2n(n-1)\|\tilde{X}_k\|_2^{2(n-2)}\langle \tilde{X}_k, \sigma(\tilde{X}_k)\xi_{k+1}\rangle,
$$

$$
B \leq \sum_{(k_1,\ldots,k_{10})\in J} 2^n \binom{n}{k_1 \ \ldots \ k_{10}} \left\|\tilde{X}_k\right\|_2^{p_1} \left\|b(\tilde{X}_k)\right\|_2^{p_2} \left\|\sigma(\tilde{X}_k)\xi_{k+1}\right\|_2^{p_3} \left\|\tilde{Z}_{k+1}\right\|_2^{p_4},
$$

the indicator set

$$
\begin{aligned}
J =\Bigg\{(k_1,\ldots,k_{10}) \in \mathbb{N}^{10} : \ &k_1 + \cdots + k_{10} = n, \\
&2k_2 + k_3 + 3k_4 + k_5 + \frac{k_6}{2} + \frac{3k_7}{2} + \frac{3k_8}{2} + \frac{5k_9}{2} + 2k_{10} > 1\Bigg\},
\end{aligned}
$$

and with slight abuse of notation, we hide the explicit dependence on $k_1,\ldots,k_{10}$ for the exponents

$$
\begin{aligned}
p_1 =&2k_1 + k_5 + k_6 + k_7, \\
p_2 =&2k_2 + k_5 + k_8 + k_9, \\
p_3 =&2k_3 + k_6 + k_8 + k_{10}, \\
p_4 =&2k_4 + k_7 + k_9 + k_{10}.
\end{aligned}
$$

By dissipativity,

$$
\mathbb{E}\left[A|\mathcal{F}_{t_k}\right] \leq -n\alpha \left\|\tilde{X}_k\right\|_2^{2n} + n\beta \left\|\tilde{X}_k\right\|_2^{2(n-1)}. \tag{37}
$$

Note that $p_1 + p_2 + p_3 + p_4 = 2n$. Since $h < 1 \wedge 1/m^2$, we may cancel out the $m$ factor in some of the terms. One can verify that the only remaining term that is $m$-dependent is

$$
\left\langle \tilde{X}_k, \tilde{Z}_{k+1}\right\rangle = \mathcal{O}(mh^{3/2}).
$$

Using this information, Lemma 17, Lemma 15, the Cauchy–Schwarz inequality, and $p_3 + p_4 \leq 2n$,

$$
\mathbb{E}\left[B|\mathcal{F}_{t_k}\right]
$$

$$\leq \sum_{(k_1,\ldots,k_{10})\in J} 2^n \binom{n}{k_1 \ \ldots \ k_{10}} \left\|\tilde{X}_k\right\|_2^{p_1} \left\|b(\tilde{X}_k)\right\|_2^{p_2} \mathbb{E}\left[\left\|\sigma(\tilde{X}_k)\xi_{k+1}\right\|_2^{p_3} \left\|\tilde{Z}_{k+1}m^{-1}\right\|_2^{p_4} |\mathcal{F}_{t_k}\right] m$$

$$\leq \sum_{(k_1,\ldots,k_{10})\in J} 2^n \binom{n}{k_1 \ \ldots \ k_{10}} \left\|\tilde{X}_k\right\|_2^{p_1} \left\|b(\tilde{X}_k)\right\|_2^{p_2} \mathbb{E}\left[\left\|\sigma(\tilde{X}_k)\xi_{k+1}\right\|_2^{2p_3} |\mathcal{F}_{t_k}\right]^{1/2} \mathbb{E}\left[\left\|\tilde{Z}_{k+1}m^{-1}\right\|_2^{2p_4} |\mathcal{F}_{t_k}\right]^{1/2} m$$

$$\leq \sum_{(k_1,\ldots,k_{10})\in J} 2^n \binom{n}{k_1 \ \ldots \ k_{10}} \left\|\tilde{X}_k\right\|_2^{p_1} \pi_{1,p_2}(b)\left(1+\left\|\tilde{X}_k\right\|_2^{p_2}\right)((2p_3-1)!!)^{1/2}\pi_{1,p_3}^{\mathrm{F}}(\sigma)\left(1+\left\|\tilde{X}_k\right\|_2^{p_3}\right)$$

$$\times \mu_2(\sigma)^{p_4}\left(8p_4^2\right)^{2p_4}\pi_{1,4p_4}^{\mathrm{F}}(\sigma)^{1/2}\left(1+\left\|\tilde{X}_k\right\|_2^{p_4}\right) m$$

$$\leq 2^n \left(1+\left\|\tilde{X}_k\right\|_2\right)^{2n} \sum_{(k_1,\ldots,k_{10})\in J} \pi_{1,p_2}(b)((2p_3-1)!!)^{1/2}\pi_{1,p_3}^{\mathrm{F}}(\sigma)\mu_2(\sigma)^{p_4}\left(8p_4^2\right)^{2p_4}\pi_{1,4p_4}^{\mathrm{F}}(\sigma)^{1/2}\binom{n}{k_1 \ \ldots \ k_{10}} m$$

$$\leq 2^n \left(1+\left\|\tilde{X}_k\right\|_2\right)^{2n} \pi_{1,2n}(b)\pi_{1,2n}^{\mathrm{F}}(\sigma)\mu_2(\sigma)^{2n}\pi_{1,4n}^{\mathrm{F}}(\sigma)^{1/2}2^{20n}n^{8n}\sum_{\substack{k_1,\ldots,k_{10}\in\mathbb{N}\\ k_1+\cdots+k_{10}=n}}\binom{n}{k_1 \ \ldots \ k_{10}} m$$

$$\leq 2^{23n-1}10^n n^{8n}\pi_{1,2n}(b)\pi_{1,8n}^{\mathrm{F}}(\sigma)^{1/2}\mu_2(\sigma)^{2n}m\left(1+\left\|\tilde{X}_k\right\|_2^{2n}\right). \tag{38}$$

By the inductive hypothesis, (37) and (38), and $h < 1 \wedge n^2\alpha^2/(4M_{3,n}^2)$, we obtain the recursion

$$\mathbb{E}\left[\mathbb{E}\left[\left\|\tilde{X}_{k+1}\right\|_2^{2n}|\mathcal{F}_{t_k}\right]\right] \leq \left(1-n\alpha h + M_{3,n}h^{3/2}\right)\mathbb{E}\left[\left\|\tilde{X}_k\right\|_2^{2n}\right] + n\beta h\mathbb{E}\left[\left\|\tilde{X}_k\right\|_2^{2(n-1)}\right] + M_3 h^{3/2}$$

$$\leq \left(1-n\alpha h + M_{3,n}h^{3/2}\right)\mathbb{E}\left[\left\|\tilde{X}_k\right\|_2^{2n}\right] + n\beta\mathcal{V}_{2(n-1)}h + M_{3,n}h^{3/2}$$

$$\leq \left(1-n\alpha h/2\right)\mathbb{E}\left[\left\|\tilde{X}_k\right\|_2^{2n}\right] + n\beta\mathcal{V}_{2(n-1)}h + M_{3,n}h^{3/2},$$

where the constant $M_{3,n} = 2^{23n-1}10^n n^{8n}\pi_{1,2n}(b)\pi_{1,8n}^{\mathrm{F}}(\sigma)^{1/2}\mu_2(\sigma)^{2n}m$.

For $h < 1 \wedge 1/m^2$, by unrolling the recursion, we obtain

$$\mathbb{E}\left[\left\|\tilde{X}_k\right\|_2^{2n}\right] \leq \mathbb{E}\left[\left\|\tilde{X}_0\right\|_2^{2n}\right] + \frac{2}{n\alpha}\left(n\beta\mathcal{V}_{2(n-1)} + M_{3,n}h^{1/2}\right) \leq \mathcal{V}_{2n}, \quad \text{for all } k \in \mathbb{N},$$

where

$$\mathcal{V}_{2n} = \mathbb{E}\left[\left\|\tilde{X}_0\right\|_2^{2n}\right] + \frac{2}{n\alpha}\left(\beta\mathcal{V}_{2(n-1)} + 2^{23n-1}10^n n^{8n}\pi_{1,2n}(b)\pi_{1,8n}^{\mathrm{F}}(\sigma)^{1/2}\mu_2(\sigma)^{2n}\right).$$

$\square$

## C.2 Local Deviation Orders

In this section, we verify the local deviation orders for SRK-ID. The proofs are again by matching up terms in the Itô-Taylor expansion of the continuous-time process to terms in the Taylor expansion of the numerical integration scheme. Extra care needs to be taken for a tight dimension dependence.

**Lemma 19.** *Suppose $X_t$ is the continuous-time process defined by (1) initiated from some iterate of the Markov chain $X_0$ defined by (10), then the second moment of $X_t$ is uniformly bounded, i.e.*

$$\mathbb{E}\left[\|X_t\|_2^2\right] \leq \mathcal{V}_2', \quad \text{for all } t \geq 0.$$

*where $\mathcal{V}_2' = \mathcal{V}_2 + \beta/\alpha$.*

*Proof.* By Itô's lemma and dissipativity,

$$\frac{\mathrm{d}}{\mathrm{d}t}\mathbb{E}\left[\|X_t\|_2^2\right] = \mathbb{E}\left[2\langle X_t, b(X_t)\rangle + \|\sigma(X_t)\|_{\mathrm{F}}^2\right] \leq -\alpha\mathbb{E}\left[\|X_t\|_2^2\right] + \beta.$$

Moreover, by Grönwall's inequality,

$$\mathbb{E}\left[\|X_t\|_2^2\right] \leq e^{-\alpha t}\mathbb{E}\left[\|X_0\|_2^2\right] + \beta/\alpha \leq \mathcal{V}_2 + \beta/\alpha = \mathcal{V}_2'.$$

$\square$

**Lemma 20** (Second Moment of Change). *Suppose $X_t$ is the continuous-time process defined by ([1](#)) initiated from some iterate of the Markov chain $X_0$ defined by ([10](#)), then*

$$\mathbb{E}\left[\|X_t - X_0\|_2^2\right] \le D_0 t, \quad \text{for all } 0 \le t \le 1,$$

*where $D_0 = 2\left(\pi_{1,2}(b) + \pi_{1,2}^{\mathrm{F}}(\sigma)\right)(1 + \mathcal{V}_2')$.*

*Proof.* By Itô isometry,

$$
\begin{aligned}
\mathbb{E}\left[\|X_t - X_0\|_2^2\right] &= \mathbb{E}\left[\left\|\int_0^t b(X_s)\,\mathrm{d}s + \int_0^t \sigma(X_s)\,\mathrm{d}B_s\right\|_2^2\right] \\
&\le 2\mathbb{E}\left[\left\|\int_0^t b(X_s)\,\mathrm{d}s\right\|_2^2 + \left\|\int_0^t \sigma(X_s)\,\mathrm{d}B_s\right\|_2^2\right] \\
&\le 2t\int_0^t \mathbb{E}\left[\|b(X_s)\|_2^2\right]\,\mathrm{d}s + 2\int_0^t \mathbb{E}\left[\|\sigma(X_s)\|_{\mathrm{F}}^2\right]\,\mathrm{d}s \\
&\le 2\pi_{1,2}(b)t\int_0^t \mathbb{E}\left[1 + \|X_s\|_2^2\right]\,\mathrm{d}s + 2\pi_{1,2}^{\mathrm{F}}(\sigma)\int_0^t \mathbb{E}\left[1 + \|X_s\|_2^2\right]\,\mathrm{d}s \\
&\le 2\left(\pi_{1,2}(b) + \pi_{1,2}^{\mathrm{F}}(\sigma)\right)(1 + \mathcal{V}_2')\,t.
\end{aligned}
$$

$\square$

To bound the fourth moment of change in continuous-time, we use the following lemma.

**Lemma 21** ([24], adapted from Lemma A.1). *Assuming $\{X_t\}_{t\ge 0}$ is the solution to the SDE ([1](#)), under the condition that the drift coefficient $b$ and diffusion coefficient $\sigma$ are Lipschitz. If $\sigma$ has satisfies the following sublinear growth condition*

$$\|\sigma(x)\|_{\mathrm{F}}^l \le \pi_{1,l}^{\mathrm{F}}(\sigma)\left(1 + \|x\|^{l/2}\right), \quad \text{for all } x \in \mathbb{R}^d, l = 1, 2, \ldots,$$

*and the diffusion is dissipative, then for $n \ge 2$, we have the following relation*

$$\mathcal{A}\|x\|_2^n \le -\frac{\alpha n}{4}\|x\|_2^n + \beta_n,$$

*where the (infinitesimal) generator $\mathcal{A}$ is defined as*

$$\mathcal{A}f(x) = \lim_{t\downarrow 0}\frac{\mathbb{E}\left[f(X_t)|X_0 = x\right] - f(x)}{t},$$

*and the constant $\beta_n = \mathcal{O}(d^{\frac{n}{2}})$.*

*Proof.* By definition of the generator and dissipativity,

$$
\begin{aligned}
\mathcal{A}\|x\|_2^n &= n\|x\|_2^{n-2}\langle x, b(x)\rangle + \frac{n}{2}\|x\|_2^{n-2}\|\sigma(x)\|_{\mathrm{F}}^2 + \frac{n(n-2)}{2}\|x\|_2^{n-4}\left\langle \mathrm{vec}(xx^\top), \mathrm{vec}(\sigma\sigma^\top(x))\right\rangle \\
&\le -\frac{\alpha n}{2}\|x\|_2^n + \frac{\beta n}{2}\|x\|_2^{n-2} + \frac{n(n-2)}{2}\|x\|_2^{n-2}\pi_{1,2}^{\mathrm{F}}(\sigma)\left(1 + \|x\|_2\right) \\
&= -\frac{\alpha n}{2}\|x\|_2^n + \frac{n(n-2)}{2}\pi_{1,2}^{\mathrm{F}}(\sigma)\|x\|_2^{n-1} + \left(\frac{\beta n}{2} + \frac{n(n-2)}{2}\pi_{1,2}^{\mathrm{F}}(\sigma)\right)\|x\|_2^{n-2}.
\end{aligned}
$$

By Young's inequality,

$$
\begin{aligned}
\frac{n(n-2)}{2}\pi_{1,2}^{\mathrm{F}}(\sigma)\|x\|_2^{n-1} &= \frac{n(n-2)}{2}\pi_{1,2}^{\mathrm{F}}(\sigma)\left(\frac{8}{\alpha n}\right)^{\frac{n-1}{n}}\cdot\|x\|_2^{n-1}\left(\frac{\alpha n}{8}\right)^{\frac{n-1}{n}} \\
&\le \frac{1}{n}\left(\frac{n(n-2)}{2}\right)^n \pi_{1,2}^{\mathrm{F}}(\sigma)^n\left(\frac{8}{\alpha n}\right)^{n-1} + \frac{n-1}{n}\frac{\alpha n}{8}\|x\|_2^n \\
&= \frac{(n-2)^n}{2^{2n-3}\alpha^{n-1}}\pi_{1,2}^{\mathrm{F}}(\sigma)^n + \frac{\alpha(n-1)}{8}\|x\|_2^n.
\end{aligned}
$$

Similarly,

$$\left(\frac{\beta n}{2} + \frac{n(n-2)}{2}\pi_{1,2}^{\mathrm{F}}(\sigma)\right)\|x\|_2^{n-2} = \left(\frac{\beta n}{2} + \frac{n(n-2)}{2}\pi_{1,2}^{\mathrm{F}}(\sigma)\right)\left(\frac{8}{\alpha n}\right)^{\frac{n-2}{n}} \cdot \|x\|_2^{n-2}\left(\frac{\alpha n}{8}\right)^{\frac{n-2}{n}}$$

$$\leq \frac{2}{n}\left(\frac{\beta n}{2} + \frac{n(n-2)}{2}\pi_{1,2}^{\mathrm{F}}(\sigma)\right)^{\frac{n}{2}}\left(\frac{\alpha n}{8}\right)^{\frac{n-2}{2}} + \frac{\alpha(n-2)}{8}\|x\|_2^n.$$

We define the following shorthand notation

$$\beta_n^{(1)} = \frac{(n-2)^n}{2^{2n-3}\alpha^{n-1}}\pi_{1,2}^{\mathrm{F}}(\sigma)^n = \mathcal{O}(d^{\frac{n}{2}}),$$

$$\beta_n^{(2)} = \frac{2}{n}\left(\frac{\beta n}{2} + \frac{n(n-2)}{2}\pi_{1,2}^{\mathrm{F}}(\sigma)\right)^{\frac{n}{2}}\left(\frac{\alpha n}{8}\right)^{\frac{n-2}{2}} = \mathcal{O}(d^{\frac{n}{2}}).$$

Putting things together, we obtain the following bound

$$\mathcal{A}\|x\|_2^n \leq -\frac{\alpha n}{2}\|x\|_2^n + \frac{\alpha(n-1)}{8}\|x\|_2^n + \frac{\alpha(n-2)}{8}\|x\|_2^n + \beta_n^{(1)} + \beta_n^{(2)}$$

$$\leq -\frac{\alpha n}{4}\|x\|_2^n + \beta_n,$$

where $\beta_n = \beta_n^{(1)} + \beta_n^{(2)} = \mathcal{O}(d^{\frac{n}{2}})$. $\hspace{1cm}\square$

**Lemma 22.** *Suppose $X_t$ is the continuous-time process defined by (1) initiated from some iterate of the Markov chain $X_0$ defined by (10), then the fourth moment of $X_t$ is uniformly bounded, i.e.*

$$\mathbb{E}\left[\|X_t\|_2^4\right] \leq \mathcal{V}_4', \quad \text{for all } t \geq 0,$$

*where $\mathcal{V}_4' = \mathcal{V}_4 + \beta_4/\alpha$.*

*Proof.* By Dynkin's formula [48] applied to the function $(t, x) \mapsto e^{\alpha t}\|x\|_2^4$ and Lemma 21,

$$e^{\alpha t}\mathbb{E}\left[\|X_t\|_2^4 |\mathcal{F}_0\right] = \|X_0\|_2^4 + \int_0^t \mathbb{E}\left[\alpha e^{\alpha s}\|X_s\|_2^4 + e^{\alpha s}\mathcal{A}\|X_s\|_2^4 |\mathcal{F}_0\right]\,\mathrm{d}s$$

$$\leq \|X_0\|_2^4 + \int_0^t \mathbb{E}\left[\alpha e^{\alpha s}\|X_s\|_2^4 - \alpha e^{\alpha s}\|X_s\|_2^4 + e^{\alpha s}\beta_4 |\mathcal{F}_0\right]\,\mathrm{d}s$$

$$= \|X_0\|_2^4 + \frac{e^{\alpha t} - 1}{\alpha}\beta_4.$$

Hence,

$$\mathbb{E}\left[\|X_t\|_2^4\right] = \mathbb{E}\left[\mathbb{E}\left[\|X_t\|_2^4 |\mathcal{F}_0\right]\right] \leq e^{-\alpha t}\mathbb{E}\left[\|X_0\|_2^4\right] + \beta_4/\alpha \leq \mathcal{V}_4 + \beta_4/\alpha = \mathcal{V}_4'.$$

$\hspace{1cm}\square$

**Lemma 23** (Fourth Moment of Change)**.** *Suppose $X_t$ is the continuous-time process defined by (1) from some iterate of the Markov chain $X_0$ defined by (10), then*

$$\mathbb{E}\left[\|X_t - X_0\|_2^4\right] \leq D_1 t^2, \quad \text{for all } 0 \leq t \leq 1,$$

*where $D_1 = 8\left(\pi_{1,4}(b) + 36\pi_{1,4}^{\mathrm{F}}(\sigma)\right)(1 + \mathcal{V}_4')$.*

*Proof.* By Theorem 12,

$$\mathbb{E}\left[\|X_t - X_0\|_2^4\right] = \mathbb{E}\left[\left\|\int_0^t b(X_s)\,\mathrm{d}s + \int_0^t \sigma(X_s)\,\mathrm{d}B_s\right\|_2^4\right]$$

$$\leq 8\mathbb{E}\left[\left\|\int_0^t b(X_s)\,\mathrm{d}s\right\|_2^4 + \left\|\int_0^t \sigma(X_s)\,\mathrm{d}B_s\right\|_2^4\right]$$

$$\leq 8t^3\int_0^t \mathbb{E}\left[\|b(X_s)\|_2^4\right]\,\mathrm{d}s + 288t\mathbb{E}\left[\int_0^t \|\sigma(X_s)\|_{\mathrm{F}}^4\,\mathrm{d}s\right]$$

$$\leq 8\left(\pi_{1,4}(b) + 36\pi_{1,4}^{\mathrm{F}}(\sigma)\right)(1 + \mathcal{V}_4')t^2.$$

$\hspace{1cm}\square$

### C.2.1 Local Mean-Square Deviation

**Lemma 24.** *Suppose $X_t$ and $\tilde{X}_t$ are the continuous-time process defined by (1) and Markov chain defined by (10) for time $t \geq 0$, respectively. If $X_t$ and $\tilde{X}_t$ are initiated from the same iterate of the Markov chain $X_0$, and they share the same Brownian motion, then*

$$\mathbb{E}\left[\left\|X_t - \tilde{X}_t\right\|_2^2\right] \leq D_3 t^3, \quad \text{for all } 0 \leq t \leq 1,$$

*where*

$$
\begin{aligned}
D_3 = \Big( & 16 D_0 \mu_1(b)^2 + \frac{16}{3}\mu_2(\sigma)^2 \pi_{1,4}^{1/2} D_1^{1/2}(1 + \mathcal{V}_2'^{1/2})m^2 + \frac{16}{3}\mu_1(\sigma)^4 m^2 D_0 \\
& + 16\mu_1(\sigma)^2 \pi_{1,2}(b)^2(1 + \mathcal{V}_2')m + 4m^3 \mu_2(\sigma)^2 \pi_{1,4}^{\mathrm{F}}(\sigma)(1 + \mathcal{V}_2') \\
& + 2^7 3^4 m^2 \mu_2(\sigma)^2 \pi_{1,4}^{\mathrm{F}}(\sigma)\,(1 + \mathcal{V}_2) \Big).
\end{aligned}
$$

*Proof.* Recall the operators $L$ and $\Lambda_i$ $(i = 1, \ldots, m)$ defined in (5). By Itô's lemma,

$$
\begin{aligned}
X_t - X_0 &= \int_0^t b(X_s)\,\mathrm{d}s + \sigma(X_0)B_t \\
&\quad + \sum_{i=1}^m \sum_{l=1}^m \int_0^t \int_0^s \Lambda_l(\sigma_i)(X_u)\,\mathrm{d}B_u^{(l)}\,\mathrm{d}B_s^{(i)} + \sum_{i=1}^m \int_0^t \int_0^s L(\sigma_i)(X_u)\,\mathrm{d}u\,\mathrm{d}B_s^{(i)} \\
&= \int_0^t b(X_s)\,\mathrm{d}s + \sigma(X_0)B_t + \sum_{i=1}^m \sum_{l=1}^m \int_0^t \int_0^s \nabla\sigma_i(X_u)\sigma_l(X_u)\,\mathrm{d}B_u^{(l)}\,\mathrm{d}B_s^{(i)} + S(t),
\end{aligned}
$$

where

$$
S(t) = \underbrace{\sum_{i=1}^m \int_0^t \int_0^s \nabla\sigma_i(X_u)b(X_u)\,\mathrm{d}u\,\mathrm{d}B_s^{(i)}}_{S_1(t)} + \underbrace{\frac{1}{2}\sum_{i=1}^m \sum_{l=1}^m \int_0^t \int_0^s \nabla^2\sigma_i(X_u)[\sigma_l(X_u),\,\sigma_l(X_u)]\,\mathrm{d}u\,\mathrm{d}B_s^{(i)}}_{S_2(t)}.
$$

By Taylor's theorem with the remainder in integral form,

$$
\begin{aligned}
\sigma_i(\tilde{H}_1^{(i)}) &= \sigma_i(X_0) + \nabla\sigma_i(X_0)\Delta\tilde{H}^{(i)} + \phi_1^{(i)}(t), \\
\sigma_i(\tilde{H}_2^{(i)}) &= \sigma_i(X_0) - \nabla\sigma_i(X_0)\Delta\tilde{H}^{(i)} + \phi_2^{(i)}(t),
\end{aligned}
$$

where

$$
\begin{aligned}
\phi_1^{(i)}(t) &= \int_0^1 (1 - \tau)\nabla^2\sigma_i(X_0 + \tau\Delta\tilde{H}^{(i)})[\Delta\tilde{H}^{(i)},\,\Delta\tilde{H}^{(i)}]\,\mathrm{d}\tau, \\
\phi_2^{(i)}(t) &= \int_0^1 (1 - \tau)\nabla^2\sigma_i(X_0 - \tau\Delta\tilde{H}^{(i)})[\Delta\tilde{H}^{(i)},\,\Delta\tilde{H}^{(i)}]\,\mathrm{d}\tau, \\
\Delta\tilde{H}^{(i)} &= \sum_{l=1}^m \sigma_l(X_0)\frac{I_{(l,i)}}{\sqrt{t}}.
\end{aligned}
$$

Hence,

$$
\begin{aligned}
X_t - \tilde{X}_t &= \int_0^t (b(X_s) - b(X_0))\,\mathrm{d}s \\
&\quad + \sum_{i=1}^m \sum_{l=1}^m \int_0^t \int_0^s (\nabla\sigma_i(X_u)\sigma_l(X_u) - \nabla\sigma_i(X_0)\sigma_l(X_0))\,\mathrm{d}B_u^{(l)}\,\mathrm{d}B_s^{(i)} \\
&\quad + S(t) - \frac{1}{2}\sum_{i=1}^m \left(\phi_1^{(i)}(t) - \phi_2^{(i)}(t)\right)\sqrt{t}.
\end{aligned}
$$

Since $b$ is $\mu_1(b)$-Lipschitz,

$$\mathbb{E}\left[\left\|\int_0^t (b(X_s) - b(X_0))\ \mathrm{d}s\right\|_2^2\right] \leq \mu_1(b)^2 t \int_0^t \mathbb{E}\left[\|X_s - X_0\|_2^2\right]\ \mathrm{d}s$$

$$\leq \mu_1(b)^2 t \int_0^t D_0 s\ \mathrm{d}s$$

$$\leq \frac{1}{2} D_0 \mu_1(b)^2 t^3.$$

We define the following,

$$A(t) = A_1(t) + A_2(t) = \sum_{i=1}^m \sum_{l=1}^m \int_0^t \int_0^s (\nabla\sigma_i(X_u)\sigma_l(X_u) - \nabla\sigma_i(X_0)\sigma_l(X_0))\ \mathrm{d}B_u^{(l)}\ \mathrm{d}B_s^{(i)},$$

where

$$A_1(t) = \sum_{i=1}^m \sum_{l=1}^m \int_0^t \int_0^s (\nabla\sigma_i(X_u)\sigma_l(X_u) - \nabla\sigma_i(X_0)\sigma_l(X_u))\ \mathrm{d}B_u^{(l)}\ \mathrm{d}B_s^{(i)},$$

$$A_2(t) = \sum_{i=1}^m \sum_{l=1}^m \int_0^t \int_0^s (\nabla\sigma_i(X_0)\sigma_l(X_u) - \nabla\sigma_i(X_0)\sigma_l(X_0))\ \mathrm{d}B_u^{(l)}\ \mathrm{d}B_s^{(i)}.$$

By Itô isometry and the Cauchy-Schwarz inequality,

$$\mathbb{E}\left[\|A_1(t)\|_2^2\right] = \sum_{i=1}^m \sum_{l=1}^m \int_0^t \int_0^s \mathbb{E}\left[\|\nabla\sigma_i(X_u)\sigma_l(X_u) - \nabla\sigma_i(X_0)\sigma_l(X_u)\|_2^2\right]\ \mathrm{d}u\ \mathrm{d}s$$

$$\leq \sum_{i=1}^m \sum_{l=1}^m \int_0^t \int_0^s \mathbb{E}\left[\|\nabla\sigma_i(X_u) - \nabla\sigma_i(X_0)\|_{\mathrm{op}}^2 \|\sigma_l(X_u)\|_2^2\right]\ \mathrm{d}u\ \mathrm{d}s$$

$$\leq \sum_{i=1}^m \sum_{l=1}^m \int_0^t \int_0^s \mathbb{E}\left[\|\nabla\sigma_i(X_u) - \nabla\sigma_i(X_0)\|_{\mathrm{op}}^4\right]^{1/2} \mathbb{E}\left[\|\sigma_l(X_u)\|_2^4\right]^{1/2}\ \mathrm{d}u\ \mathrm{d}s$$

$$\leq \mu_2(\sigma)^2 \pi_{1,4}(\sigma)^{1/2} m^2 \int_0^t \int_0^s \mathbb{E}\left[\|X_u - X_0\|_2^4\right]^{1/2} \mathbb{E}\left[1 + \|X_u\|_2^2\right]^{1/2}\ \mathrm{d}u\ \mathrm{d}s$$

$$\leq \mu_2(\sigma)^2 \pi_{1,4}(\sigma)^{1/2} D_1^{1/2} \left(1 + \mathcal{V}_2'^{1/2}\right) m^2 \frac{t^3}{6}. \tag{39}$$

Similarly,

$$\mathbb{E}\left[\|A_2(t)\|_2^2\right] = \sum_{i=1}^m \sum_{l=1}^m \int_0^t \int_0^s \mathbb{E}\left[\|\nabla\sigma_i(X_0)\sigma_l(X_u) - \nabla\sigma_i(X_0)\sigma_l(X_0)\|_{\mathrm{op}}^2\right]\ \mathrm{d}u\ \mathrm{d}s$$

$$\leq \sum_{i=1}^m \sum_{l=1}^m \int_0^t \int_0^s \mathbb{E}\left[\|\nabla\sigma_i(X_0)\|_{\mathrm{op}}^2 \|\sigma_l(X_u) - \sigma_l(X_0)\|_2^2\right]\ \mathrm{d}u\ \mathrm{d}s$$

$$\leq \mu_1(\sigma)^2 \sum_{i=1}^m \sum_{l=1}^m \int_0^t \int_0^s \mathbb{E}\left[\|\sigma_l(X_u) - \sigma_l(X_0)\|_2^2\right]\ \mathrm{d}u\ \mathrm{d}s$$

$$\leq \mu_1(\sigma)^4 m^2 \int_0^t \int_0^s \mathbb{E}\left[\|X_u - X_0\|_2^2\right]\ \mathrm{d}u\ \mathrm{d}s$$

$$\leq \frac{1}{6} \mu_1(\sigma)^4 m^2 D_0 t^3. \tag{40}$$

By Itô isometry,

$$\mathbb{E}\left[\|S_1(t)\|_2^2\right] = \sum_{i=1}^m \int_0^t s \int_0^s \mathbb{E}\left[\|\nabla\sigma_i(X_u)b(X_u)\|_2^2\right]\ \mathrm{d}u\ \mathrm{d}s$$

$$\leq \sum_{i=1}^{m} \int_0^t s \int_0^s \mathbb{E}\left[\|\nabla \sigma_i(X_u)\|_{\mathrm{op}}^2 \|b(X_u)\|_2^2\right] \mathrm{d}u \, \mathrm{d}s$$

$$\leq \mu_1(\sigma)^2 \pi_{1,2}(b)^2 \sum_{i=1}^{m} \int_0^t s \int_0^s \mathbb{E}\left[1 + \|X_u\|_2^2\right] \mathrm{d}u \, \mathrm{d}s$$

$$= \frac{1}{2}\mu_1(\sigma)^2 \pi_{1,2}(b)^2 \left(1 + \mathcal{V}_2'\right) m t^3. \tag{41}$$

Similarly,

$$\mathbb{E}\left[\|S_2(t)\|_2^2\right] = \frac{1}{4}\sum_{i=1}^{m} \int_0^t \mathbb{E}\left[\left\|\int_0^s \sum_{l=1}^{m} \nabla^2 \sigma_i(X_u)[\sigma_l(X_u),\, \sigma_l(X_u)]\,\mathrm{d}u\right\|_2^2\right] \mathrm{d}s$$

$$\leq \frac{1}{4}m \sum_{i=1}^{m}\sum_{l=1}^{m} \int_0^t s \int_0^s \mathbb{E}\left[\left\|\nabla^2 \sigma_i(X_u)[\sigma_l(X_u),\, \sigma_l(X_u)]\right\|_2^2\right] \mathrm{d}u \, \mathrm{d}s$$

$$\leq \frac{1}{4}m \sum_{i=1}^{m}\sum_{l=1}^{m} \int_0^t s \int_0^s \mathbb{E}\left[\left\|\nabla^2 \sigma_i(X_u)\right\|_{\mathrm{op}}^2 \|\sigma_l(X_u)\|_2^4\right] \mathrm{d}u \, \mathrm{d}s$$

$$\leq \frac{1}{4}\sigma_2(\sigma)^2 \pi_{1,4}(\sigma) m^3 \int_0^t s \int_0^s \mathbb{E}\left[1 + \|X_u\|_2^2\right] \mathrm{d}u \, \mathrm{d}s$$

$$\leq \frac{1}{8}\sigma_2(\sigma)^2 \pi_{1,4}(\sigma) m^3 \left(1 + \mathcal{V}_2'\right) t^3. \tag{42}$$

By Corollary 13,

$$\mathbb{E}\left[\left\|\Delta \tilde{H}^{(i)}\right\|_2^4\right] = \mathbb{E}\left[\mathbb{E}\left[\left\|\Delta \tilde{H}^{(i)}\right\|_2^4 | \mathcal{F}_{t_k}\right]\right]$$

$$\leq 6^4 \pi_{1,4}^{\mathrm{F}}(\sigma)\mathbb{E}\left[1 + \left\|\tilde{X}_k\right\|_2^2\right] t^2$$

$$\leq 6^4 \pi_{1,4}^{\mathrm{F}}(\sigma) \left(1 + \mathcal{V}_2\right) t^2.$$

Now, we bound the second moments of $\phi_1^{(i)}(t)$ and $\phi_2^{(i)}(t)$,

$$\mathbb{E}\left[\left\|\phi_1^{(i)}(t)\right\|_2^2\right] = \mathbb{E}\left[\left\|\int_0^1 (1-\tau)\nabla^2 \sigma_i(X_0 + \tau\Delta \tilde{H}^{(i)})[\Delta \tilde{H}^{(i)},\, \Delta \tilde{H}^{(i)}]\,\mathrm{d}\tau\right\|_2^2\right]$$

$$\leq \mathbb{E}\left[\left\|\nabla^2 \sigma_i(X_0 + \tau\Delta \tilde{H}^{(i)})\right\|_{\mathrm{op}}^2 \left\|\Delta \tilde{H}^{(i)}\right\|_2^4\right]$$

$$\leq 6^4 \mu_2(\sigma)^2 \pi_{1,4}^{\mathrm{F}}(\sigma)(1 + \mathcal{V}_2)t^2. \tag{43}$$

Similarly,

$$\mathbb{E}\left[\left\|\phi_2^{(i)}(t)\right\|_2^2\right] \leq 6^4 \mu_2(\sigma)^2 \pi_{1,4}^{\mathrm{F}}(\sigma)(1 + \mathcal{V}_2)t^2. \tag{44}$$

Hence, by (43) and (44),

$$\mathbb{E}\left[\left\|\frac{1}{2}\sum_{i=1}^{m}\left(\phi_1^{(i)}(t) - \phi_2^{(i)}(t)\right)\sqrt{t}\right\|_2^2\right] \leq \frac{m}{4}t \sum_{i=1}^{m} \mathbb{E}\left[\left\|\phi_1^{(i)}(t) - \phi_2^{(i)}(t)\right\|_2^2\right]$$

$$\leq 2^2 3^4 m^2 \mu_2(\sigma)^2 \pi_{1,4}^{\mathrm{F}}(\sigma)(1 + \mathcal{V}_2)t^3. \tag{45}$$

Combining (39), (40), (41), (42), and (45),

$$\mathbb{E}\left[\left\|X_t - \tilde{X}_t\right\|_2^2\right] \leq 32\mathbb{E}\left[\left\|\int_0^t (b(X_s) - b(X_0))\,\mathrm{d}s\right\|_2^2\right]$$

$$+ 32\mathbb{E}\left[\|A_1(t)\|_2^2 + \|A_2(t)\|_2^2\right]$$

$$+ 32\mathbb{E}\left[\|S_1(t)\|_2^2 + \|S_2(t)\|_2^2\right]$$

$$+ 32\mathbb{E}\left[\left\|\frac{1}{2}\sum_{i=1}^{m}\left(\phi_1^{(i)}(t) - \phi_2^{(i)}(t)\right)\sqrt{t}\right\|_2^2\right]$$

$$\leq \Big(16 D_0 \mu_1(b)^2 + \frac{16}{3}\mu_2(\sigma)^2 \pi_{1,4}^{1/2} D_1^{1/2}(1 + \mathcal{V}_2'^{1/2})m^2 + \frac{16}{3}\mu_1(\sigma)^4 m^2 D_0$$

$$+ 16\mu_1(\sigma)^2 \pi_{1,2}(b)^2(1 + \mathcal{V}_2')m + 4m^3\mu_2(\sigma)^2\pi_{1,4}^{\mathrm{F}}(\sigma)(1 + \mathcal{V}_2')$$

$$+ 2^7 3^4 m^2 \mu_2(\sigma)^2 \pi_{1,4}^{\mathrm{F}}(\sigma)\left(1 + \mathcal{V}_2\right)\Big)t^3.$$

$$\square$$

### C.2.2 Local Mean Deviation

**Lemma 25.** *Suppose $X_t$ and $\tilde{X}_t$ are the continuous-time process defined by (1) and Markov chain defined by (10) for time $t \geq 0$, respectively. If $X_t$ and $\tilde{X}_t$ are initiated from the same iterate of the Markov chain $X_0$, and they share the same Brownian motion, then*

$$\mathbb{E}\left[\left\|\mathbb{E}\left[X_t - \tilde{X}_t | \mathcal{F}_0\right]\right\|_2^2\right] \leq D_4 t^4, \quad \text{for all } 0 \leq t \leq 1,$$

*where*

$$D_4 = \Big(\frac{4}{3}\mu_1(b)^2 \pi_{1,2}(b)\left(1 + \mathcal{V}_2'\right) + \frac{1}{3}\mu_2(b)^2\pi_{1,4}(\sigma)\left(1 + \mathcal{V}_2'\right)m^2$$

$$+ 2^4 3^5 5^6 \mu_3(\sigma)^2 \pi_{1,6}^{\mathrm{F}}(\sigma)\left(1 + \mathcal{V}_4^{3/4}\right)\Big).$$

*Proof.* Recall the operators $L$ and $\Lambda_i$ ($i = 1, \ldots, m$) defined in (5). By Itô's lemma,

$$X_t - X_0 = b(X_0)t + \sum_{i=1}^{m}\int_0^t \sigma_i(X_s)\,\mathrm{d}B_s^{(i)}$$

$$+ \sum_{i=1}^{m}\int_0^t\int_0^s \Lambda_i(b)(X_u)\,\mathrm{d}B_u^{(i)}\,\mathrm{d}s + \int_0^t\int_0^s L(b)(X_u)\,\mathrm{d}u\,\mathrm{d}s$$

$$= b(X_0)t + \sum_{i=1}^{m}\int_0^t \sigma_i(X_s)\,\mathrm{d}B_s^{(i)} + \sum_{i=1}^{m}\int_0^t\int_0^s \nabla b(X_u)\sigma_i(X_u)\,\mathrm{d}B_u^{(i)}\,\mathrm{d}s + \bar{S}(t),$$

where

$$\bar{S}(t) = \underbrace{\int_0^t\int_0^s \nabla b(X_u)b(X_u)\,\mathrm{d}u\,\mathrm{d}s}_{\bar{S}_1(t)} + \underbrace{\frac{1}{2}\sum_{i=1}^{m}\int_0^t\int_0^s \nabla^2 b(X_u)[\sigma_i(X_u), \sigma_i(X_u)]\,\mathrm{d}u\,\mathrm{d}s}_{\bar{S}_2(t)}.$$

Now, we bound the second moments of $\bar{S}_1(t)$ and $\bar{S}_2(t)$,

$$\mathbb{E}\left[\|\bar{S}_1(t)\|_2^2\right] = \mathbb{E}\left[\left\|\int_0^t\int_0^s \nabla b(X_u)b(X_u)\,\mathrm{d}u\,\mathrm{d}s\right\|_2^2\right]$$

$$\leq t\int_0^t s\int_0^s \mathbb{E}\left[\|\nabla b(X_u)b(X_u)\|_2^2\right]\,\mathrm{d}u\,\mathrm{d}s$$

$$\leq t\int_0^t s\int_0^s \mathbb{E}\left[\|\nabla b(X_u)\|_{\mathrm{op}}^2 \|b(X_u)\|_2^2\right]\,\mathrm{d}u\,\mathrm{d}s$$

$$\leq \mu_1(b)^2\pi_{1,2}(b)t\int_0^t s\int_0^s \mathbb{E}\left[1 + \|X_u\|_2^2\right]\,\mathrm{d}u\,\mathrm{d}s$$

$$\leq \frac{1}{3}\mu_1(b)^2\pi_{1,2}(b)\left(1 + \mathcal{V}_2'\right)t^4. \tag{46}$$

Similarly,

$$
\begin{aligned}
\mathbb{E}\left[\left\|\bar{S}_2(t)\right\|_2^2\right] =&\mathbb{E}\left[\left\|\frac{1}{2}\sum_{i=1}^m\int_0^t\int_0^s\nabla^2 b(X_u)[\sigma_i(X_u),\,\sigma_i(X_u)]\,\mathrm{d}u\,\mathrm{d}s\right\|_2^2\right]\\
\leq&\frac{m}{4}\sum_{i=1}^m t\int_0^t s\int_0^s\mathbb{E}\left[\left\|\nabla^2 b(X_u)[\sigma_i(X_u),\,\sigma_i(X_u)]\right\|_2^2\right]\,\mathrm{d}u\,\mathrm{d}s\\
\leq&\frac{m}{4}\sum_{i=1}^m t\int_0^t s\int_0^s\mathbb{E}\left[\left\|\nabla^2 b(X_u)\right\|_{\mathrm{op}}^2\left\|\sigma_i(X_u)\right\|_2^4\right]\,\mathrm{d}u\,\mathrm{d}s\\
\leq&\frac{m}{4}\mu_2(b)^2\sum_{i=1}^m t\int_0^t s\int_0^s\mathbb{E}\left[\left\|\sigma_i(X_u)\right\|_2^4\right]\,\mathrm{d}u\,\mathrm{d}s\\
\leq&\frac{m^2}{4}\mu_2(b)^2\pi_{1,4}(\sigma)t\int_0^t s\int_0^s\mathbb{E}\left[1+\|X_u\|_2^2\right]\,\mathrm{d}u\,\mathrm{d}s\\
\leq&\frac{1}{12}\mu_2(b)^2\pi_{1,4}(\sigma)\left(1+\mathcal{V}_2'\right)m^2 t^4.
\end{aligned}
\tag{47}
$$

By Corollary 13,

$$
\begin{aligned}
\mathbb{E}\left[\left\|\Delta\tilde{H}^{(i)}\right\|_2^6\right] =&\mathbb{E}\left[\mathbb{E}\left[\left\|\Delta\tilde{H}^{(i)}\right\|_2^6|\mathcal{F}_{t_k}\right]\right]\\
\leq&3^6 5^6\pi_{1,6}^{\mathrm{F}}(\sigma)\mathbb{E}\left[1+\left\|\tilde{X}_k\right\|_2^3\right]t^3\\
\leq&3^6 5^6\pi_{1,6}^{\mathrm{F}}(\sigma)\left(1+\mathcal{V}_4^{3/4}\right)t^3.
\end{aligned}
$$

Now, we bound the second moment of the difference between $\phi_1^{(i)}(t)$ and $\phi_2^{(i)}(t)$,

$$
\begin{aligned}
\mathbb{E}\left[\left\|\phi_1^{(i)}(t)-\phi_2^{(i)}(t)\right\|_2^2\right] \leq&\mathbb{E}\left[\int_0^1\left\|\nabla^2\sigma_i(X_0+\tau\Delta\tilde{H}^{(i)})-\nabla^2\sigma_i(X_0-\tau\Delta\tilde{H}^{(i)})\right\|_{\mathrm{op}}^2\left\|\Delta\tilde{H}^{(i)}\right\|_2^4\,\mathrm{d}\tau\right]\\
\leq&\mu_3(\sigma)^2\int_0^1\mathbb{E}\left[\left\|2\tau\Delta\tilde{H}^{(i)}\right\|_2^2\left\|\Delta\tilde{H}^{(i)}\right\|_2^4\right]\,\mathrm{d}\tau\\
\leq&\frac{4}{3}\mu_3(\sigma)^2\mathbb{E}\left[\left\|\Delta\tilde{H}^{(i)}\right\|_2^6\right]\\
\leq&2^2 3^5 5^6\mu_3(\sigma)^2\pi_{1,6}^{\mathrm{F}}(\sigma)\left(1+\mathcal{V}_4^{3/4}\right)t^3.
\end{aligned}
\tag{48}
$$

Hence, combining (46), (47), and (48),

$$
\begin{aligned}
\mathbb{E}\left[\left\|\mathbb{E}\left[X_t-\tilde{X}_t|\mathcal{F}_0\right]\right\|_2^2\right] =&\mathbb{E}\left[\left\|\mathbb{E}\left[\bar{S}(t)|\mathcal{F}_0\right]-\mathbb{E}\left[\frac{1}{2}\sum_{i=1}^m\left(\phi_1^{(i)}(t)-\phi_2^{(i)}(t)\right)\sqrt{t}|\mathcal{F}_0\right]\right\|_2^2\right]\\
\leq&4\mathbb{E}\left[\left\|\bar{S}_1(t)\right\|_2^2+\left\|\bar{S}_2(t)\right\|_2^2\right]+4\mathbb{E}\left[\left\|\frac{1}{2}\sum_{i=1}^m\left(\phi_1^{(i)}(t)-\phi_2^{(i)}(t)\right)\sqrt{t}\right\|_2^2\right]\\
\leq&\left(\frac{4}{3}\mu_1(b)^2\pi_{1,2}(b)\left(1+\mathcal{V}_2'\right)+\frac{1}{3}\mu_2(b)^2\pi_{1,4}(\sigma)\left(1+\mathcal{V}_2'\right)m^2\right.\\
&\left.+2^4 3^5 5^6\mu_3(\sigma)^2\pi_{1,6}^{\mathrm{F}}(\sigma)\left(1+\mathcal{V}_4^{3/4}\right)\right)t^4.
\end{aligned}
$$

$\square$

### C.3  Invoking Theorem 1

Now, we invoke Theorem 1 with our derived constants. We obtain that if the constant step size

$$
h<1\wedge C_h\wedge\frac{1}{2\alpha}\wedge\frac{1}{8\mu_1(b)^2+8\mu_1^{\mathrm{F}}(\sigma)^2},
$$

where

$$C_h = \frac{1}{m^2} \wedge \frac{\alpha^2}{4M_1^2} \wedge \frac{\alpha^2}{M_{3,2}^2},$$

and the smoothness conditions in Theorem 3 of the drift and diffusion coefficients are satisfied for a uniformly dissipative diffusion, then the uniform local deviation bounds (7) hold with $\lambda_1 = D_3$ and $\lambda_2 = D_4$, and consequently the bound (8) holds. This concludes that to converge to a sufficiently small positive tolerance $\epsilon$, $\tilde{\mathcal{O}}(d^{3/4}m^2\epsilon^{-1})$ iterations are required, since $D_3$ is of order $\mathcal{O}(d^{3/2}m^3)$, and $D_4$ is of order $\mathcal{O}(d^{3/2}m^2)$. Note that the dimension dependence worsens if one were to further convert the Frobenius norm dependent constants to be based on the operator norm.

## D  Convergence Rate for Example 2

### D.1  Moment Bound

Verifying the order conditions in Theorem 1 of the EM scheme for uniformly dissipative diffusions requires bounding the second moments of the Markov chain. Recall, dissipativity (Definition C.1) follows from uniform dissipativity of the Itô diffusion.

**Lemma 26.** *If the second moment of the initial iterate is finite, then the second moments of Markov chain iterates defined in* (4) *are uniformly bounded, i.e.*

$$\mathbb{E}\left[\left\|\tilde{X}_k\right\|_2^2\right] \le \mathcal{W}_2, \quad \textit{for all } k \in \mathbb{N},$$

*where $\mathcal{W}_2 = \mathbb{E}\left[\left\|\tilde{X}_0\right\|_2^2\right] + 2(\pi_{1,2}(b) + \beta)/\alpha$, if the constant step size $h < 1 \wedge \alpha/(2\pi_{1,2}(b))$.*

*Proof.* By direct computation,

$$\begin{aligned}
\left\|\tilde{X}_{k+1}\right\|_2^2 = &\left\|\tilde{X}_k\right\|_2^2 + \left\|b(\tilde{X}_k)\right\|_2^2 h^2 + \left\|\sigma(\tilde{X}_k)\xi_{k+1}\right\|_2^2 h \\
&+ 2\left\langle \tilde{X}_k, b(\tilde{X}_k)\right\rangle h + 2\left\langle \tilde{X}_k, \sigma(\tilde{X}_k)\xi_{k+1}\right\rangle h^{1/2} \\
&+ 2\left\langle b(\tilde{X}_k), \sigma(\tilde{X}_k)\xi_{k+1}\right\rangle h^{3/2}.
\end{aligned}$$

Recall by Lemma 15 and dissipativity,

$$\mathbb{E}\left[2\left\langle \tilde{X}_k, b(\tilde{X}_k)\right\rangle h + \left\|\sigma(\tilde{X}_k)\xi_{k+1}\right\|_2^2 h | \mathcal{F}_{t_k}\right] \le -\alpha\left\|\tilde{X}_k\right\|_2^2 h + \beta h.$$

By odd moments of Gaussian variables being zero and the step size condition,

$$\begin{aligned}
\mathbb{E}\left[\left\|\tilde{X}_{k+1}\right\|_2^2 | \mathcal{F}_{t_k}\right] \le &(1 - \alpha h)\left\|\tilde{X}_k\right\|_2^2 + \left\|b(\tilde{X}_k)\right\|_2^2 h^2 + \beta h \\
\le &(1 - \alpha h + \pi_{1,2}(b)h^2)\left\|\tilde{X}_k\right\|_2^2 + \pi_{1,2}(b)h^2 + \beta h \\
\le &(1 - \alpha h/2)\left\|\tilde{X}_k\right\|_2^2 + \pi_{1,2}(b)h^2 + \beta h.
\end{aligned}$$

By unrolling the recursion,

$$\mathbb{E}\left[\left\|\tilde{X}_k\right\|_2^2\right] \le \mathbb{E}\left[\left\|\tilde{X}_0\right\|_2^2\right] + 2(\pi_{1,2}(b) + \beta)/\alpha, \quad \textit{for all } k \in \mathbb{N}.$$

$\square$

### D.2  Local Deviation Orders

Before verifying the local deviation orders, we first state two auxiliary lemmas. We omit the proofs, since they are almost identical to that of Lemma 6 and Lemma 7, respectively.

**Lemma 27.** *Suppose $X_t$ is the continuous-time process defined by* (1) *initiated from some iterate of the Markov chain $X_0$ defined by* (4), *then the second moment of $X_t$ is uniformly bounded, i.e.*

$$\mathbb{E}\left[\|X_t\|_2^2\right] \leq \mathcal{W}_2 + \beta/\alpha = \mathcal{W}_2', \quad for \ all \ t \geq 0.$$

**Lemma 28.** *Suppose $X_t$ is the continuous-time process defined by* (1) *initiated from some iterate of the Markov chain $X_0$ defined by* (4), *then*

$$\mathbb{E}\left[\|X_t - X_0\|_2^2\right] \leq E_0 t, \quad for \ all \ t \geq 0,$$

*where $E_0 = 2\left(\pi_{1,2}(b) + \pi_{1,2}^{\mathrm{F}}(\sigma)\right)(1 + \mathcal{W}_2')$.*

### D.2.1 Local Mean-Square Deviation

**Lemma 29.** *Suppose $X_t$ and $\tilde{X}_t$ are the continuous-time process defined by* (1) *and Markov chain defined by* (4) *for time $t \geq 0$, respectively. If $X_t$ and $\tilde{X}_t$ are initiated from the same iterate of the Markov chain $X_0$ and share the same Brownian motion, then*

$$\mathbb{E}\left[\left\|X_t - \tilde{X}_t\right\|_2^2\right] \leq E_1 t^2, \quad for \ all \ 0 \leq t \leq 1,$$

*where $E_1 = \left(\mu_1(b)^2 + \mu_1^{\mathrm{F}}(\sigma)^2\right)E_0$.*

*Proof.* By Itô isometry and Lipschitz of the drift and diffusion coefficients,

$$
\begin{aligned}
\mathbb{E}\left[\left\|X_t - \tilde{X}_t\right\|_2^2\right] \leq & 2\mathbb{E}\left[\left\|\int_0^t (b(X_s) - b(X_0))\ \mathrm{d}s\right\|_2^2\right] + 2\mathbb{E}\left[\left\|\int_0^t (\sigma(X_s) - \sigma(X_0))\ \mathrm{d}B_s\right\|_2^2\right] \\
\leq & 2t\mathbb{E}\left[\int_0^t \|b(X_s) - b(X_0)\|_2^2\ \mathrm{d}s\right] + 2\mathbb{E}\left[\int_0^t \|\sigma(X_s) - \sigma(X_0)\|_{\mathrm{F}}^2\ \mathrm{d}s\right] \\
\leq & 2\left(\mu_1(b)^2 t + \mu_1^{\mathrm{F}}(\sigma)^2\right)\int_0^t \mathbb{E}\left[\|X_s - X_0\|_2^2\right]\ \mathrm{d}s \\
\leq & \left(\mu_1(b)^2 + \mu_1^{\mathrm{F}}(\sigma)^2\right)E_0 t^2.
\end{aligned}
$$

$\square$

### D.2.2 Local Mean Deviation

**Lemma 30.** *Suppose $X_t$ and $\tilde{X}_t$ are the continuous-time process defined by* (1) *and Markov chain defined by* (4) *for time $t \geq 0$, respectively. If $X_t$ and $\tilde{X}_t$ are initiated from the same iterate of the Markov chain $X_0$ and share the same Brownian motion, then*

$$\mathbb{E}\left[\left\|\mathbb{E}\left[X_t - \tilde{X}_t|\mathcal{F}_0\right]\right\|_2^2\right] \leq E_2 t^3, \quad for \ all \ 0 \leq t \leq 1,$$

*where $E_2 = \mu_1(b)E_0/2$.*

*Proof.* By Itô's lemma,

$$
\begin{aligned}
X_t - X_0 = & \int_0^t b(X_s)ds + \sigma(X_0)B_t \\
& + \sum_{i=1}^m \sum_{l=1}^m \int_0^t \int_0^s \Lambda_l(\sigma_i)(X_u)\ \mathrm{d}B_u^{(l)}\ \mathrm{d}B_s^{(i)} + \sum_{i=1}^m \int_0^t \int_0^s L(\sigma_i)(X_u)\ \mathrm{d}u\ \mathrm{d}B_s^{(i)}.
\end{aligned}
$$

Since the last two terms in the above inequality are Martingales,

$$\mathbb{E}\left[X_t - X_0|\mathcal{F}_0\right] = \mathbb{E}\left[\int_0^t (b(X_s) - b(X_0))\ \mathrm{d}s|\mathcal{F}_0\right].$$

Hence, by Jensen's inequality,

$$
\begin{aligned}
\mathbb{E}\left[\left\|\mathbb{E}\left[X_t - \tilde{X}_t|\mathcal{F}_0\right]\right\|_2^2\right] =& \mathbb{E}\left[\left\|\mathbb{E}\left[\int_0^t (b(X_s) - b(X_0))\ \mathrm{d}s|\mathcal{F}_0\right]\right\|_2^2\right] \\
\leq& \mathbb{E}\left[\left\|\int_0^t (b(X_s) - b(X_0))\ \mathrm{d}s\right\|_2^2\right] \\
\leq& \mu_1(b)t \int_0^t \mathbb{E}\left[\|X_s - X_0\|_2^2\right]\ \mathrm{d}s \\
\leq& \mu_1(b)E_0 t^3/2.
\end{aligned}
$$

$\square$

### D.3 Invoking Theorem 1

Now, we invoke Theorem 1 with our derived constants. We obtain that if the constant step size

$$
h < 1 \wedge \frac{\alpha}{2\pi_{1,2}(b)} \wedge \frac{1}{2\alpha} \wedge \frac{1}{8\mu_1(b)^2 + 8\mu_1^{\mathrm{F}}(\sigma)^2},
$$

and the smoothness conditions of the drift and diffusion coefficients are satisfied for a uniformly dissipative diffusion, then the uniform local deviation bounds (7) hold with $\lambda_1 = E_1$ and $\lambda_2 = E_2$, and consequently the bound (8) holds. This concludes that for a sufficiently small positive tolerance $\epsilon$, $\tilde{\mathcal{O}}(d\epsilon^{-2})$ iterations are required, since both $E_1$ and $E_2$ are of order $\mathcal{O}(d)$. If one were to convert the Frobenius norm dependent constants to be based on the operator norm, then $E_1$ is of order $\mathcal{O}(d(d+m)^2)$, and $E_2$ is of order $\mathcal{O}(d(d+m))$. This yields the convergence rate of $\tilde{\mathcal{O}}(d(d+m)^2\epsilon^{-2})$.

## E  Convergence of SRK-LD Under an Unbiased Stochastic Oracle

We provide an informal analysis on the scenario where the oracle is stochastic. We denote the new interpolated values under the stochastic oracle as $\hat{H}_1$ and $\hat{H}_2$, and the new iterate value as $\hat{X}_k$. We assume (i) the stochastic oracle is unbiased, i.e. $\mathbb{E}[\hat{\nabla}f(x)] = f(x)$ for all $x \in \mathbb{R}^d$, (ii) the stochastic oracle has finite variance at the Markov chain iterates and "interpolated" values, i.e. $\mathbb{E}[\|\hat{\nabla}f(Y) - \nabla f(Y)\|_2^2] \leq \sigma^2 d$, for some finite $\sigma$, where $Y$ may be $\hat{X}_k$, $\hat{H}_1$, or $\hat{H}_2$[6], and (iii) the randomness in the stochastic oracle is independent of that of the Brownian motion.

Fix iteration index $k \in \mathbb{N}$, let $\tilde{D}_h^{(k)}$ and $\hat{D}_h^{(k)}$ denote the local deviations under the exact and stochastic oracles, respectively. Then, assuming the step size is chosen sufficiently small such that the Markov chain moments are bounded,

$$
\begin{aligned}
\mathbb{E}\left[\left\|\hat{D}_h^{(k)}\right\|_2^2\right] \leq& 2\mathbb{E}\left[\left\|\tilde{D}_h^{(k)}\right\|_2^2\right] + 2\mathbb{E}\left[\left\|\tilde{D}_h^{(k)} - \hat{D}_h^{(k)}\right\|_2^2\right] \\
\leq& 2\mathbb{E}\left[\left\|\tilde{D}_h^{(k)}\right\|_2^2\right] + 4\mathbb{E}\left[\left\|\hat{\nabla}f(\hat{H}_1) - \nabla f(\tilde{H}_1)\right\|_2^2\right] + 4\mathbb{E}\left[\left\|\hat{\nabla}f(\hat{H}_2) - \nabla f(\tilde{H}_2)\right\|_2^2\right] \\
\leq& 2\mathbb{E}\left[\left\|\tilde{D}_h^{(k)}\right\|_2^2\right] + 4\sigma^2 d + 4\mathbb{E}\left[\left\|\hat{\nabla}f(\hat{H}_2) - \nabla f(\hat{H}_2) + \nabla f(\hat{H}_2) - \nabla f(\tilde{H}_2)\right\|_2^2\right] \\
\leq& \mathcal{O}(h^4 + \sigma^2).
\end{aligned}
$$

Similarly, one can derive the new local mean deviation,

$$
\mathbb{E}\left[\left\|\mathbb{E}\left[\hat{D}_h^{(k)}|\mathcal{F}_{t_{k-1}}\right]\right\|_2^2\right] \leq \mathbb{E}\left[\left\|\mathbb{E}\left[\tilde{D}_h^{(k)}|\mathcal{F}_{t_{k-1}}\right] + \mathbb{E}\left[\hat{D}_h^{(k)} - \tilde{D}_h^{(k)}|\mathcal{F}_{t_{k-1}}\right]\right\|_2^2\right]
$$

$$\leq \mathbb{E}\left[\left\|\mathbb{E}\left[\tilde{D}_h^{(k)}|\mathcal{F}_{t_{k-1}}\right]\right\|_2^2\right] + \mathbb{E}\left[\left\|\hat{D}_h^{(k)} - \tilde{D}_h^{(k)}\right\|_2^2\right]$$

$$= \mathcal{O}(h^5 + \sigma^2).$$

One can replace the corresponding terms in (15) and obtain a recursion. Note however, to ensure unrolling the recursion gives a convergence bound, one need that $\sigma^2 < \mathcal{O}(\alpha h)$.

## F   Auxiliary Lemmas

We list standard results used to develop our theorems and include their proofs for completeness.

**Lemma 31.** *For $x_1, \ldots, x_m \in \mathbb{R}$ and $m, n \in \mathbb{N}_+$, we have*

$$\left(\sum_{i=1}^m x_i\right)^n \leq m^{n-1}\sum_{i=1}^m x_i^n.$$

*Proof.* Recall the function $f(x) = x^n$ is convex for $n \in \mathbb{N}_+$. Hence,

$$\left(\frac{\sum_{i=1}^m x_i}{m}\right)^n \leq \frac{\sum_{i=1}^m x_i^n}{m}.$$

Multiplying both sides of the inequality by $m^n$ completes the proof. $\qquad\square$

**Lemma 32.** *For the d-dimensional Brownian motion $\{B_t\}_{t\geq 0}$,*

$$Z_t = \int_0^t \int_0^s \mathrm{d}B_u \ \mathrm{d}s \sim \mathcal{N}\left(0, t^3 I_d/3\right).$$

*Proof.* We consider the case where $d = 1$. The multi-dimensional case follows naturally, since we assume different dimensions of the Brownian motion vector are independent. Let $t_k = \delta k$, we define

$$S_m = \sum_{k=0}^{m-1} B_{t_k}(t_{k+1} - t_k) = \sum_{k=1}^{m-1}\left(B_{t_{k+1}} - B_{t_k}\right)(t_k - t).$$

Since $S_m$ is a sum of Gaussian random variables, it is also Gaussian. By linearity of expectation and independence of Brownian motion increments,

$$\mathbb{E}\left[S_m\right] = 0,$$

$$\mathbb{E}\left[S_m^2\right] = \sum_{k=1}^{m-1}(t_k - t)^2\,\mathbb{E}\left[\left(B_{t_{k+1}} - B_{t_k}\right)^2\right] \to \int_0^t (s-t)^2 \ \mathrm{d}s = t^3/3 \quad \text{as} \quad m \to \infty.$$

Since $S_m \overset{\text{a.s.}}{\to} Z_t$ as $m \to \infty$ by the strong law of large numbers, we conclude that $Z_t \sim \mathcal{N}\left(0, t^3/3\right)$.
$\qquad\square$

**Lemma 33.** *For $n \in \mathbb{N}$ and the d-dimensional Brownian motion $\{B_t\}_{t\geq 0}$,*

$$\mathbb{E}\left[\|B_t\|_2^{2n}\right] = t^n d(d+2)\cdots(d+2n-2).$$

*Proof.* Note $\|B_t\|_2^2$ may be expressed as the sum of squared Gaussian random variables, i.e.

$$\|B_t\|_2^2 = t\sum_{i=1}^d \xi_i^2, \quad \text{where} \quad \xi_i \overset{\text{i.i.d.}}{\sim} \mathcal{N}(0,1).$$

Observe that this is also a multiple of the chi-squared random variable with $d$ degrees of freedom $\chi(d)^2$. Its $n$th moment has the following closed form [57],

$$\mathbb{E}\left[\chi(d)^{2n}\right] = 2^n\frac{\Gamma\left(n + \frac{d}{2}\right)}{\Gamma\left(\frac{d}{2}\right)} = d(d+2)\cdots(d+2n-2).$$

Thus,

$$\mathbb{E}\left[\|B_t\|_2^{2n}\right] = t^n\mathbb{E}\left[\chi(d)^{2n}\right] = t^n d(d+2)\cdots(d+2n-2).$$

$\qquad\square$

**Lemma 34.** *For $f : \mathbb{R}^d \to \mathbb{R}$ which is $C^3$, suppose its Hessian is $\mu_3$-Lipschitz under the operator norm and Euclidean norm, i.e.*

$$\left\|\nabla^2 f(x) - \nabla^2 f(y)\right\|_{\mathrm{op}} \leq \mu_3 \left\|x - y\right\|_2, \quad \textit{for all } x, y \in \mathbb{R}^d.$$

*Then, the vector Laplacian of its gradient is bounded, i.e.*

$$\left\|\vec{\Delta}(\nabla f)(x)\right\|_2 \leq d\mu_3, \quad \textit{for all } x \in \mathbb{R}^d.$$

*Proof.* See proof of Lemma 6 in [12]. □

**Lemma 35.** *For $f : \mathbb{R}^d \to \mathbb{R}$ which is $C^4$, suppose its third derivative is $\mu_4$-Lipschitz under the operator norm and Euclidean norm, i.e.*

$$\left\|\nabla^3 f(x) - \nabla^3 f(y)\right\|_{\mathrm{op}} \leq \mu_4 \left\|x - y\right\|_2, \quad \textit{for all } x, y \in \mathbb{R}^d.$$

*Then, the vector Laplacian of its gradient is $d\mu_4$-Lipschitz, i.e.*

$$\left\|\vec{\Delta}(\nabla f)(x) - \vec{\Delta}(\nabla f)(y)\right\|_2 \leq d\mu_4 \left\|x - y\right\|_2.$$

*Proof.* Let $g(x) = \Delta(f)(x)$. Since $f \in C^4$, we may switch the order of partial derivatives,

$$\left\|\vec{\Delta}(\nabla f)(x) - \vec{\Delta}(\nabla f)(y)\right\|_2 = \left\|\nabla g(x) - \nabla g(y)\right\|_2.$$

By Taylor's theorem with the remainder in integral form,

$$
\begin{aligned}
\left\|\nabla g(x) - \nabla g(y)\right\|_2 &= \left\|\int_0^1 \nabla^2 g\left(y + \tau(x - y)\right)(x - y)\, \mathrm{d}\tau\right\|_2 \\
&\leq \int_0^1 \left\|\nabla^2 g\left(y + \tau(x - y)\right)\right\|_{\mathrm{op}} \left\|x - y\right\|_2 \, \mathrm{d}\tau \\
&\leq \sup_{z \in \mathbb{R}^d} \left\|\nabla^2 g(z)\right\|_{\mathrm{op}} \left\|x - y\right\|_2.
\end{aligned}
$$

Note that $\nabla^2 g(x)$ can be written as a sum of $d$ matrices, each being a sub-tensor of $\nabla^4 f(x)$, due to the the trace operator, i.e.

$$\nabla^2 g(x) = \sum_{i=1}^d G_i(x), \quad \text{where} \quad G_i(x)_{jk} = \partial_{iijk} f(x).$$

Since the operator norm of $\nabla^4 f(x)$ upper bounds the operator norm of each of its sub-tensor,

$$\left\|\nabla^2 g(x)\right\|_{\mathrm{op}} \leq \sum_{i=1}^d \left\|G_i(x)\right\|_{\mathrm{op}} \leq d \left\|\nabla^4 f(x)\right\|_{\mathrm{op}}$$

Recall the third derivative is $\mu_4$-Lipschitz, we obtain

$$\left\|\nabla g(x) - \nabla g(y)\right\|_2 \leq d\mu_3 \left\|x - y\right\|_2.$$

□

# G   Estimating the Wasserstein Distance

For a Borel measure $\mu$ defined on a compact and separable topological space $\mathcal{X}$, a sample-based empirical measure $\mu_n$ may asymptotically serve as a proxy to $\mu$ in the $W_p$ sense for $p \in [1, \infty)$, i.e.

$$W_p(\mu, \hat{\mu}_n) \xrightarrow{\mu\text{-a.s.}} 0.$$

This is a consequence of the Wasserstein distance metrizing weak convergence [62] and that the empirical measure converges weakly to $\mu$ almost surely [60].

However, in the finite-sample setting, this distance is typically non-negligible and worsens as the dimensionality increases. Specifically, generalizing previous results based on the 1-Wasserstein distance [17, 16], Weed and Bach [64] showed that for $p \in [1, \infty)$,

$$W_p(\mu, \hat{\mu}_n) \gtrsim n^{-1/t},$$

where $t$ is less than the lower Wasserstein dimension $d_*(\mu)$. This presents a severe challenge in estimating the 2-Wasserstein distance between probability measures using samples.

To better detect convergence, we zero center a simple sample-based estimator by subtracting the null responses and obtain the following new estimator:

$$\tilde{W}_2^2(\mu, \nu) = \frac{1}{2} \left( W_2^2(\hat{\mu}_n, \hat{\nu}_n) + W_2^2(\hat{\mu}'_n, \hat{\nu}'_n) - W_2^2(\hat{\mu}_n, \hat{\mu}'_n) - W_2^2(\hat{\nu}_n, \hat{\nu}'_n) \right),$$

where $\hat{\nu}_n$ and $\hat{\nu}'_n$ are based on two independent samples of size $n$ from $\mu$, and similarly for $\hat{\nu}_n$ and $\hat{\nu}'_n$ from $\nu$. This estimator is inspired by the contruction of distances in the maximum mean discrepancy family [31] and the Sinkhorn divergence [49]. Note that the 2-Wasserstein distance between finite samples can be computed conveniently with existing packages [25] that solves a linear program. Although the new estimator is not guaranteed to be unbiased across all settings, it is unbiased when the two distributions are the same.

Since our correction is based on a heuristic, the new estimator is still biased. To empirically characterize the effectiveness of the correction, we compute the discrepancy between the squared 2-Wasserstein distance for two continuous densities and the finite-sample estimate obtained from i.i.d. samples. When $\mu$ and $\nu$ are Gaussians with means $m_1, m_2 \in \mathbb{R}^d$ and covariance matrices $\Sigma_1, \Sigma_2 \in \mathbb{R}^{d \times d}$, we have the following convenient closed-form

$$W_2^2(\mu, \nu) = \|m_1 - m_2\|_2^2 + \mathrm{Tr}\left( \Sigma_1 + \Sigma_2 - 2(\Sigma_1^{1/2} \Sigma_2 \Sigma_1^{1/2})^{1/2} \right).$$

(a) different in mean      (b) different in covariance

Figure 2: Absolute value between $W_2^2(\mu, \nu)$ and the sample averages of estimators $\hat{W}_2^2$ (vanilla) and $\tilde{W}_2^2$ (corrected) for Gaussian $\mu$ and $\nu$. Darker curves correspond to larger number of samples used to compute the empirical estimate (ranging from 100 to 1000). (a) $m_1 = 0, m_2 = \mathbf{1}_d, \Sigma_1 = \Sigma_2 = I_d$. (b) $m_1 = m_2 = 0, \Sigma_1 = I_d, \Sigma_2 = I_d/2 + \mathbf{1}_d \mathbf{1}_d^\top / 5$.

We compare the vanilla estimate $\hat{W}_2^2(\mu, \nu, n)$ and the corrected estimate $\tilde{W}_2^2(\mu, \nu, n)$ by their magnitude of deviation from the true value $W_2^2(\mu, \nu)$:

$$\left| W_2^2(\mu, \nu) - \mathbb{E}[\hat{W}_2^2(\mu, \nu, n)] \right|, \quad \left| W_2^2(\mu, \nu) - \mathbb{E}[\tilde{W}_2^2(\mu, \nu, n)] \right|,$$

where the expectations are approximated via averaging 100 independent draws. Figure 2 reports the deviation across different sample sizes and dimensionalities, where $\mu$ and $\nu$ differ only in either mean or covariance. While the corrected estimator is not unbiased, it is relatively more accurate.

In addition, Figure 3 demonstrates that our bias-corrected estimator becomes more accurate as the two distributions are closer. This indicates that our proposed estimator may provide a more reliable estimate of the 2-Wasserstein distance when the sampling algorithm is close to convergence.

Figure 3: Absolute value between $W_2^2(\mu, \nu)$ and the sample averages of estimators $\hat{W}_2^2$ (vanilla) and $\tilde{W}_2^2$ (corrected) for Gaussian $\mu$ and $\nu$. Darker curves correspond to larger number of samples used to compute the empirical estimate (ranging from 100 to 1000). We fix $d = 20$ and interpolate the mean and the covariance matrix, i.e. $m = \alpha m_1 + (1 - \alpha)m_2, \Sigma = \alpha \Sigma_1 + (1 - \alpha)\Sigma_2, \alpha \in [0, 1]$. (a) $m_1 = 0, m_2 = 2\mathbf{1}_d, \Sigma_1 = \Sigma_2 = I_d$. (b) $m_1 = m_2 = 0, \Sigma_1 = 2I_d, \Sigma_2 = I_d/2 + \mathbf{1}_d\mathbf{1}_d^\top/5$.

## H  Additional Numerical Studies

In this section, we include additional numerical studies complementing Section 5.

### H.1  Strongly Convex Potentials

We first include additional plots of error estimates in $W_2$ and the energy distance for sampling from a Gaussian mixture and the posterior of BLR. The results indicate that the reduction in asymptotic error is consistent across problems with varying dimensionalities that we consider. In the end, we conduct a wall time analysis and show that SRK-LD is competitive in practice.

#### H.1.1  Additional Results

Figure 4 shows the estimated $W_2$ error as the number of iterations increase for the 2D and 20D Gaussian mixture and BLR problems with the parameter settings described in Section 5. We observe consistent improvement in the asymptotic error across different settings in which we experimented.

(a) Gaussian Mixture (2D)     (b) Gaussian Mixture (20D)     (c) BLR (20D)

Figure 4: Error in $W_2^2$ for strongly log-concave sampling. Legend denotes "scheme (step size)".

In addition to reporting the estimated squared $W_2$ values, we also evaluate the two schemes by estimating the energy distance [58, 59] under the Euclidean norm. For probability measures $\mu$ and $\nu$ on $\mathbb{R}^d$ with finite first moments, this distance is defined to be the square root of

$$D_E(\mu, \nu)^2 = 2\mathbb{E}\left[\|Y - Z\|_2\right] - \mathbb{E}\left[\|Y - Y'\|_2\right] - \mathbb{E}\left[\|Z - Z'\|_2\right], \tag{49}$$

where $Y, Y' \overset{\text{i.i.d.}}{\sim} \mu$ and $Z, Z' \overset{\text{i.i.d.}}{\sim} \nu$. The moment condition is required to ensure that the expectations in (49) is finite. This holds in our settings due to derived moment bounds. Since exactly computing

the energy distance is intractable, we estimate the quantity using the following (biased) V-statistic [55]

$$\hat{D}_E(\mu,\nu)^2 = \frac{2}{mn}\sum_{i=1}^{m}\sum_{j=1}^{n}\|Y_i - Z_j\|_2 - \frac{1}{m^2}\sum_{i=1}^{m}\sum_{j=1}^{m}\|Y_i - Y_j\|_2 - \frac{1}{n^2}\sum_{i=1}^{n}\sum_{j=1}^{n}\|Z_i - Z_j\|_2,$$

where $Y_i \overset{\text{i.i.d.}}{\sim} \mu$ for $i = 1,\ldots,m$ and $Z_j \overset{\text{i.i.d.}}{\sim} \nu$ for $j = 1,\ldots,n$. Figure 5 shows the estimated energy distance as the number of iteration increases on a semi-log scale. We use 5k samples each for the Markov chain and the target distribution to compute the V-statistic, where the target distribution is approximated following the same procedure as described in Section 5.1. These plots show that SRK-LD achieves lower asymptotic errors compared to the EM scheme, where the error is measured in the energy distance. This is consistent with the case where the error is estimated in $W_2^2$.

Figure 5: Error in $D_E^2$ for strongly log-concave sampling. Legend denotes "scheme (step size)".

### H.1.2 Asymptotic Error vs Dimensionality and Step Size

Figure 6 (a) and (b) respectively show the asymptotic error against dimensionality and step size for Gaussian mixture sampling. We perform least squares regression in both plots. Plot (a) shows results when a step size of $0.5$ is used. Plot (b) is on semi-log scale, where the quantities are estimated for a 10D problem.

### H.1.3 Wall Time

Figure 7 shows the wall time against the estimated $W_2^2$ of SRK-LD compared to the EM scheme for a 20D Gaussian mixture sampling problem. On a 6-core CPU with 2 threads per core, we observe that SRK-LD is roughly $\times 2.5$ times as costly as EM per iteration. However, since SRK-LD is more stable for large step sizes, we may choose a step size much larger for SRK-LD compared to EM, in which case its iterates converge to a lower error within less time.

### H.2 Non-Convex Potentials

We first discuss how we approximate the iterated Itô integrals, after which we include additional numerical studies varying the dimensionality of the sampling problem.

Figure 6: Asymptotic error vs dimensionality and step size.

Figure 7: Wall time for sampling from a 20D Gaussian mixture.

### H.2.1 Approximating Iterated Itô Integrals

Simulating both the iterated Itô integrals $I_{(l,i)}$ and the Brownian motion increments $I_{(i)}$ exactly is difficult. We adopt the Kloeden-Platen-Wright approximation, which has an MSE of order $h^2/n$, where $n$ is the number of terms in the truncation [33]. The infinite series can be written as follows:

$$I_{(l,i)} = \frac{I_{(l)}I_{(i)} - h\delta_{li}}{2} + A_{(l,i)},$$

$$A_{(l,i)} = \frac{h}{2\pi}\sum_{k=1}^{\infty}\frac{1}{k}\left(\xi_{l,k}\left(\eta_{i,k} + \sqrt{2/h}\Delta B_h^{(i)}\right) - \xi_{i,k}\left(\eta_{l,k} + \sqrt{2/h}\Delta B_h^{(l)}\right)\right),$$

where $\xi_{l,k}, \xi_{i,k}, \eta_{i,k}, \eta_{l,k} \overset{\text{i.i.d.}}{\sim} \mathcal{N}(0,1)$. $A_{(l,i)}$ is known as the *Lévy area* and is notoriously hard to simulate [66].

For SDE simulation, in order for the scheme to obtain the same strong convergence order under the approximation, the MSE in the approximation of the Itô integrals must be negligible compared to the local mean-square deviation of the numerical integration scheme. For our experiments, we use $n = 3000$, following the rule of thumb that $n \propto h^{-1}$ [33]. Although simulating the extra terms can become costly, the computation may be vectorized, branched off from the main update, and parallelized on an additional thread, since it does not require any information of the current iterate.

Wiktorsson et al. [66] proposed to add a correction term to the truncated series, which results in an approximation that has an MSE of order $h^2/n^2$. In this case, $n \propto h^{-1/2}$ terms are effectively required. We note that analyzing and comparing between different Lévy area approximations is beyond the scope of this paper.

### H.2.2 Additional Results

Figure 8 shows the MSE of simulations starting from a faithful approximation to the target. We adopt the same simulation settings as described in Section 5.2. We observe diminishing gains as the dimen-

sionality increases across all settings with differing $\beta$ and $\gamma$ parameters in which we experimented. These empirical findings corroborate our theoretical results. Note that the corresponding diffusion in all settings are still uniformly dissipativity, yet the potential may become convex when $\beta$ is large. Nevertheless, the potential is never strongly convex when $\beta$ is positive due to the linear growth term.

(a) $\beta = 2, \gamma = 0.5$

(b) $\beta = 3, \gamma = 0.5$

Figure 8: MSE for non-convex sampling.

## Footnotes

[6]There is slight ambiguity in terms of which iteration's interpolated values should $\tilde{H}_1$ and $\tilde{H}_2$ correspond to. For notational simplicity, we have avoided using a subscript or superscript for the iteration index $k$, and almost always make $\tilde{H}_1$ and $\tilde{H}_2$ appear along with the original iterate $\tilde{X}_k$.