[Reviews · NeurIPS 2019]

Reviewer 1



The full paper is much longer and compressing it into an 8-page NeurIPS version makes for a terse read; I would advise the authors to summarize certain results so as to make the shorter version easier to read. 1. The discretization in (13) and (14) are given out of the blue. A derivation starting from (6) will be helpful. 2. Line 177: Theorem 1 should say “has _uniform_ local deviation orders”? 3. Section 4.2 is scant on details. It was not clear to me upon a first reading what the non-convex potential is, especially since Section 4.1 uses overdamped Langevin equation with constant diffusion for a strongly-convex potential. 4. Can you plot the dependence of W-2 with dimension in Figure 1? This will help ascertain the improved convergence rate numerically. Similarly, you should use the number of gradient evaluations as the X-axis in Fig. 1b. The SRK scheme uses three gradient evaluations whereas the EM scheme uses one. 5. How do you compute the MSE in Fig. 1c? Why does it increase with the number of iterations?

Reviewer 2



Thanks to the authors for responding to my questions. I am content with the submission and have kept my score at 8. ========= This work proposes a Stochastic Runge Kutta discretization of the overdamped Langevin Monte Carlo chain and establishes a d/eps^{2/3} mixing rate (upto eps accuracy in Wasserstein distance) for d-dimensional strongly log concave distributions which is 1/eps^{1/3} faster than the previously best known rate of order d/eps for the unadjusted Langevin algorithm. They also establish speed-ups for low dimensional non-convex cases when the Hessian operator norm grows as c+sqrt{||x||}. In fact their result in Theorem 1 is general enough to apply to general discretization schemes that satisfy certain uniform (in iteration) local deviation order. The authors note that establishing this condition uniformly is non-trivial in general and in their case, they proceed by bounding the Markov chain moments carefully. Overall, the paper is pretty well written with clearly stated results, contributions, and highlights about the key ideas. In their numerical experiments, they use same step size for ULA and SRK and show that SRK-LD converges to a smaller asymptotic bias (which is consistent with their results). I have three remarks: — If would be nice if the authors could clearly point out the condition number dependency in their theorem 2. — What are some other possible ways to establish a uniform deviation condition? — Do we expect any benefit with such higher-order discretization? (it would have to be in dimension dependency since the mixing rates already depend only logarithmically in epsilon, e.g., Theorem 2 of https://arxiv.org/pdf/1905.12247v1.pdf).

Reviewer 3



After Rebuttal: Thank you for the responses. I that believe the paper will be even stronger with the inclusion of the stochastic gradient-variant. ########################################################### Originality: Given an integrator with some local deviation orders, the paper provides a generic theorem to obtain finite-time Wasserstein bounds. This is a very valuable theorem, which will be useful for other theoreticians working in this field. On the other hand, to the best of my knowledge, this is the first paper that uses a stochastic Runge-Kutta integrator for sampling from strongly log-concave densities with explicit guarantees. The authors further show that their proposed numerical scheme improves upon the existing guarantees when applied to the overdamped Langevin dynamics. Quality: On the theoretical side, I have gone through the proofs; however, I didn’t go over them line by line. The proofs are overall clearly written and the roadmap is clear. On the experimental side, I think the experiments are sufficient since the main contributions of this paper are theoretical. However, I must admit that I don’t believe that the experimental results are charming so that a practitioner would decide to use this integrator in their applications. In this line of research, I think the authors should mention the following paper: “Stochastic Gradient Richardson-Romberg Markov Chain Monte Carlo”, Durmus et al, Neurips 2016, where the authors aim at obtaining a higher-order integration scheme for the overdamped Langevin equation. Clarity: The paper is very-well written and organized. I enjoyed reading it. Significance: Sampling methods based on diffusions have been a major subject in large-scale Bayesian machine learning and computational statistics. In this context the paper is very well-placed and is of great interest to the community. I don’t believe that the paper proposes a practical algorithm that could replace the existing simpler methods, but it has strong theoretical contributions, which should be sufficient for this conference proceeding. A small criticism in terms of significance could be the following. What makes diffusion-based sampling techniques interesting to the ML community is their ability to scale up to large problems since they enable the possibility of replacing full gradients with stochastic gradients. In that respect, an algorithm which requires full gradients is not very interesting to a large portion of the ML community. In the current version of this paper, it is not clear at all what would happen if we apply the same ideas to SGLD. To make the algorithm more appealing for the ML community, I suggest the authors to at least add a discussion for the case where the full gradients are replaced with stochastic ones.

[Author Response · NeurIPS 2019]

We thank all reviewers for their valuable and constructive feedback. Below are our responses.

**Revs. 1,2,3 - Formatting & Citation:**  We will fix any typos, cite the reference (Durmus et al., 2016), clarify the

definitions of $\mathrm{d}B_u^{(i)}$, $\bar{X}_t$, and "smoothness", supply a reference for eq (6), and correct the reference letter case issue.

**Revs. 1,3 - Eq (13-14):**  We will provide a short instructive derivation for eqs (13-14) in the appendix.

**Rev. 1 - Sec. 4.2:**  We will add more clarification on the class of non-convex potentials at the beginning of the section.

**Rev. 1 - dimension dependence:**  Figure on the right shows the asymptotic bias vs

dimensionality for a Gaussian mixture problem. We estimated $W_2$ with the Monte

Carlo estimator detailed in App. F and fitted a line with least squares.

**Rev. 1 - x-axis in Fig. 1(b):**  The original motivation was to showcase the evolution

of the distance of SRK-LD iterates to the target distribution across iterations. The key

point made is that SRK-LD produces a sequence of iterates with lower asymptotic

bias than that produced by EM. However, we agree that a refined measure is required

to demonstrate the algorithm's performance in terms of computational resource usage.

To this end, we performed a wall time analysis, whose results were included in App.

G.1.3 of our original submission. To summarize the findings, SRK-LD is 2.5-3 times

as costly as EM (on a CPU) per iteration. However, the scheme can still beat EM in terms of total cost if we select a

much larger step size for it than that for EM. Recall, Sec. 5.1 and Fig. 1(a) demonstrated that SRK-LD can be run with

large step sizes, at which running EM results in divergence.

**Rev. 1 - MSE in Fig. 1c:**  We simulate with EM using a small step size of $10^{-6}$ until we obtain an initial batch

of particles roughly distributed as the target distribution. We evolve this same initial batch in three ways: *(i)* using

EM with a small step size of $10^{-6}$, *(ii)* using SRK-ID with a step size of $10^{-3}$, and *(iii)* using EM with a step size

of $10^{-3}$. We treat *(i)* as the true continuous-time process and estimate the MSE of EM and SRK-ID at each iteration

with the formula $\frac{1}{N}\sum_{i=1}^{N}\|X_t^{(i)} - X_{hk}^{(i)}\|$, where $X_t$ is the continuous-time process computed by *(i)*, and $X_{hk}$ is the

discrete-time Markov chain (either *(ii)* or *(iii)*). Here, $hk = t$, $h$ is the step size, and $k$ is the iteration index. The

superscript is used to index entries in a batch with $N$ samples. The MSE increases since we started off from an initial

batch that is roughly distributed as the target distribution, and the discrete-time Markov chains *(ii)* and *(iii)* each has

their own biases. The purpose was to demonstrate that *(ii)* has a smaller bias than *(iii)*. Note the setting was detailed in

Section 5.2 and App. G.2 in our original submission. We preferred to show these plots as opposed to error-vs-iteration

plots since we found it hard to choose a good initialization such that neither method took very long to converge.

**Rev. 2 - condition number dependence in Thm. 2:**  A convergence bound where the exponent of the Lipschitz

constant $L$ in the numerator matches up with the exponent of the strong convexity constant $m$ in the denominator would

give us insight into the convergence behaviour at varying worst-case curvature cases. Yet, the two exponents don't

match up in our bound (such type of bounds aren't uncommon; see e.g. [17, Thm. 5]). Plugging constants $C_2$ and $C_3$

into (35) shows that the dominating term decided by the condition number is $\mathcal{O}(\kappa^{5/3})$, where $\kappa = L/m$. However, our

(slightly pessimistic) bound has additional terms dependent on smoothness with the dominant term being $\mathcal{O}(1 \vee L^6)$.

**Rev. 2 - uniform deviation:**  Local deviation orders are well-studied in the SDE literature typically using the Itô-

Taylor expansion on the continuous process [38, 42]. In our case, to unroll the recursion and obtain a converging

bound, it is instructive to rely on contraction of diffusion, high-order Lipschitz smoothness, and Markov chain moment

bounds. We chose the more hands-on approach for tight constants, but note there are established results without explicit

constants [42, Lem. 2.2.2]. We believe our approach might be the most straightforward in deriving uniform bounds.

**Rev. 2 - benefit with higher-order discretization:**  Indeed, Metropolis-adjusted algorithms allow exact sampling

with fast convergence rates. We thank the reviewer for the pointer to Chen et al., 2019; we will discuss this in the

final version. On the other hand, to the best of our knowledge, there has been limited success in modifying these

methods to take advantage of data subsampling, which is typically required for scaling to large datasets. By contrast,

Bayesian learning has witnessed considerable success in practice with SGLD [54], a gradient-subsampled version

of the (asymptotically biased) gradient Langevin dynamics algorithm. We may expect direct discretizations (w/o

Metropolis-adjustment) of exponentially contracting diffusions to take advantage of the data subsampling paradigm.

A systematic study on this topic warrants an independent investigation. Yet, SRK-LD can be easily adapted to this

scenario using existing techniques in the literature; see the next comment for more on this.

**Revs. 2,3 - stochastic oracle:**  Thank you for this recommendation. We assume an oracle model where *(i)* the

randomness in the oracle is independent of that of the Brownian motion, *(ii)* the stochastic gradient $\hat{\nabla}f$ is unbiased,

i.e. $\mathbb{E}[\hat{\nabla}f(x)] = \nabla f(x)$ for all $x \in \mathbb{R}^d$, and *(iii)* the stochastic gradient $\hat{\nabla}f$ has bounded variance at iterates of the

Markov chain and the "interpolated" random variables, i.e. $\mathbb{E}[\|\nabla f(Y)\|_2^2] \leq d\sigma^2$, where $Y$ may be $\tilde{X}_k$, $\tilde{H}_1$ or $\tilde{H}_2$.

Compared to the full gradient case, the key differences in this setting are *(a)* Markov chain moment bounds become

slightly "inflated", and *(b)* local deviations have extra terms dependent on the variance of the oracle. Note that only

*(b)* affects the rate's dependence on $\epsilon$. Under assumptions *(i,ii,iii)*, one may show that SRK-LD achieves the same

convergence rate when the variance $d\sigma^2 = \mathcal{O}(\epsilon^2/h)$ for step size $h$. On the other hand, we recover the rate of $\tilde{\mathcal{O}}(d\epsilon^{-2})$

for gradient Langevin dynamics if the variance is large. The analysis and proof are mostly mechanical, following along

similar lines of [12]. We will include a detailed discussion on this result.

[Meta-Review · NeurIPS 2019]

This paper presents a unified mathematical framework for quantifying the "acceleration phenomenon" in Langevin Monte Carlo schemes. Specifically, the formalism developed in this work allows one to transfer rapid convergence results for a suitable Ito diffusion to its discretized counterpart using Runge-Kutta numerical integration. The results are of high technical quality, and the ideas presented in the paper open up a number of promising directions for further research. All three reviewers are uniformly enthusiastic about this work, which makes a strong contribution to a topic of considerable current interest.